# Molecular near-infrared triplet-triplet annihilation upconversion with eigen oxygen immunity

Xinyu Wang[1,4], Fangwei Ding[1,4], Tao Jia[1], Feng Li[1], Xiping Ding[1], Ruibin Deng[1], Kaifeng Lin[1], Yulin Yang [1], Wenzhi Wu [2], Debin Xia [1] ✉ & Guanying Chen [1,3] ✉

Molecular triplet-triplet annihilation upconversion often experiences drastic luminescence quenching in the presence of oxygen molecules, posing a significant constraint on practical use in aerated conditions. We present an oxygen-immune near-infrared triplet-triplet annihilation upconversion system utilizing non-organometallic cyanine sensitizers ($\lambda_{ex} = 808$ nm) and chemically synthesized benzo[4,5]thieno[2,3-b][1,2,5]thiadiazolo[3,4-g]quinoxaline dyes with a defined dimer structure as annihilators ($\lambda_{em} = 650$ nm). This system exhibits ultrastable upconversion under continuous laser irradiance (>480 mins) or extended storage (>7 days) in aerated solutions. Mechanistic investigations reveal rapid triplet-triplet energy transfer from sensitizer to annihilators, accompanied by remarkably low triplet oxygen quenching efficiencies ($\eta_{O_2} < 13\%$ for the sensitizer, <3.7% for the annihilator), endowing the bicomponent triplet-triplet annihilation system with inherent oxygen immunity. Our findings unlock the direct and potent utilization of triplet-triplet annihilation upconversion systems in real-world applications, demonstrated by the extended and sensitive nanosensing of peroxynitrite radicals in the liver under in vivo nitrosative stress.

Triplet–triplet annihilation (TTA) upconversion (TTA-UC) is an important means to implement anti-stokes wavelength conversion using low incoherent light irradiance (<100 mW/cm²)[1–3]. This photochemical process typically involves a pair of sensitizer/annihilator molecules, in which annihilator molecule in triplet state ($T_1$) receives energy from the $T_1$ of sensitizer, and then undergoes TTA (or triplet fusion) process to populate its higher energy singlet state ($S_1$) that produces upconverted emissions[4–6]. A precise control of energy level structures of sensitizers and annihilators allows a precise modulation of both excitation and emission wavelengths, fueling a broad spectrum of technological applications ranging from photonics to biophotonics[7–9]. In particular, near-infrared (NIR)-to-visible upconversion finds wide uses in fields, from solar cells to photoredox catalysis,

and from biological imaging to optogenetics, because NIR light is abundant in the solar spectrum and silent to biological tissues[6,10–13]. Yet, limited number of organometallic triplet photosensitizers are identified to perform TTA-UC with red or NIR incident light excitation (600–800 nm), due to the exponential increase in non-radiative losses in sensitizers with smaller energy gaps[12,14–17]. The use of inorganic semiconductor quantum dots as sensitizers is able to perform triplet-to-triplet energy transfer (TTET) to annihilators at longer excitation wavelengths (>800 nm)[18–21].

Nevertheless, current TTA-UC processes, ever since the first observation in 1962[22], commonly suffer from drastic emission quenching in aerated milieu, as photoexcited molecular triplet states can be substantially depleted by ground state triplet oxygen ($^3O_2$)

[1]School of Chemistry and Chemical Engineering, Harbin Institute of Technology, Harbin, China. [2]School of Electronic Engineering, Heilongjiang University, Harbin, China. [3]Key Laboratory of Micro-systems and Micro-structures, Ministry of Education, Harbin Institute of Technology, Harbin, China. [4]These authors contributed equally: Xinyu Wang, Fangwei Ding. ✉e-mail: xia@hit.edu.cn; chenguanying@hit.edu.cn

molecules that are pervasive in real-world environments[23]. To solve the problem of oxygen quenching, attempts have been made to sustain upconversion in air by dispersing TTA-paired molecules into high-viscosity organic solvents or polymer matrix that can efficiently isolate them from surrounding oxygen molecules, or introducing anti-oxidants to consume diffusing oxygen molecules before quenching upconversion[24–28]. Yet, the inherent problem of oxygen quenching in TTA systems persists, posing a grand challenge to long-term technological applications in the real world.

Here, we develop a class of NIR-excited TTA-UC with eigen oxygen immunity, in which nonorganometallic IR806 molecules with large extinction coefficient were utilized as sensitizers ($\varepsilon = 2.1 \times 10^5$ M$^{-1}$ cm$^{-1}$ at 808 nm), while a series of chemically synthesized BTTQD

(benzo[4,5]thieno[2,3-b][1,2,5]thiadiazolo[3,4-g]quinoxaline) dyes with defined dimer structures and modest photoluminescence quantum yield (PLQY, 55, 63, and 51% for BTTQD 1–3) as annihilators (Fig. 1a). We unraveled that the low triplet oxygen quenching efficiencies and high oxidation potentials of both the sensitizer and annihilator underpin the observed highly stable upconversion under long-term continuous laser irradiance ($\lambda_{ex} = 808$ nm) or storage in aerated milieu.

## Results and discussion

Aromatic dimer compounds can produce corresponding photophysical processes due to the strong interaction between two monomers at close distances[8,29,30], rendering them suitable as annihilator

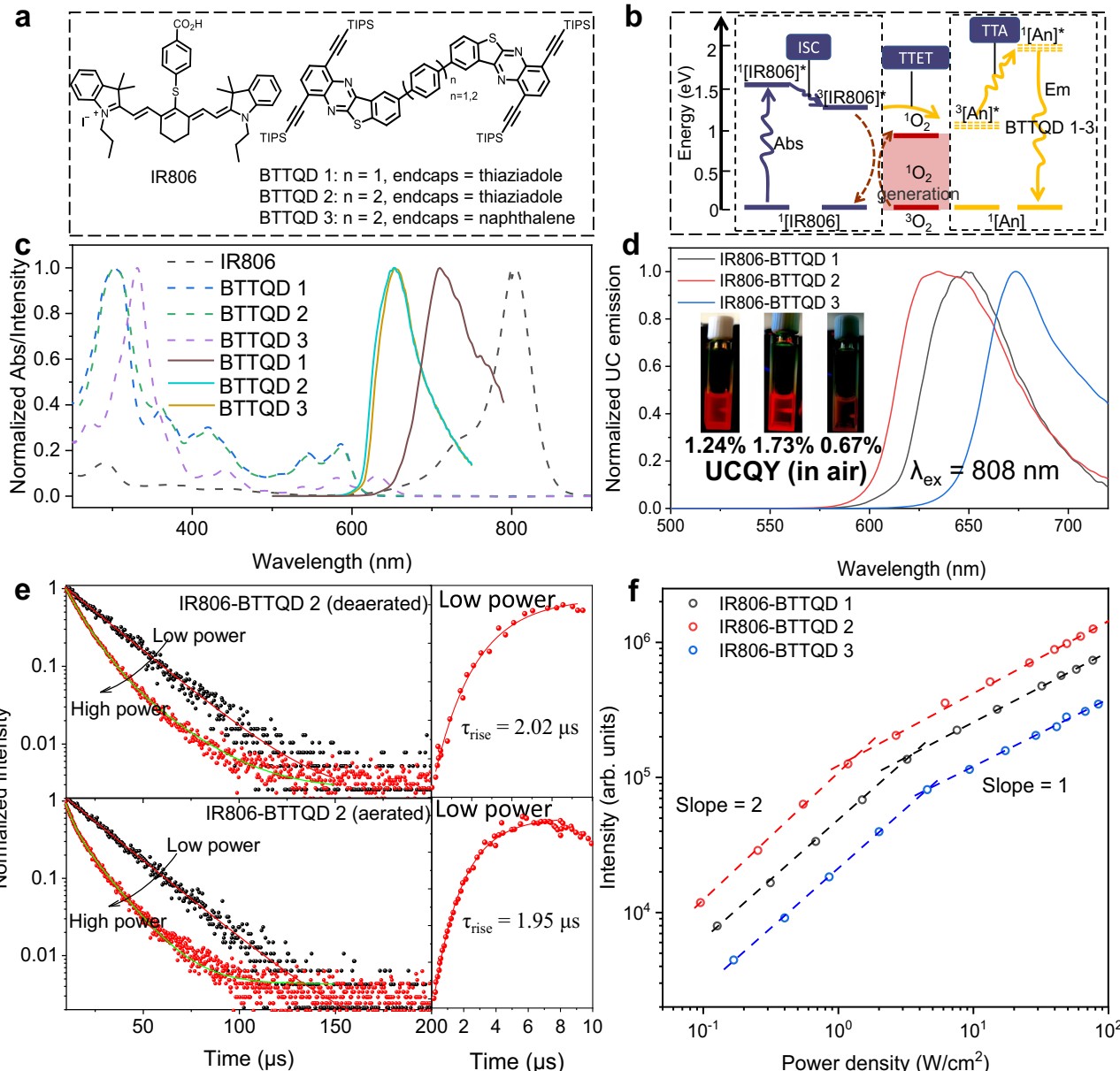

**Fig. 1 | Triplet–triplet annihilation upconversion (TTA-UC) of the IR806-BTTQDs systems. a** Chemical structures of IR806 and BTTQD 1–3. BTTQD 1 contains a phenyl linker and thiadiazole end cap, while BTTQD 2 contains a biphenyl bridge and thiadiazole end cap, contrasted with BTTQD 3 contains a biphenyl bridge and naphthalene end cap. **b** Schematic illustration of involved TTA-UC processes in the IR806-BTTQD 1–3 systems. ISC intersystem crossing, TTET triplet-to-triplet energy transfer, TTA triplet-to-triplet annihilation. **c** Normalized absorption and emission spectra of IR806 and BTTQD 1–3 in aerated chloroform. **d** Normalized upconversion emission spectra of IR806-BTTQD 1–3 solutions (in chloroform) with corresponding photographic images shown in the inset ($\lambda_{ex} = 808$ nm, 10 W/cm$^2$). **e** Time-resolved luminescence decay of TTA-UC emission of IR806-BTTQD 2 at 650 nm in deaerated (top) and aerated (below) solutions under low- and high-power excitation, with corresponding rising profiles shown on the right. low power: 0.02 mJ per pulse; high power: 1 mJ per pulse, pulse width: 25 kHz. **f** Logarithmic plots of upconversion emission intensities against excitation power densities for IR806-BTTQD 1–3 in aerated chloroform. **c–f** $c_{IR806} = 1 \times 10^{-5}$ M, $c_{BTTQD\ 1-3} = 1 \times 10^{-4}$ M.

with enhanced triplet fusion for TTA-UC[29]. We therefore synthesized three azaarene dimers of BTTQDs with near-identical structures but distinct linker lengths and cap ends through palladium-catalyzed Suzuki reaction (Supplementary Information on BTTQDs synthesis and Supplementary Figs. 1–10), and adopted them as triplet annihilators (Fig. 1a). Different endcaps have varied electron-withdrawing ability allowing fine tuning of molecular orbital energy levels, while heteroatoms inserted into the framework of polycyclic aromatic hydrocarbons enhance the molecular structural stability of the dimers (Supplementary Fig. 11). Single crystal x-ray structure studies reveal that BTTQD 1 form a one-dimensional chain structure by sliding stacking. In the one-dimensional chain, π-π interaction exists between the bending parts of adjacent molecules, and the π plane distance is 3.32 Å. Meanwhile, because of the introduction of the thiadiazole terminal group, N−S and N−N interactions existed among the molecular chains in the same layer, and N−S interaction distance was 3.10 Å and N−N interaction distance was 3.05 Å (Supplementary Fig. 12). The abundant interaction makes the stacking structure of crystals more stable. The crystal data and structure refinements of BTTQD 1 are summarized in Supplementary Table 3. Density functional theory results clearly showed that BTTQD 1–3 have HOMO and LUMO levels with varied electron density distribution (significantly affected by the phenyl and biphenyl linkers), and have corresponding energy band gaps of 2.19, 2.20, and 2.50 eV (Supplementary Figs. 13–15), consistent with cyclic voltammetry experiment results (Supplementary Fig. 16) and measured absorption spectra (Fig. 1c). Several studies have shown that LUMO energy levels < −3.7 eV can improve the ambient stability of organic molecules[31–34], and LUMOs of BTTQD 1–3 were calculated to be −3.79, −3.78, and −3.55 eV, respectively, implicating their ambient stabilities (Supplementary Table 4). Importantly, the doubled energy of the $T_1$ state of BTTQDs are higher than that of the $S_1$ of BTTQDs ($2T_1 > S_1$), priming their uses as triplet annihilators.

Cyanine dyes have been frequently used as exogenous molecular probes for fluorescence bioimaging due to their high molar extinction coefficient and easily modified chemical and optical properties[35–37]. Lengthening the length of polymethylene chain can extend fluorescence into NIR region[38], while installing a rigid ring system (such as cyclohexenyl) at the center of the molecular chain can enhance the photostability, as experimentally shown in Supplementary Fig. 17 (see refs. 39–41). We demonstrated IR806 cyanine dye ($\lambda_{abs}$ = 806 nm) is such a case and has an efficient intersystem crossing, allowing them to be used as triplet photosensitizer of rubrene for NIR-to-visible TTA-UC ($\lambda_{ex}$ = 808 nm, $\lambda_{em}$ = 580 nm)[25]. Note that IR806 has a higher $T_1$ state of (1.27 eV) than that of BTTQD 1–3 (1.10, 1.17, and 1.08 eV for BTTQD 1–3, respectively), favoring exothermic TTET transfer[13,23].

We therefore utilized IR806 as sensitizer and BTTQD 1–3 as annihilator (Fig. 1a) to study NIR TTA-UC in aerated environments. Photoexcitation at 808 nm indeed induced intense upconversion emissions centered at 640, 650, and 690 nm in the mixture solution of IR806 and BTTQD 1–3 (dissolved in aerated chloroform) (Fig. 1d). The observed long rise time, with a timescale about 2 μs in both aerated and deareated samples, signifies the taking place of triplet fusion upconversion (Fig. 1e)[18]. At low excitation power, the TTA-UC intensity decays exponentially, with a characteristic decay time $\tau_{UC}$ = 21.4 and 20.6 μs for deaerated and aerated solutions, respectively. In this case, the primary depoapulation channel of annihilator triplet is the spontaneous radiative/non-radiative decay (negligible TTA), and the triplet lifetime of the annihilator could be estimated to be twice $\tau_{UC}$ ($2\tau_{UC} = \tau_T^A$), giving a $\tau_T^A$ of ~40 μs[42–44]. At high excitation power, an elevated concentration of annihilator triplets induce efficient TTA process, resulting in a shortened TTA-UC lifetime, as shown in Fig. 1e[45].

As both TTET and TTA processes are sensitive to the annihilator concentration[29], we varied the concentration of BTTQDs at an invariable concentration of IR806 (10 μM), and determined its optimal concentration to be 80, 100, and 70 μM for BTTQD 1, BTTQD 2, and

BTTQD 3, respectively (Supplementary Fig. 18). This means that about one magnitude higher annihilator concentration than sensitizer concentration is needed to maximize TTA-UC. Threshold excitation light irradiance is an important parameter to characterize TTA-UC, at which the dependence of upconversion emission on incident light irradiance transitions from quadratic to linear. At the threshold, triplet fusion process begins to surrogate first-order loss processes as the dominant process to depopulate the triplets. As such, the triplet population varies with the square root of excitation photon flux, resulting in a linear dependence of upconversion on pump power and thereby reaching a saturated maximum upconversion brightness. Figure 1f shows the dependence of upconversion emission intensity on the excitation power density, in which a cross-over point was observed at 3.5, 1.8, and 5.7 W/cm² for BTTQD 1, BTTQD 2, and BTTQD 3, respectively. We evaluated the upconversion quantum yield ($\Phi_{UC}$) of IR806-BTTQD 1–3 to be 1.24%, 1.73%, and 0.67% (out of 50%, incident power density ≈ 10 W/cm²) in aerated chloroform. The achieved maximum $\Phi_{UC}$ in IR806-BTTQD 2 is about six times higher than that of the reported IR806-rubrene system (threshold: 3.6 W/cm²)[25], and comparable to that using organometallics as triplet sensitizer[13,46]. One factor that cannot be ignored is the back energy transfer from BTTQD 2 to IR806 that quenches TTA-UC. We evaluated the back energy transfer efficiency ($\Phi_{FRET}$) to be 12.5% (Supplementary Fig. 19), therefore determining the internal upconversion quantum yield, $\Phi_{UC, g} = \Phi_{UC}/(1 − \Phi_{FRET})$ to be ~2% in aerated cholorform[47]. Note that PLQY of BTTQD 2 was decreased about twice (~27.0%) at $c_{BTTQD\ 2} = 1 \times 10^{-4}$ M, compared to that of the diluted solution (Supplementary Fig. 20). Further approaches to maintain annihilator PLQYs at high concentrations can further increase $\Phi_{UC}$. We would like to emphasize that TTA-UC is also feasible using the monomer of BTTQD 2, which exhibits comparable singlet and triplet energy levels (Supplementary Table 4). However, it results in a twofold lower $\Phi_{UC}$ of ~1.06% and a twofold higher TTA threshold of around 4 W/cm² (Supplementary Fig. 21). This rationalizes the preference for utilizing the BTTQD 2 dimer as the annihilator, possibly attributed to the dimer having two simultaneously available triplet groups under photoexcitation[29].

We then studied the chemical stability of the TTA-UC systems in air, in which the sensitizer and annihilators were mixed in chloroform in air (without degassing treatment). We utilized the most well-known PtOEP-DPA TTA-UC system in the visible range and our previously established IR806-rubrene TTA-UC system in the NIR range as two controls for comparison. Emission intensities of IR806-BTTQDs were retained (by more than 80%) even after storing in air for 7 days (Fig. 2a), with no noticeable solution color change (Fig. 2b). In sharp contrast, serious oxygen quenching was observed in both IR806-rubrene and PtOEP-DPA TTA-UC systems, resulting in a nearly complete upconversion quenching after storing in air in less than 2 days and 1 day, respectively (Fig. 2a). The oxygen sensitivity could also be seen through the time-evolved color change of TTA solution in air (Fig. 2b). We further vacuum-degassed IR806-BTTQD 2 solution which showed only a slightly improved TTA-UC stability than non-degassed counterpart over a 7-day scale storage, demonstrating the oxygen insensitivity of IR806-BTTQD 2 (Supplementary Fig. 22). The high chemical stability of IR806-BTTQDs solution in air promise their variety of applications in aerated environments pervasive in real world.

The effect of dioxygen on the TTA-UC photostability of IR806-BTTQD 2 sample was also investigated under prolonged continuous-wave laser irradiance at 808 nm. TTA-UC of IR806-BTTQD 2 in both aerated and vacuum-degassed solutions maintained >80% upconversion under laser irradiance for 8 h, showing high resistance to oxygen molecules (Fig. 2c). While a drastic TTA-UC decay was observed in both PtOEP-DPA and IR806-rubrene systems in aerated solution under prolonged laser irradiance, showing >90% luminescence loss in less than 20 min and 200 min, respectively. Degassing of PtOEP-DPA and IR806-rubrene solutions significantly improved the photostability,

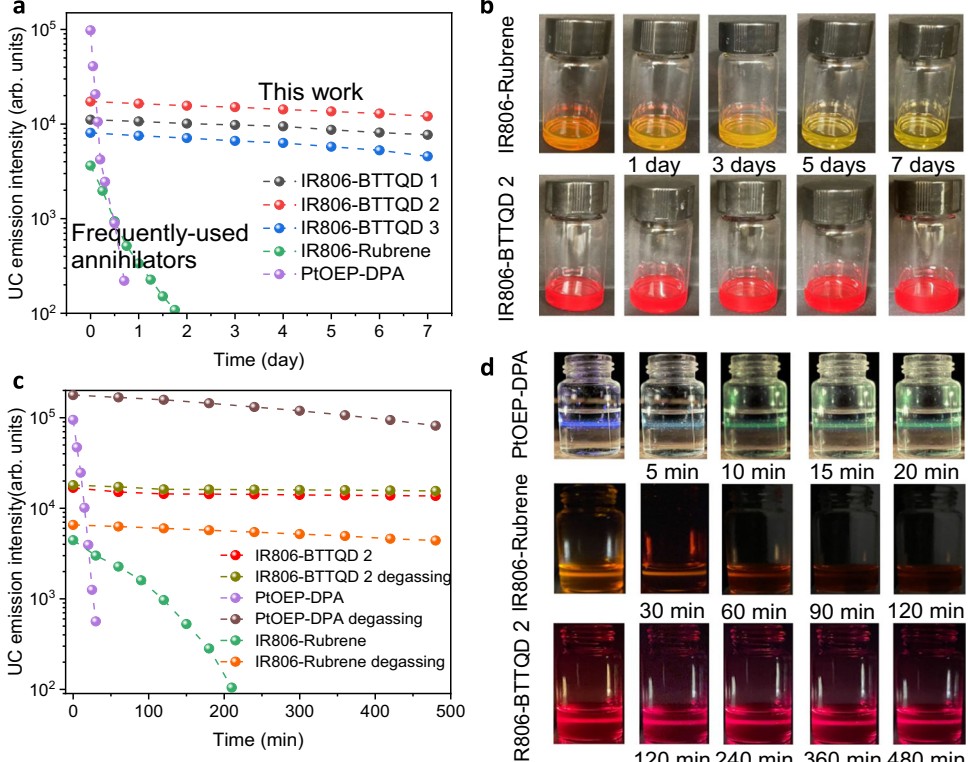

**Fig. 2 | Stability and photostability of the TTA-UC systems in air. a** The changes of upconversion emission intensities of IR806-BTTQD 1–3, IR806-rubrene, and PtOEP-DPA solutions during 7 days storage in air. **b** Photographic images of IR806-BTTQD 2 and IR806-rubrene solution over the 7-day storage in air. **c** Emission stabilities of upconversion in IR806-BTTQD 2, IR806-rubrene, and PtOEP-DPA solutions with/without degassing under continuous-wave laser irradiation at 808 or 532 nm. **d** Photographic images of TTA upconversion emission in (**c**) taken at varied time in aerated solution. IR806-BTTQD 1–3 and IR806-rubrene were dissolved in chloroform, while PtOEP-DPA was dissolved in tetrahydrofuran with concentrations of $c_{IR806} = 1 \times 10^{-5}$ M, $c_{BTTQD} = 1 \times 10^{-4}$ M, $c_{Rubrene} = 1 \times 10^{-4}$ M, $c_{PtOEP} = 1 \times 10^{-4}$ M, $c_{DPA} = 1 \times 10^{-2}$ M. For IR806-BTTQD 1–3 and IR806-rubrene, $\lambda_{ex} = 808$ nm, power density = 100 mW/cm², while for PtOEP-DPA, $\lambda_{ex} = 532$ nm, power density = 5 mW/cm². An optical filter with 750 nm short pass was placed in front of the camera (iPhone 12, Apple Inc.) to accurately acquire the TTA-UC luminescence of IR806-rubrene and IR806-BTTQ 2 systems in (**d**).

rendering them photostable over 8 h investigated here. The high oxygen-immune photostability of IR806-BTTQD 2 under laser irradiance could also be clearly observed in TTA-UC emission in aerated solution, in marked contrast to that of IR806-rubrene and PtOEP-DPA controls (Fig. 2d).

To further illustrate the phenomenon of oxygen immunity, we examined the change of $\Phi_{UC}$ for both IR806-BTTQD 2 and the IR806-rubrene control in Fig. 2c (Supplementary Fig. 23). The $\Phi_{UC}$ of IR806-BTTQD 2 remained consistent at ~1.7% before laser overexposure in both aerated and deaerated solutions. Subsequently, it displayed only a marginal 10% decrease after 48 h of laser irradiation in both aerated and deaerated solutions. This consistency aligns with the observed TTA-UC intensity change in Fig. 2c. Conversely, the $\Phi_{UC}$ of the IR806-rubrene control showed variation in aerated (0.2%) and deaerated (0.3%) solutions before laser overexposure, indicating pronounced oxygen-induced triplet quenching. Furthermore, the $\Phi_{UC}$ was observed to plummet to near-zero (close to a 100% decrease) in the aerated solution but remained relatively stable (with only a 23% decrease) in the deaerated solution after 48 h of laser irradiation. This distinctive trend suggests the instability of the molecular structure of the TTA pair in aerated IR806-rubrene solution during the TTA-UC process. In conclusion, oxygen molecules exert limited (or negligible) effects on $\Phi_{UC}$ and molecular structure stability in the developed IR806-BTTQD 2 system.

To reveal the underlying mechanism of oxygen immunity, we carried out ultrafast transient absorption (TA) spectra measurement on both IR806 and IR806-BTTQD 2 samples. A ground state bleaching (GSB) band of IR806 around 800–830 nm was observed in both samples (Fig. 3a, b). Decay of singlet excited state absorption (450–550 nm) was observed within 2 ns[48–50], coinciding with the growth of an absorption band centered at 585 nm, assigned to $T_1 \longrightarrow T_n$ transition of IR806 (Fig. 3a). Unlike the singlet excitation state absorption band, the GSB band of IR806 remained intense within the data acquisition time window, confirming the formation of a long-lived transient species, that is, the triplet state. A blue shift was observed for the excited state absorption around 550–650 nm in IR806-BTTQD 2 sample, which implies the generation of unknown transient species in this wavelength range. Indeed, femtosecond TA experiment on pure BTTQD 2 (with 580-nm excitation) showed a positive transient triplet excitation absorption band centered at 530 nm (Supplementary Fig. 24). Transient kinetics of IR806-BTTQD at 580 nm showed first rise and then decay processes (Fig. 3b, lower panel), consistent with transient peak maxima ascending and then descending with time (inset of Fig. 3b), verifying the occurrence of triplet−triplet energy transfer (TTET) from IR806 to BTTQD 2.

To probe the long-lived triplet state, nanosecond pulses were adopted to excite these samples, with observed triplet excitation absorption at 580 nm in both IR806 (Fig. 3c) and IR806-BTTQD 2 (Fig. 3d)[35,51,52]. Triplet state lifetimes of IR806 in aerated and deaerated solutions were determined to be 3.01 and 4.57 µs, respectively, showing oxygen quenching of IR806 triplets with a rate of $1.13 \times 10^5$ s⁻¹. Further, we studied the effect of oxygen molecules on TTET process from IR806 to BTTQD 2. In the presence of BTTQD 2, triplet state lifetimes of IR806 were measured to be 1.18 and 1.29 µs for aerated and

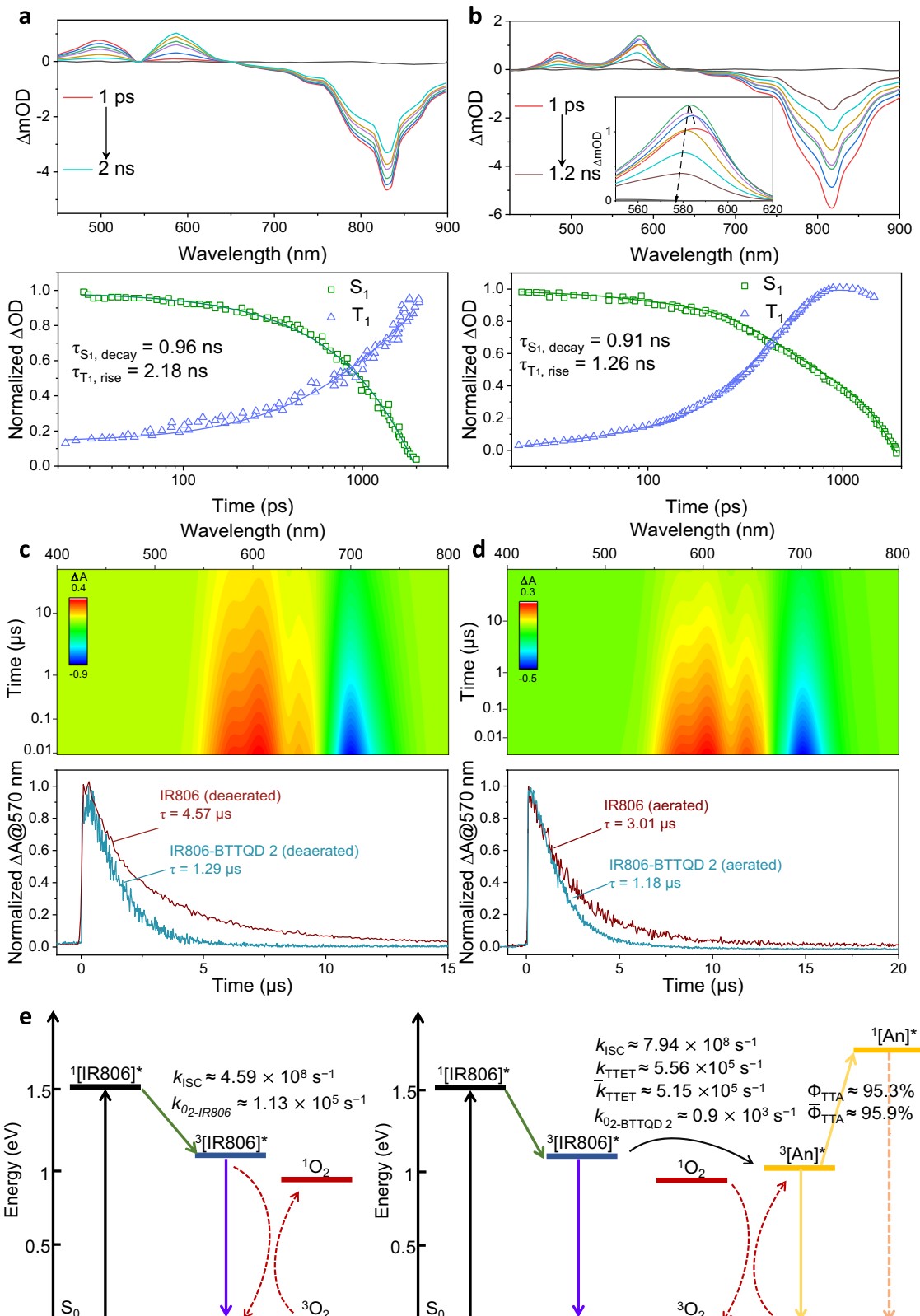

**Fig. 3 | Transient absorption spectroscopy studies of the IR806-BTTQD 2 TTA-UC system. a, b** Femtosecond transient absorption spectra of IR806 (**a**) and IR806-BTTQD 2 (**b**) in deaerated chloroform using 800 nm pulsed laser excitation (2 µJ, FWHM = 50 fs), with corresponding transient kinetics of singlet ($S_1$) decay and triplet ($T_1$) rise shown below (down panel). **c, d** 2D nanosecond transient absorption color plots of IR806 (**c**) and IR806-BTTQD 2 (**d**) in deaerated chloroform using 700 nm pulsed laser excitation (2 µJ, FWHM = 50 fs), with corresponding transient kinetics at 580 nm in deaerated and aerated chloroform shown below (down panel). **e** Simplified Jablonski diagram showing the dominant photophysical routes and rates involved in IR806 (left) and IR806-BTTQD 2 (right) TTA system. **a**–**d** $c_{IR806} = 1 \times 10^{-5}$ M, $c_{BTTQD} = 1 \times 10^{-4}$ M.

deaerated samples, respectively. Different from the results of pure IR806, oxygen-induced reduction of triplet state lifetime is negligible when introducing BTTQD **2** annihilator. The TTET rate constants ($k_{TTET}$) of IR806-BTTQD 2 pair in the absence and presence of oxygen were determined to be $5.56 \times 10^5 \, s^{-1}$ and $5.15 \times 10^5 \, s^{-1}$, determining $\Phi_{TTET}$ (TTET quantum efficiency at the IR806 triplet) to be 60.8% for the aerated sample, and 71.8% for the deaerated sample, respectively (Fig. 3e). We also estimated the TTET rate and oxygen quenching rate through the classical steady/transient state measurements (see Supplementary Discussion 2.2 and Supplementary Figs. 25 and 26)[53], and the results showed a good correlation with the transient absorption results. Moreover, oxygen molecules were found to have very limited influence on the TTET rate, which was further verified by calculating the ratio of TTET rate in the presence of oxygen molecules and the one in the absence of oxygen molecules using the photophysical parameters of the paired molecules via the previously developed method (see Supplementary Discussion 2.3)[44]. This observation is in sharp contrast to the typically investigated standard PtOEP-DPA system, where oxygen molecules was observed to enhance TTET rate by orders of magnitude[54]. The distinct influence of oxygen molecules on the two TTA systems is reasonable, as DPA annihilator molecule containing polycondensated aromatic rings has a near-completely forbidden $T_1 \rightarrow S_0$ transition (decay rate, $200 \, s^{-1}$)[55], more prone to be influenced by the paramagnetic properties of oxygen molecules, while the BTTQD 2 annihilator in this work, containing a dimer structure with heteroatoms inserted into the framework of polycyclic aromatic hydrocarbons, is relaxed in spin forbiddance (decay rate, $23,400 \, s^{-1}$), therefore influenced less by the paramagnetic effect of oxygen molecules.

Triplet quenching by oxygen is the main reason to depopulate the triplets, resulting in quenching of TTA-UC[23]. In our TTA-UC system, the oxygen quenching efficiencies ($\eta_{O_2}$) for the sensitizer and annihilator triplets were evaluated to be 12.7% and <3.7%, respectively (see Supplementary Discussion 2.4), clearly showing that the involved triplets in both sensitizer and annihilator are insensitive to surrounding oxygen molecules.

An additional factor potentially contributing to TTA-UC quenching is the structural instability of TTA molecule pairs. In this context, the transfer of triplet energy to surrounding oxygen molecules might generate reactive oxygen species (ROS), such as singlet oxygen, capable of damaging the molecular structure of TTA pairs. This has been implied from distinct $\Phi_{UC}$ change trend in aerated and deaerated IR806-rubrene solutions (Fig. 2c). To investigate this possibility, we also examined the corresponding absorption spectra of TTA pairs shown in Fig. 2c. For the developed IR806-BTTQD 2 system, the absorption peaks of both IR806 and BTTQD 2 remained nearly constant in both aerated and deaerated solutions during long-term TTA-UC (Supplementary Fig. 27b, d). In contrast, an evident and rapid decrease in the rubrene absorption peak, contrasted with the unchanged absorption peak of IR806, was observed in aerated IR806-rubrene solution (Supplementary Fig. 27a). Furthermore, absorption peaks of both rubrene and IR806 remained nearly identical in deaerated solution (Supplementary Fig. 27c). These results reveal that ROS produced during triplet depletion can impair the structure of rubrene[23], but are unable to compromise the structures of synthesized BTTQD 2 emitter and IR806 sensitizer. This is reasonable, as rubrene has a low oxidation potential (~0.4 V), while both BTTQD 2 (~1.4 V) and IR806 (0.9 V) (Supplementary Fig. 28) have higher oxidation potential[56–58], thereby showing higher structural stability against produced ROS during TTA-UC.

Alongside that NIR excitation and red upconversion photons can deeply penetrate biological tissues, high stability and considerable upconversion emission efficiency render IR806-BTTQDs TTA-UC suitable for biological applications. To this end, we further encapsulated IR806-BTTQD 2 pair into an amphiphilic polymer F127-NH$_2$ to form a nanomicell via self-assembly process (Fig. 4a), in which lipophilic IR806 and BTTQD 2 molecules were adopted into in the hydrophobic core of nanomicell. Assembled hydrophilic segment of F127-NH$_2$ cloaks the hydrophobic core and stabilizes them in aqueous medium. By comparing the absorbance changes before and after the experiment, we evaluated the encapsulation efficiency of IR806 and BTTQD 2 to be 31% and 73%, respectively (Supplementary Fig. 29). The morphology of TTA-NMs was determined by transmission electron microscopy (TEM) to be quasi-spherical (Fig. 4b). The darker region on individual particles corresponds to the dense hydrophobic core, while the brighter surrounding areas correspond to the external hydrophilic chains. This is consistent with the reported typical microscopic morphology of nanomicelles[59–61]. Furthermore, the particle size distribution extracted from TEM images indicates that the core diameter of the nanomicelles is around ~18 nm (Fig. 4c), which correlates well with the DLS (Dynamic Light Scattering) results (~17 nm, Supplementary Fig. 30). Aqueous TTA nanomicells (TTA-NMs) solution was observed to emit intense upconversion emission around 680 nm under 808 nm NIR light excitation, resembling triplet fusion upconversion in organic phase (Fig. 4d). The maximum $\Phi_{UC}$ of TTA-NMs was determined to be about 0.44%, about threefold lower than that of the organic phase (Supplementary Fig. 31). As both excitation (808 nm) and emission (~680 nm) are within the first biological window (650–1000 nm), light scattering and absorption would be minimized, thus allowing high contrast deep-tissue imaging and sensing[62]. A lateral resolution of 1 mm was demonstrated for imaging through biological tissue of thickness of 5 mm, showing their promise for deep-tissue biophotonics (Supplementary Fig. 32). Moreover, remarkable upconversion photostability was demonstrated in PBS buffer under intermittent irradiation of 808 nm laser for 7 days (100 mW/cm$^2$, fluorescence spectra were measured after continuous light exposure for 2 h each day). The exceptionally high chemical and photostability of TTA-NMs in PBS buffer favors their long-term biological experiments (Fig. 4e).

We showed that these TTA-NMs could be internalized into HeLa cells, exhibiting enhanced cytoplasmic TTA-UC along an increase of incubation time (0–3 h) (Fig. 4f). Cell viability assay results of methyl thiazolyl tetrazolium (MTT) proved that these TTA-NMs have low toxicities at all investigated nanomicell concentrations (0–2 mg/mL), in which the survival rates of HeLa cell were higher than 90% in all cases after 24 h of incubation (Supplementary Fig. 33). Moreover, at 3 day post intravenous administration of TTA-NMs (injection of 100 μL with 10 mg/mL), we sacrificed Kuming mouse and implemented hematoxylin and eosin (H&E) staining of internal organs (liver, heart, kidney, lung, and spleen); no noticeable pathological changes were observed (Supplementary Fig. 34). A blood biochemical analysis was also performed to evaluate a whole set of biomarkers, including white blood cells, lymphocyte, monocyte, granulocyte, hemoglobin, red blood cells, hematocrit, mean corpuscular volume, mean corpuscular hemoglobin, mean corpuscular hemoglobin concentration, standard deviation of red blood cell distribution width, red blood cell volume distribution width, platelets, procalcitonin, platelet crit, mean platelet volume, platelet distribution width and platelet-large cell ratio (Supplementary Table 5). All measured data are commensurate with non-treated control group, falling within normal reference range. Taken together, these results confirmed the low toxicity and high biocompatibility of TTA-NMs, making them suitable as molecular probes for in vivo applications.

To showcase an in vivo application, we further utilized TTA-NMs to construct a turn-on upconversion biosensor for sensitive detection of reactive nitrogen species (RNS) in live mouse. Physiological levels of RNS, including peroxynitrite (ONOO$^-$) and nitrogen dioxide (NO$_2$), are necessary to maintain the normal function of living organisms, while excessive levels can adversely damage key biomolecular systems including lipids, proteins, and nucleic acids, leading to cellular dysfunctions. In essence, in vivo RNS levels have been implicated with the

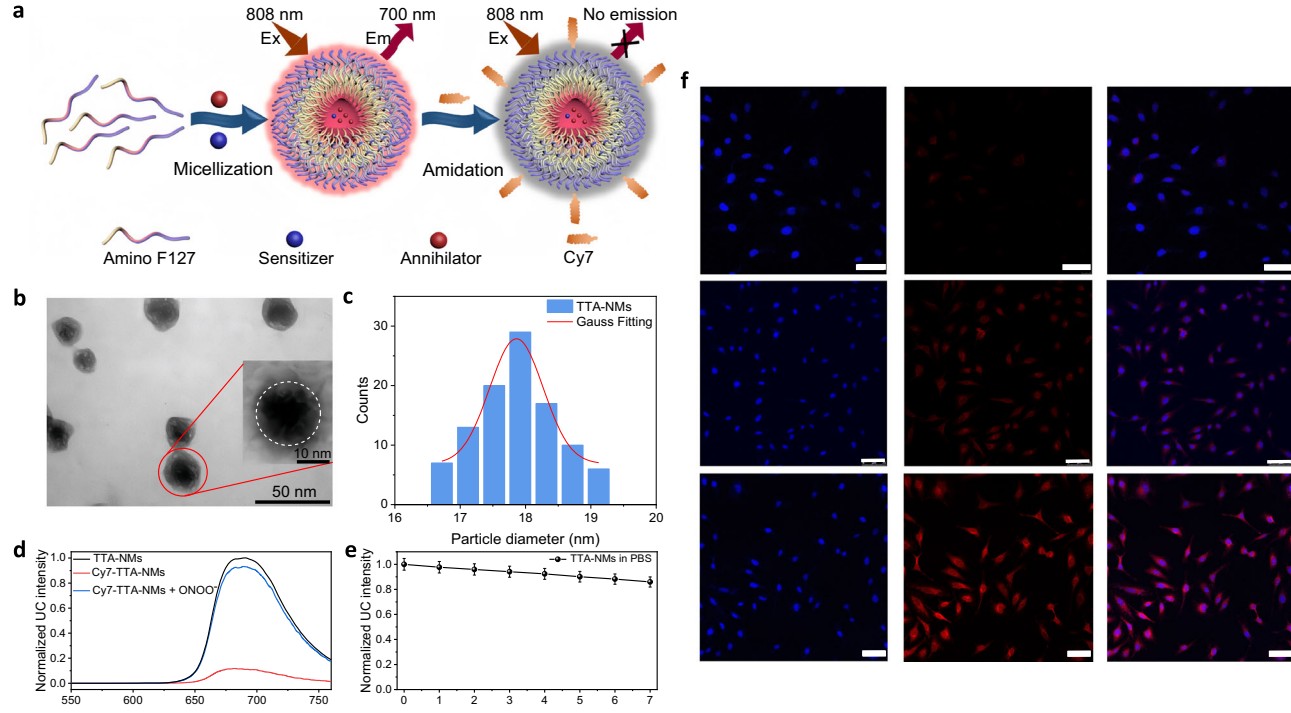

**Fig. 4 | Design and characterizations of ONOO⁻-responsive TTA nanomicelles (TTA-NMs). a** Schematic illustration of building ONOO⁻-responsive TTA-NMs with quenched upconversion via surface-attached Cy7 acceptor. **b** Transmission electron microscopy (TEM) of TTA-NMs in PBS buffer, scale bar is 100 nm, all experiments were independently repeated three times, yielding similar results. **c** The corresponding histograms of nanomicelle sizes obtained from analyzing TEM images (-100 particles). **d** Upconversion emission spectra of the TTA-NMs (5 mg/mL), and TTA-NMs treated with Cy7 and Cy7/ONOO⁻ (30 μM) in PBS buffer ($\lambda_{ex}$ = 808 nm, 10 W/cm²). **e** Upconversion emission stability of TTA-NMs in PBS buffer upon intermittent irradiation of 808-nm laser (spectra were measured after continuous laser exposure of 100 mW/cm² for two hours each day). Luminescence intensities are presented as mean ± standard deviations (SD) (n = 3). **f** Confocal laser scanning microscopy (CLSM) images of HeLa cells incubated with TTA-NMs for 0.5, 1, and 1.5 h at 37 °C, scale bar is 75 μm; three independent experiments were repeated, yielding similar results.

development and progress of several diseases, such as neurodegenerative diseases, cancer, as well as acute and chronic inflammatory diseases[63,64]. Note that TTA-UC materials can achieve optical upconversion at excitation power densities acceptable to organisms (all in vivo experiments use a uniform power density of 100 mW/cm² in this work). Moreover, the anti-Stokes emission enables imaging and detection virtually with zero-background, since autofluorescence, if any, will have longer wavelength than the excitation while TTA-UC have shorter wavelength than the excitation.[24] These advantages are not available in organic small molecules and inorganic semiconductor nanomaterials (QDs or CDs)[63,65,66]. Therefore, TTA-based nanomaterials would be more suitable for imaging and sensing applications in vivo. Especially, considering that conventional TTA-UC systems would be significantly quenched after exposure to oxygen, that are enriched in blood, rendering them challenge for in vivo applications[23], while the insensitivity to oxygen molecules single out our TTA-UC system for in vivo applications, such as in vivo biosensing ONOO⁻ here.

We grafted RNS-responsive cyanine dyes (Cy7) with carboxylic moiety onto the surface of TTA-NMs through covalent amide bonding with the amino group of F127-NH₂. Surface-attached Cy7 dyes have three important roles: (i) Elicitation of Förster resonance energy transfer (FRET) from BTTQD 2 to Cy7 due to spectral overlapping between Cy7 absorption and BTTQD 2 emission, resulting in upconversion quenching. (ii) Competition with IR806 for absorbing TTA excitation photons, further enhancing TTA-UC quenching. It is the two reasons account for the -12-fold quenching of TTA-UC after Cy7 loading (Fig. 4d). The FRET process is determined to be responsible for -35.5% of the quenching, while the competitive absorption contributes to about 64.5% of the observed quenching effect (Supplementary Fig. 35a–c). (iii) Surface Cy7 could specifically react with ONOO⁻, cleaving the polymethyl chains in its molecular structure through one-

electron oxidation reactions (Supplementary Fig. 35d)[63,64], thereby turning on the quenched TTA-UC for biosensing. Note that IR806 are stable toward ONOO⁻ according to selective experiments (Supplementary Fig. 36). Thus, with the introduction of ONOO⁻, Cy7 on the surface of TTA-NMs will be cleaved, while the internal sensitizer IR806 will remain intact, that is, upconversion emission will be restored. According to the standard calibration curve for Cy7 based on its absorbance at 780 nm, the amount of Cy7 conjugated to the TTA-NMs was determined as 42.6 wt% (Supplementary Fig. 37). Considering that 31% of IR806 was encapsulated into the hydrophobic core of TTA-NMs, the ratio of surface Cy7 molecules to internal IR806 molecules was estimated to be -52: 1. Next, we evaluated the long-term stability of Cy7, TTA-NMs and Cy7-coated TTA-NMs in simulated body fluids (SBF, pH = 7.4) and blood (fetal bovine serum, 10%). The results showed that the particle size and absorption intensity of TTA-NMs and Cy7-coated TTA-NMs did not change in the simulated in vivo environment, indicating the less likelihood of agglomeration and component loss of these nanoparticles in vivo (Supplementary Figs. 38 and 39).

Treatment of 20 μM ONOO⁻ resulted in a near-full recovery of upconversion emission and an obvious concomitant decrease of Cy7 absorbance, caused by the induced irreversible oxidative degradation of Cy7 molecular structure. A plot of upconversion emission intensity against ONOO⁻ concentration (0–30 μM) was measured to be linear (Supplementary Fig. 40), demonstrating their suitability as ONOO⁻ biosensors. Furthermore, measurement of the plot slope (K) and the standard deviations (δ) of five replicate measurements of blank TTA-NMs solutions (Supplementary Table 6) determined the limited of detection (LOD = 3δ/K) to be 9 nM, about 2–50 times lower than literature results on ONOO⁻ sensing (Supplementary Table 7)[66–68]. To evaluate the specificity of Cy7 toward ONOO⁻ under physiological conditions, a set of radicals, such as superoxide anion radical (•O₂⁻),

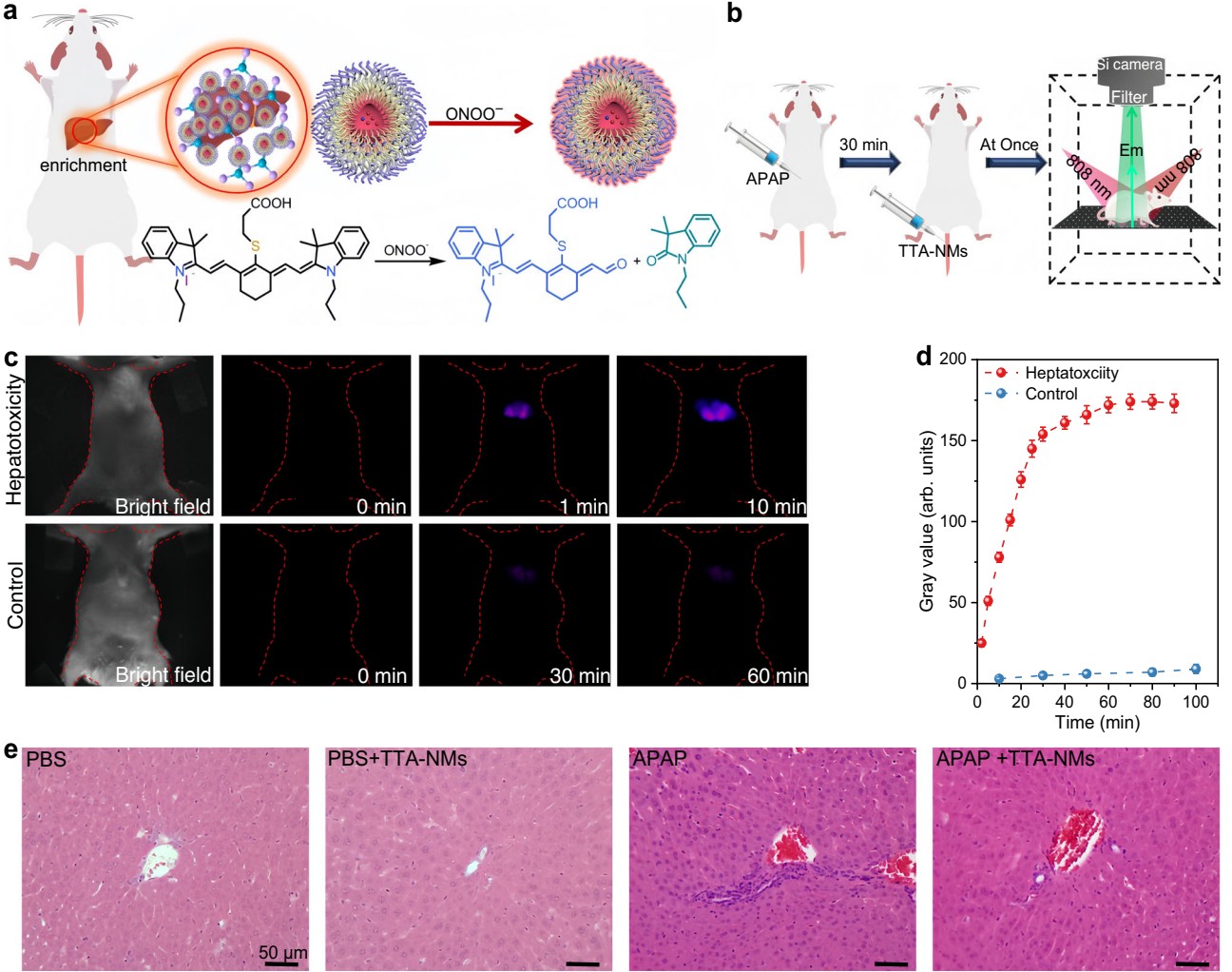

**Fig. 5 | In vivo upconversion fluorescence imaging of ONOO⁻ using the TTA nanomicelles (TTA-NMs). a** A schematic representation of the Cy7-coated TTA-NMs for turn-on nitrosative hepatotoxicity monitoring in vivo. Parts of the figure was drawn using elements from Servier Medical Art under a Creative Commons license CC BY 3.0. **b** Schematic illustration of the in vivo imaging protocols for mice hepatotoxicity. Parts of the figure was drawn using elements from Servier Medical Art under a Creative Commons license CC BY 3.0. **c** Representative upconversion imaging of TTA-NMs in mouse pre-treated with APAP (top) and PBS (bottom). A 750 nm short-pass optical filter was used to isolate 808 nm excitation photons. **d** Temporal changes of upconversion emission signals in liver in normal and hepatotoxic mice. Upconversion emission intensities are presented as mean ± standard deviations (SD) (n = 3). **e** H&E-stained histological sections of the liver organ in different treatment groups, scale bar: 50 μm; three independent experiments were repeated, yielding similar results.

hydroxy radical (•OH), and chlorine oxide radical (ClO⁻) were tested, demonstrating high specific detection of ONOO⁻ using Cy7-coated TTA-NMs (Supplementary Fig. 41). It has been shown that excessive intake of acetaminophen (APAP) can result in liver injury and over-produce ONOO⁻ [63,64]. We therefore utilized APAP-induced hepatotoxicity model to validate the use of developed Cy7-coated TTA-NMs biosensors for selective ONOO⁻ detection in the internal liver of living mouse (Fig. 5a, b).

After intravenous injection into mice, nanomedicines commonly tend to accumulate in the liver. This inclination is attributed to the liver's role as a biological filtration system, which sequesters a substantial percentage (30–99%) of administered nanoparticles from the bloodstream [64,69]. This feature also enables passive accumulation of the developed Cy7-coated TTA-NMs nanoprobes (~20 nm) into the liver organ. We first administrated intraperitoneal injection of APAP (0.05 mM, 100 μL) to an anesthetic Kunming mouse to induce hepatotoxicity, while an equal volume of PBS solution to the other mice to create a control. Detection of hepatitis-related biomarkers (AST and ALT) confirmed that APAP treatment induced a hepatotoxicity (Supplementary Fig. 42)[70], while the employment of aminophenyl

fluorescein (APF), a standard probe specific to ONOO⁻ [71], further confirmed that APAP-induced hepatotoxicity is due to the produced ONOO⁻ (Supplementary Fig. 43). After a 30-minute interval, a PBS solution containing Cy7-loaded TTA-NMs (5 mg/mL, 100 μL) was intravenously administered to mice that had been treated with APAP (n = 3), forming the experimental group. In parallel, the same injection was administered to mice without APAP treatment, establishing the control group (n = 3). Subsequently, dynamic upconversion emission imaging was conducted (Fig. 5b), and Fig. 5c displays representative imaging results from one mouse in each group. Imaging of APAP-treated mouse showed first a remarkable enhancement of TTA-UC in the liver region, reaching a plateau around 30 mins, while a constant low level of luminescence was detected in the control mouse (Fig. 5d). After imaging, two mice were sacrificed to harvest organs for ex vivo imaging (Supplementary Fig. 44). The strongest luminescence signal was found in the liver, consistent with in vivo imaging results. Moreover, liver injury was only observed in mice treated by APAP and no hepatic inflammation was observed in mice treated with PBS or PBS + TTA-NMs (Fig. 5e), demonstrating the correctness of the used mouse model for in vivo TTA-NMs biosensing.

In summary, we have developed an oxygen-immune class of NIR-excited TTA upconversion ($\lambda_{ex}$ = 808 nm, $\lambda_{em}$ = 680 nm), which avoids the long-standing common problem of oxygen quenching of molecular triplets in absolutely most, if not all, TTA-UC. This photochemical TTA upconvesion utilized IR806 molecules as sensitizers and chemically synthesized BTTQD dyes with designated molecular dimer structure as annihilators, producing stable upconversion under long-term continuous-wave laser irradiance (>480 mins) and storage (>7 days) in aerated solution. The low oxygen triplet quenching efficiency ($\eta_{O_2}$ = 12.7% and 3.7% for IR806 and BTTQD 2, respectively), and the high oxidation potentials of both sensitizer and annihilator molecules against the damage of ROS, generated during the triplet depletion, confer the oxygen immunity to the IR806-BTTQD 2 biocomponent TTA system. We further assembled IR806-BTTQD 2 and surface-modified amphiphilic triblock polymer of F127-NH$_2$ into low toxic and biocompatible TTA-NMs nanosensors, which demonstrated sensitive and specific detection of ONOO$^-$ via turning on upconversion, and therefore enabled long-term dynamic observation of drug-induced nitrosative hepatotoxicity in liver of living Kunming mouse. This exceptional air stability of TTA-UC untaps their wide range of technological applications, ranging from solar cells to photocatalysis and from nanothermometry to in vivo imaging, in real-world environments.

## Methods

### Synthesis
The synthesis processes and chemical characterizations of all organic molecules and TTA-NMs are described in detail in the supplementary information.

### Chemicals
All chemicals were used as received. Benzothiadiazolyl diamine compound 1 (97%), ninhydrin compound 2 (98%), and triphenyldiamine compound 4 (97%) were obtained from Xingwei Lithium Energy Technology Co., LTD. Intermediate compounds 3 and 5, as well as NIR dyes IR806 and Cy7, were synthesized and purified with details provided in the Supplementary Information. All remaining chemicals, including tetrahydrofuran (THF, 99.9%), dichloromethane (DCM, 99.9%), chloroform (CHCl$_3$, 99.9%) and methanol (CH$_3$OH, 99.9%), were purchased from Sigma Aldrich.

### Upconversion luminescence imaging
All animal experiments are conducted in accordance with the Institutional Animal Care and approved by the Ethics Committee of Harbin Institute of Technology. The 4-week-old female Kunming mice were purchased from the Second Affiliated Hospital of Harbin Medical University. The mice were then housed in regular rectangular cages and maintained under standard conditions in conventional housing facilities (light/dark cycle: 16/8 h; Temp: 25 °C; relative humidity: 40%). In vivo upconversion luminescence imaging was performed with a custom-made small animal bioimaging system equipped with a Si-camera (Retigal LUMO CCD camera, QImaging Corporation, Canada) and a fiber-coupled 808 continuous-wave lasers that output a homogenous laser irradiance of 100 mW/cm$^2$ with an imaging area of 25 × 25 cm$^2$. A band-pass filter of 808 nm and a short-pass filter of 750 nm was placed in front of the imaging.

### Cytophagocytosis experiment
HeLa cells were obtained from the Cell Bank of the Chinese Academy of Sciences which cultured in DMEM (HyClone; Cytiva) with 10% FBS (HyClone; Cytiva), 100 μg/ml penicillin and 100 μg/ml streptomycin (HyClone; Cytiva) at 37 °C in a humidified atmosphere with 5% CO$_2$. The uptake of TTA-NMS by HeLa cells was investigated using confocal laser scanning microscopy (CLSM). First, HeLa cells were inoculated and cultured overnight to obtain monolayer cells in a 6-well culture plate. Next, 1 mL TTA-NMs (2 mg/mL) was added to the wells and incubated for 0.5, 1, and 1.5 h, respectively. After these cells were stained by 4',6-diamidino-2-phenylindole (DAPI, 0.02 mg/mL) for 20 min, fixed and rinsed by glutaraldehyde (2.5%) and PBS. Corresponding CLSM imaging was carried out under 488-nm excitation.

### Cytotoxicity
For the cytotoxicity study of TTA-NMs, MTT (3-[4,5-dimethylthiazol-2-yl]-2,5 diphenyl tetrazolium bromide) cell assay was used. First, about 1 × 10$^4$ HeLa cells per well were inoculated into 96 well plates, allowing the cells adhere to the wall after cultivation overnight. Next, TTA-NMs nanoprobes with surface-attached Cy7 in PBS were added to DMEM cell culture, resulting in a variety of conceninttrations of 0, 0.13, 0.25, 0.5, 1, and 2 mg/mL. The HeLa cells and TTA-NMs nanoprobes were allowed to incubate for 24 h. Then, MTT (methyl thiazolyl tetrazolium, 5 mg/mL) culture medium was added to each well, and allowed to incubate for an additional 4 h. Subsequently, culture medium was removed and DMOS of 150 μL were added to the well. After placing in the dark for 10 mins, the absorbance of 490 nm was monitored by a microplate reader, enabling to calculate the cell survival rate.

### X-ray crystallography of BTTQD dyes
Crystal data of BTTQD 1 was collected on a Brooker D8 Quest Ultra diffractometer (Germany) using graphite-monochromated Mo Kα radiation ($\lambda$ = 0.71073 Å) at room temperature. The sample was a 0.1 mm crystal without cracks. After collecting diffraction patterns, data were reduced to obtain the coarse structure of the target product. Structures of BTTQD 1 were solved by using Patterson methods (SHELXS-97), expanded using Fourier methods, and refined using SHELXL-97 (full-matrix least-squares on F2) and WinGX v1.70.01 programs packages. All non-hydrogen atoms were refined anisotropically. Empirical absorption corrections based on equivalent reflections were applied. Crystallographic data for BTTQD 1 were deposited in the Cambridge Crystallographic Data Center with CCDC nos. 2099068.

### Electrochemical experiment
In the cyclic voltammetry experiment, scan was performed at a certain speed from the starting potential to the ending potential. In this work, all scanning speeds were set as 50 MV s$^{-1}$, and reverse scans were set at the same rate, yielding the current potential curve. Test procedures of cyclic voltammetry in this work are as follows: preparing 10$^{-3}$ M anhydrous dichloromethane solution of the target product (5 ml); adding the supporting electrolyte Bu$_4$NPF$_6$, and keeping the concentration of the electrolyte at 0.1 M. Before the test, the solution was deoxidized with argon. The test used a three-electrode system, in which platinum wire electrode was used as the counter electrode, Ag/AgCl electrode as the reference electrode, and ferrocene was used to calibrate the zero point.

### Computational methods
To have a deeper understanding of the frontier orbital energy levels, density functional theory calculations (DFT) were carried out for all target products using Gaussian 09 software and B3LYP 6-311 + + G** algorithm. The file containing optimized atomic coordinates through DFT is provided as Supplementary Data 1.

### Licenses
Partial drawing elements of Fig. 5a, b and Supplementary Fig. 34a were sourced from the Servier Medical Art database under a Creative Commons license CC BY 3.0.

### Reporting summary
Further information on research design is available in the Nature Portfolio Reporting Summary linked to this article.

## Data availability

The authors declare that all the data supporting the findings of this study are available from the authors on request. Source Data has been deposited in figshare under accession code (https://doi.org/10.6084/m9.figshare.22976144)[72]. Source data are provided with this paper.

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

## Acknowledgements

This work is supported by the National Natural Science Foundation of China (52272270 and 51972084 to G.C.), Natural Science Foundation of Heilongjiang Youth Fund (YQ2021B002 to D.X.), the National Key Research and Development Program of China (No. 2021YFE0105800 to D.X.), Heilongjiang Postdoctoral Scientific Research Developmental Fund (LBH-Q20018 to D.X.), and the Fundamental Research Funds for the Central Universities, China (AUGA5710052614 to G.C.).

## Author contributions

G.C. conceived the idea; X.W. conducted most experiments; G.C. and X.W. performed data analysis and wrote the manuscript; F.D. and D.X. are responsible for organic synthesis. T.J. and F.L. performed in vitro and in vivo experiments; X.D., R.D., K.L. and Y.Y. provided supports on experiments; W.W. measured femtosecond transient absorption spectra. D.X. and G.C. supervised the project.

## Competing interests

The authors declare no competing interests.
