## [Peer Review File · Nature Communications]

Molecular Near-Infrared Triplet-Triplet Annihilation Upconversion with Eigen Oxygen ImmunityEditorial Note: Parts of this Peer Review File have been redacted as indicated to remove third-party material where no permission to publish could be obtained.

REVIEWER COMMENTS

Reviewer #1 (Remarks to the Author):

Wang and co-authors present a new bicomponent TTA upconversion system that works in the NIR spectral region with an excellent resistance to the presence of oxygen. The quenching of triplet energy by molecular oxygen is indeed one of the most challenging points to face, in order to fully exploit the TTA upconversion process in real life applications.

The system proposed shows indeed a very oxygen resistance with respect to other standard model upconverters, and it has been successfully embedded in biocompatible nanomicelles and exploited for *in vitro* imaging and sensing experiments. The obtained results are interesting, thus supporting the further development of these upconverting materials.

However, there is a lack of information and experiments that makes the model uncompleted and unclear. The photophysical model proposed by the author to explain this extraordinary oxygen resistance and the global UC process observed. These are the points that should be addressed as mandatory before re-considering and re-evaluating the paper for publication.

General comments:

There is no experimental investigation of the energy transfer process between the proposed sensitizer and the emitters. How much is efficient? The classical steady state measurement does not correlate well with the ET rate calculated from transient absorption measurements? Does the oxygen presence affect the ET rate as previously demonstrated (PHYSICAL REVIEW B 82, 125113, 2010,)?

What is the effect of molecular oxygen on single molecules of sensitizer and emitters? These data should be reported at least in the supporting information file. What are the phosphorescence/fluorescence yields in presence/absence of oxygen?

Specific requests to be addressed:

- Page 4 line 89:

“Single crystal x-ray structure studies revealed π - π and S-N locking 90 interactions between dimers, favoring intermolecular triplet fusion (Figure S7)”

Please explain this sentence, why π - π interactions should help the intermolecular triplet fusion? In general, the interaction can change the triplet electronic energies and recombination rates, with also detrimental interactions.

- Page 6 line 144:

The upconversion QY values are reported as normalized to 100%. As recently discussed, this should be explicitly declared to allow reliable comparison with literature (Chemical Physics Reviews 2022, 3 (4),

041301 - ACS Energy Lett. 5 (7), 2322–2326 (2020).) at which excitation intensity these values have been recorded?

Why so low value is observed? It could be due to a back energy transfer from upconverted singlets of the emitters towards the sensitizers? As showed in Fig. 1c, there is a not negligible overlap between the emitter photoluminescence and the sensitizers absorption. This back transfer should be investigated by time resolved photoluminescence spectroscopy. This could be the reason why at emitter concentration higher than 80 – 100 μM the process yield starts to decrease (Fig. S14).

- Page 9 line 208

The authors calculate the sensitizer to emitter ET rate by looking at the change in time of the sensitizer transient singlet absorption band lifetime at 500, which accelerates in presence of the emitter. But I guess they should look at the triplet state transient absorption, which lifetime should be significantly reduced in the case of efficient ET. IN any case the ET rate estimate here, in terahertz regime, does not correlate at all with the rise time of the upconverted emission reported in the Fig. S13 (megahertz scale) which should mirror the ET process. Some comparison with complementary ET rate measurements are need to clarify this point and the flowing discussion on the ET step in the system.

- Page 9 line 220

“The observed 3-fold higher intersystem crossing rate of IR806-BTTQD 2 ($k_{\text{ISC}}=6.1 \times 10^9 \text{ s}^{-1}$ 220) than that of IR806 ($k_{\text{ISC}}=1.7 \times 10^9 \text{ s}^{-1}$ 221) further confirmed the occurrence of TTET, which expedites the inter system crossing via depleting triplets of IR 806 via BTTQD 2”

These conclusions must be supported more strongly. Considering the ET is a diffusion limited process in solution, so I wonder how it can be so fast to be competitive with the ISC process at the dye concentrations employed. Moreover, the ISC occur prior to ET, therefore all molecules during excitation does not have electron in the triplet state. So, the ET should not affect at all the ISC. The observed variation should be ascribed to other mechanism due to the presence of the UC emitter.

- Figure S13.

The author present the data obtained by time resolved spectroscopy of the upconverted emission. The provided comment are not clear at all. First, the rise time mirrors no only the TTA rate that created upconverted singlets, but mainly the ET rate that produced the triplet population. This rise time should therefore be discussed referring to the ET rate.

Second, the upconversion emission intensity does not reach plateau, so the steady state equilibrium condition is not reached. Which laser excitation is used here? What is the pulse width and the energy per pulse?

Third, the upconversion emission decays quickly, with no very long time signal in the ms time scale. The intensity decay appears to be a single exponential function. Is that realistic? The excitation intensity is so low to have the limit case in which TTA is negligible and therefore we are looking at the emitters triplet lifetime? Again excitation conditions are mandatory to point out what happen at get much details and information on the proposed system (The Journal of Chemical Physics 153 (11), 114302)

FIGURES:

Figure captions and panel should be carefully revised for example to avoid that labels and legends overlap the data plots and to give the required detailed information of the reported data crucial for the

reader.

Reviewer #2 (Remarks to the Author):

In this article, the authors presented a class of near-infrared (NIR) TTA upconversion with eigen oxygen immunity consisting of IR806 molecules as sensitizers and three chemically synthesized BTTQD dyes with defined dimer structure as annihilators. The authors claimed that such TTA system can be used for sensitive and specific detection of ONOO⁻ and related hepatotoxicity in vivo. Basically, the concept of TTA is not new, the authors utilized IR806 molecules as sensitizers and chemically synthesized BTTQD dyes as annihilators, which put in some new components in TTA. However, such technical input, although interesting, does not greatly change TTA performance as compared to other counterparts in conventional TTA. Importantly, the authors did not present the rationale and necessity why TTA here can be used for free radical sensing, especially many well-functioned probe systems like small molecule, polymers and nanoparticles etc have been established and performed very well for radical species imaging in vitro and in vivo. Along the whole system, many important information is missing in the main context. Together with the weak points below, the novelty and current scope studies do not support the manuscript published in Nature Communications.

Main concerns:

1. Figure 2c, Figure 4e and others, to check the oxygen resistance and stability in aqueous solutions, the laser power density of 100 W/cm² was applied, will the molecules survive under such large power irradiation? And such high dosage may also limit the applications in living systems.
2. Similarly, figure 2 c, d, under the degassed conditions, the some certain photostability ~ 2hr was reported here, which is not long in terms for bioimaging and other studies. What about prolonged the stability? And also, the polymer encapsulated TTA- NMs in Figure 4e, the stability is not so good.
3. Looks like the QY values of TTA-UC IR806/BTTQD 1-3 are not high and efficient?
4. And in Figure 4, TTA was encapsulated within polymer nanoparticles, some very important data like entrapment efficiency of each TTA component inside the particles was missing.
5. In order to sustain "on-off" effect, sounds like extra Cy7 was amidated onto TTA-NMs. The rationale why use Cy7 (also IR806 already inside polymer structure) and detailed characterization of Cy7 coating (e.g. coating efficiency etc..) all are missing.
6. What about stability of Cy7, Cy7 coated TTA-NMs and TTA-NMs like in blood, body? The fluorescent detection or imaging of peroxynitrite ONOO⁻ by small molecule probes, semi-conducting polymers, and nanoparticles etc have been well established. For what reason, the authors claimed " Yet, imaging ONOO⁻ levels in vivo remain a challenge....." At Page 13, line 299? This also raised the concern regarding the necessity and rationale to use TTA-NMs for ONOO⁻ imaging? Any obvious merits? The authors did not clearly show all these important information.
7. And the detection selectivity in Figure S24 is not convincing. And in Figure S25, the referee is quite skeptical of the free radical production done by APAP treatment. Lack of standard probes to clearly prove the suitability.
8. Unless the real detection limit tested, the authors can not claimed "..... which demonstrated sensitive

and specific detection of ONOO⁻ via turning on upconversion,.....” as in page 15, line 365

Others:

1. The authors described the synthesis of the molecules. Yet, the authors only scanned some NMR spectra, but no detailed data assignment.
2. Fig S20, data not convincing and the trend of cell viability quite inconsistent.
3. Many figures in supporting information lack of experimental error bar.

Reviewer #3 (Remarks to the Author):

This manuscript reports a molecular system for triplet-triplet annihilation aided upconversion (TTA UC). The TTA UC of the system is not quenched by oxygen, which is unique. The results are interesting, I recommend acceptance of the manuscript after revisions.

Line 220-222, with intermolecular TTET, the ISC quantum yield of IR860 could not be higher than the native IR806, except there is strong interaction between IR860 and the energy acceptor, such as aggregation. The description should be clarified;

In solution, the intermolecular TTET should be on time scale of microseconds, because the process is diffusion controlled, it can not be much faster. But the authors observed fast TTET, this is unusual, more discussion should be made on this issue.

It seems the authors did not give clear, convincing explanation about the 'eigen oxygen immunity' of their system. According to the general photochemistry in solution, the conclusion is unlikely, e.g. the poorly quenched triplet state in aerated solution, fast intermolecular TTET in solution, etc.

Response to the Comments

Reviewer 1:

Wang and co-authors present a new bicomponent TTA upconversion system that works in the NIR spectral region with an excellent resistance to the presence of oxygen. The quenching of triplet energy by molecular oxygen is indeed one of the most challenging points to face, in order to fully exploit the TTA upconversion process in real life applications.

The system proposed shows indeed a very oxygen resistance with respect to other standard model upconverters, and it has been successfully embedded in biocompatible nanomicelles and exploited for in vitro imaging and sensing experiments. The obtained results are interesting, thus supporting the further development of these upconverting materials.

However, there is a lack of information and experiments that makes uncompleted and unclear the photophysical model proposed by the author to explain this extraordinary oxygen resistance and the global UC process observed. These are the points that should be addressed as mandatory before re-considering and re-evaluating the paper for publication.

Response: We would like to take this opportunity to thank the reviewer for a series of valuable comments on our work. In the revised manuscript, we performed a range of additional experiments to clarify the photophysical model underlying the observed TTA upconversion with oxygen-insensitive nature. We hope that the revised manuscript can now be accepted for publication. We would also be happy to revise more based on further specific comments.

General comments:

(1) There is no experimental investigation of the energy transfer process between the

propose sensitizer and the emitters. How much is efficient? The classical steady state measurement does correlate well with the ET rate calculated from transient absorption measurements? Does the oxygen presence affect the ET rate as previously demonstrated (PHYSICAL REVIEW B 82, 125113, 2010,)?

Response: The triplet-to-triplet energy transfer (TTET) efficiency (Φ_{TTET}) was evaluated from the following equation:¹

$$\Phi_{TTET} = 1 - \frac{\tau_{T1,D-A}}{\tau_{T1,D}} \quad (\text{Eq. R1.1})$$

$$TTET \text{ Rate} = 1/\tau_{T1,D-A} - 1/\tau_{T1,D} \quad (\text{Eq. R.1.2})$$

where $\tau_{T1,D-A}$ is the measured lifetime of donor (IR806) triplet in the presence of the acceptor (BTTQD2), and $\tau_{T1,D}$ is the measured lifetime of donor triplet in the absence of BTTQD2. The kinetic curves of transient absorption (TA) spectra at 570 nm indicate $\tau_{T1,D} = 4.57$ and $3.01 \mu\text{s}$ for deaerated and aerated samples, and $\tau_{T1,D-A} = 1.29$ and $1.18 \mu\text{s}$ for deaerated and aerated samples, respectively (Figure 3c). As a result, the Φ_{TTET} was determined to be 60.8% for the aerated sample, and 71.8% for the deaerated sample, respectively. Correspondingly, the TTET rate was calculated to be $5.56 * 10^5 \text{ s}^{-1}$ and $5.15 * 10^5 \text{ s}^{-1}$ for the deaerated and aerated samples, respectively.

Following the suggestion of the reviewer, we also evaluated the TTET rate through the classical steady/transient-state measurements. Specifically, quenching IR806 triplet phosphorescence (peaked at 970 nm in Figure R1.1a) via TTET to BTTQD2 can be described by the Stern–Volmer (SV) equations:²

$$\frac{I_0}{I} = k_{SV}[M] + 1 = k_{TTET} \times \tau_0[M] + 1 \quad (\text{Eq R1.3})$$

$$\frac{1}{\tau} = k_{TTET}[M] + \frac{1}{\tau_0} \quad (\text{Eq R1.4})$$

Where I_0/τ_0 and I/τ are the phosphorescence intensities/lifetimes of IR806 in the absence and presence of BTTQD emitter, respectively; $[M]$ is the molar concentration of BTTQD2, k_{SV} and k_{TTET} are the Stern-Volmer constant and TTET kinetic rate constant, respectively. Figure R1.1 and R1.2 plots IR806 phosphorescence intensity and lifetime against BTTQD 2 concentration in the absence and presence of oxygen molecules, respectively. Observation of linear plotting in Figure R1.1 and R1.2 gives

clear evidence that TTET dynamically quenches IR806 triplets in both aerated and deaerated samples. Using phosphorescence intensity changes in Figure R1b and c, k_{SY} in deaerated and aerated conditions was calculated to be 25900 M^{-1} and 15700 M^{-1} , thereby evaluating k_{TTET} to be $5.66 * 10^9 \text{ M}^{-1} \text{ s}^{-1}$ and $5.22 * 10^9 \text{ M}^{-1} \text{ s}^{-1}$ in absence and presence of oxygen, respectively. Likewise, based on the changes in the triplet lifetimes of IR806, k_{TTET} was calculated, according to Eq R1.4, to be $5.74 * 10^9 \text{ M}^{-1} \text{ s}^{-1}$ and $5.21 * 10^9 \text{ M}^{-1} \text{ s}^{-1}$ in absence and presence of oxygen (Figure R1.2), respectively, well consistent with the ones achieved in Figure R1.1 (<1.4% deviation). Therefore, k_{TTET} rate constant obtained from the lifetime measurement (Figure R1.2) was adopted to calculate TTET rate at $[M]=1.0 * 10^{-4} \text{ M}$, the one utilized for TA measurements, giving $5.74 * 10^5 \text{ s}^{-1}$ and $5.21 * 10^5 \text{ s}^{-1}$ in the absence and presence of oxygen, respectively. Both values agree well with the ones of $5.56 * 10^5 \text{ s}^{-1}$ and $5.15 * 10^5 \text{ s}^{-1}$ for the deaerated and aerated samples, obtained from TA measurements.

Action: We have included both Figure R1.1 and Figure R1.2 as Figure S22 and Figure S23 in the revised supporting information, and added corresponding discussions in the revised manuscript on page 11 and paragraph 1, as follows:

“The TTET rate constants (k_{TTET}) of IR806-BTTQD 2 pair in absence and presence of oxygen were determined to be $5.56 \times 10^5 \text{ s}^{-1}$ and $5.15 \times 10^5 \text{ s}^{-1}$, determining Φ_{TTET} (TTET quantum efficiency at the IR 806 triplet) to be 60.8% for the aerated sample, and 71.8% for the deaerated sample, respectively (Figure 3e). We also estimated the TTET rate and oxygen quenching rate through the classical steady/transient state measurements ($k_{TTET} = 5.59 * 10^5 \text{ s}^{-1}$ and $5.38 * 10^5 \text{ s}^{-1}$ in the absence and presence of oxygen, see Section 4.2, Figure S22 and S23 in supporting information for more details), and the results showed a good correlation with the transient absorption results.”

Figure R1.1. (a) Measured phosphorescence spectra of IR806 (chloroform, 1 μM). (b, c) Plotting the intensity of IR806 phosphorescence peaked at 970 nm against various concentrations of BTTQD 2 in the absence (b) and presence (c) of oxygen molecules. The solid red lines represent Stern-Volmer fitting. IR806 concentration was fixed at $1 \times 10^{-5} \text{ M}$, while BTTQD 2 concentration was varied from 0.1 to $1.0 \times 10^{-4} \text{ M}$. Phosphorescence spectra were measured at 77K under continuous-wave 808 nm laser excitation.

Figure R1.2. Plotting IR806 phosphorescence lifetimes at 970 nm against various concentrations of BTTQD 2 in the absence (a) and presence (b) of oxygen molecules. The solid lines represent Stern-Volmer fitting. IR806 concentration was fixed at $1 \times 10^{-5} \text{ M}$, while BTTQD 2 concentration was varied from 0.1 to $1.0 \times 10^{-4} \text{ M}$. The phosphorescence lifetimes were measured at 77K under continuous-wave 808 nm laser excitation.

experiments were performed at 77K under 808 nm laser excitation operating in pulsed mode.

Both steady-state measurement and TA measurement indicate that the TTET rate in our biocomponent TTA system was nearly unchanged (or slightly decreased) by the presence of oxygen molecules. The ratio of energy transfer rate under aerated (\bar{k}_{TTET}) and deaerated (k_{TTET}) conditions was calculated to be 0.926 for TA measurement. Monguzzi *et al.* previously established an approach to estimate $\frac{\bar{k}_{TTET}}{k_{TTET}}$ as function of the photophysical parameters of the employed molecules (*Physical Review B* 2010, 82 (12), 125113; *Physical Review B* 2008, 78 (19), 195112).³⁻⁴ We also adopt this approach here to evaluate $\frac{\bar{k}_{TTET}}{k_{TTET}}$. Specifically, when the excitation power density is below the TTA threshold, the main deactivation channel of the emitter triplet is self-quenching (or first order quenching), and $\frac{\bar{k}_{TTET}}{k_{TTET}}$ can be given by the following expression:

$$\frac{\bar{k}_{TTET}}{k_{TTET}} = \left[\frac{k_{rad}^D}{\bar{k}_{rad}^D} \left(\frac{\bar{k}_T^A}{k_T^A} \right)^2 \frac{k_{exp}^D}{\bar{k}_{exp}^D} \frac{I_A/I_D}{I_A/I_D} \right]^{\frac{1}{2}} \quad (\text{Eq. R1.5})$$

where k_{rad}^D is the radiative decay rate of the donor (IR806), k_{exp}^D represents the triplet decay rate of the donor measured experimentally (equal to the sum of k_T^D and k_{TTET}), k_T^A is the triplet decay rate of the acceptor (BTTQD 2), I_A and I_D are the TTA-UC fluorescence of the acceptor and the phosphorescence of the donor, respectively. The bar (—) overhead the parameter indicates the parameter in the presence of oxygen. $\frac{k_{rad}^D}{\bar{k}_{rad}^D}$ can be given by the ratio of the time-resolved photoluminescence intensities of the deaerated and aerated samples at the delay time $\tau = 0$. The lifetime of the acceptor triplet state (the inverse of k_T^A) can be estimated from the long component lifetime in the TTA-UC decay profile ($\tau_{T1}^A = 2\tau_{UC}$). Finally, I_A and I_D can be simply obtained from time-integrated photoluminescence intensities of steady state measurements. The corresponding parameters have been summarized in Table R1.

Table R1. The involved parameters obtained by classical steady/transient state measurements.

	k_{rad}^D	k_{exp}^D	k_T^A	$\frac{\bar{I}_A/\bar{I}_D}{I_A/I_D}$
deaerated	$2.06 * 10^5 \text{ s}^{-1}$	$7.90 * 10^5 \text{ s}^{-1}$	$2.34 * 10^4 \text{ s}^{-1}$	1.31
aerated	$3.12 * 10^5 \text{ s}^{-1}$	$8.33 * 10^5 \text{ s}^{-1}$	$2.43 * 10^4 \text{ s}^{-1}$	

According to Eq R1.5 and the parameters in Table R1, $\frac{\bar{k}_{TTET}}{k_{TTET}}$ was estimated to be 0.941, in good agreement with the one of 0.926 estimated from TA measurement. These results indicate oxygen does not enhance TTET rate in our biocomponent TTA-UC system, different from the reported observation of drastic enhancement of TTET rate in the presence of oxygen, by A. Monguzzi et al, in typically investigated biocomponent TTA system (Donor: PtOEP, Acceptor: DPA, *Physical Review B* 2010, 82 (12), 125113). One of the most likely reasons is that the triple state decay rate of the BTTQD emitter was almost not influenced by surrounding oxygen molecules, in sharp contrast to that of the DPA emitter in which the $T_1 \rightarrow S_0$ transition dipole moment was drastically enhanced by oxygen molecules.³ This is reasonable, as DPA emitter molecule containing polycondensated aromatic rings has a near-completely forbidden $T_1 \rightarrow S_0$ transition (decay rate, 200 s^{-1}), while the BTTQD 2 emitter in this work, containing a dimer structure with heteroatoms inserted into the framework of polycyclic aromatic hydrocarbons, is more relaxed in spin forbiddance (decay rate, 23400 s^{-1}), therefore influenced less by the paramagnetic oxygen molecules.

Action: We have included Eq. R1.5 and Table R1 as equation 9 and Table S1 in the revised supporting information, and included corresponding discussions in section 4.3 in the revised manuscript. Moreover, we have added these comments in the revised manuscript on page 11, paragraph 1 as follows:

“Moreover, oxygen molecules were found to have very limited influence on the TTET rate, which was further verified by calculating the ratio of TTET rate in presence of oxygen molecules and the one in absence of oxygen molecules using the photophysical

parameters of the paired molecules via the previously developed method (Section 4.3 in Supporting Information).⁴⁴ This observation is in sharp contrast to the typically investigated standard Pd-OEP and DPA TTA system, where oxygen molecules was observed to enhance TTET rate by orders of magnitude. The distinct influence of oxygen molecules on the two TTA systems is reasonable, as DPA emitter molecule containing polycondensated aromatic rings has a near-completely forbidden $T_1 \rightarrow S_0$ transition (decay rate, 200 s^{-1}), more prone to be influenced by the paramagnetic properties of oxygen molecules, while the BTTQD2 emitter in this work, containing a dimer structure with heteroatoms inserted into the framework of polycyclic aromatic hydrocarbons, is relaxed in spin forbiddance (decay rate, 23400 s^{-1}), therefore influenced less by the paramagnetic effect of oxygen molecules.”

(2) What is the effect of molecule oxygen on single molecules of sensitizer and emitters?

These data should be reported at least in the supporting information file. What are the phosphorescence/fluorescence yield in presence/absence of oxygen?

Response: We have included the decay lifetimes of the donor IR806 triplet and the emitter BTTQD 2 triplet, the IR806 fluorescence quantum yield, the IR806 triplet phosphorescence quantum yield, the BTTQD 2 fluorescence quantum yield, TTA-UC quantum yield, the intersystem crossing efficiency, and the TTET efficiency in presence and absence of oxygen as Table S2 in the revised supporting information (Table R2 in the response letter). As one can see, all parameters, except IR806 triplet quantum yield in absence of BTTQD 2, which decreases slightly, clearly demonstrating that our bicomponent system have high resistance ability to oxygen molecules. Moreover, we use both absorption and photoluminescence spectra to probe structural stability of IR806 and BTTQD 2 (Figure R1.3 and R1.4), in which both molecules show stable over 7 days in presence of oxygen molecules. We have included Figure R1.3 and R1.4 as Figure S7 and Figure S13 in the revised manuscript. These results confirm that both photophysical parameters and molecule structures in our bicomponent TTA-UC system are resistant to oxygen molecules.

Figure R1.3. Absorption (a, deaerated; c, aerated) and emission (b, deaerated; d, aerated) spectra from IR806 observed over 7 days (chloroform, $c = 1 * 10^{-5}$ M).

Figure R1.4. Absorption (a, deaerated; c, aerated) and emission (b, deaerated; d, aerated) spectra of BTTQD 2 observed over 7 days (chloroform, $c = 1 * 10^{-4}$ M).

Absolute fluorescence quantum yields of both IR 806 and BTTQD2 were evaluated using an integrating sphere following the following expression:

$$\Phi = \frac{\int L_{emission}}{\int E_{reference} - \int E_{sample}} \quad (\text{Eq. R1.6})$$

where $L_{emission}$ is the emission spectrum of the sample, E_{sample} is the spectrum of the incidence excitation light not absorbed by the sample, and $E_{reference}$ is the spectrum of the excitation light not absorbed by the reference in the sphere. Pure organic solvent (chloroform) was used as reference samples and xenon lamps are used as excitation light sources. The aerated sample was prepared in an atmospheric environment and the deaerated sample was prepared in a glove box using an anhydrous solvent. The fluorescence quantum yield values of IR806 and BTTQD **1-3** are summarized in Table R2, which corresponds Table S2 in the revised Supporting Information.

Table R2. Key parameters of the compounds in the absence and presence of oxygen.

	deaerated	aerated
$\tau_{S_1(S)}$	1.18 ns ^a	1.15 ns ^a
	0.96 ns ^b	—
$\tau_{S_1(A)}$	18.4 ns ^a	18.4 ns ^a
	4.86 μ s ^a	3.21 μ s ^a
$\tau_{T_1(S)}$	4.57 μ s ^b	3.01 μ s ^b
	42.8 μ s ^c	41.2 μ s ^c
$QY_{S_1(S)}$	7.0% ^d	6.8% ^d
$QY_{S_1(A)}$	62.8% ^d	62.7% ^d
$QY_{T_1(S)}$	44.0% ^e	—
$QY_{T_1(A)}$	—	—
$\bar{\Phi}_{UC,g}$	4.5% ^f	4.0% ^f
Φ_{ISC}	44.0% ^g	—
Φ_{TRET}	71.8% ^h	60.8% ^h
Φ_{TTA}	95.3% ⁱ	95.9% ⁱ

^a The data are derived from classical transient state measurements in chloroform $c = 1 * 10^{-5} \text{ M}^{-1}$, $\lambda_{ex} = 785$ and 405 nm for sensitizer and emitter, respectively. ^b The data are

determined with femtosecond/nanosecond transient absorption spectroscopy in chloroform. ^c The triplet lifetime of the emitter is estimated by the formula $2\tau_{UC} = \tau_T^A$ (vide infra). ^d Fluorescence quantum yields were measured at room temperature using an integrating sphere. ^e The triplet quantum yield at low temperatures is assumed to be equal to the ISC efficiency (vide infra). ^f Upconversion quantum yield (normalized) with ICG as standard (12% in DMSO), the bar (-) above the parameter represents the result after the value is normalized. ^g It is evaluated by the decay kinetics of the singlet state and the rise kinetics of the triplet state in the femtosecond transient absorption spectra. ^h The data are determined with nanosecond transient absorption spectra in chloroform. ⁱ The TTA efficiency is estimated by the Eq R1.8 (vide infra). ⁻ Not observed. The letter S stands for sensitizer, A for annihilator.

We are unable to directly measure the phosphorescence yield (Φ_{PL}) of IR806 triplet using an integrating sphere. Instead, when all non-radiative pathways are assumed to be suppressed at 77K, the triplet quantum yield of IR806 in the absence of oxygen should equal the intersystem crossing efficiency ($\Phi_{PL} = \Phi_{ISC}$) of 44.04% (see Figure 3a). With the introduction of oxygen, a new triplet decay channel is activated (Φ_{O_2}), and the evaluation of Φ_{PL} should consider oxygen quenching ($\Phi_{PL} = \Phi_{ISC}(1 - \Phi_{O_2})$). According to the decay rate of IR806 triplet lifetime after the introduction of oxygen, the Φ_{O_2} can be estimated to be 34.14% (see Figure 3c). Therefore, the Φ_{PL} of IR806 under aerated condition could be assessed as 29.01%. For BTTQD emitters, the large energy gap between T_1 and S_1 and the almost completely spin-forbidden nature of the $T_1 \rightarrow S_0$ transition results in phosphorescence with very low quantum yield, theoretically close to ~ 0 , regardless of the presence of oxygen molecules.³

Action: We have added the procedure for PLQYs measurement in section 5.2, and include Table R2 as Table S2 in the revised supporting information.

(3) Page 4 line 89:

“Single crystal x-ray structure studies revealed pi-pi and S-N locking 90 interactions between dimers, favoring intermolecular triplet fusion (Figure S7)” Please explain this sentence, why pi-pi interactions should help the intermolecular triplet fusion? In general, the interaction can change the triple electronic energies and recombination rates, with

also detrimental interactions.

Response: We thank the reviewer for pinpointing the inappropriate description. In fact, our intention was to emphasize that the dimer molecules form a one-dimensional chain structure by sliding packing. In the one-dimensional chain, π - π interaction exists between the curved parts of neighboring molecules, with a π -plane distance of 3.32 Å, and this tight interaction stabilizes the single crystal structure of the crystal to a certain extent.

Action: To avoid confusion, we have removed this sentence in the revised manuscript and added corresponding clarification on page 4, paragraph 1 as follows:

“Single crystal x-ray structure studies reveal that BTTQD **1** form a one-dimensional chain structure by sliding stacking. In the one-dimensional chain, π - π interaction exists between the bending parts of adjacent molecules, and the π plane distance is 3.32 Å. Meanwhile, because of the introduction of the thiadiazole terminal group, N-S and N-N interactions existed among the molecular chains in the same layer, and N-S interaction distance was 3.10 Å and N-N interaction distance was 3.05 Å (**Figure S8**). The abundant interaction makes the stacking structure of crystals more stable to some extent. The crystal data and structure refinements of BTTQD **1** are summarized in **Table S3**.”

(4) Page 6 line 144:

The upconversion QY values are reported as normalized to 100%. As recently discussed, this should be explicitly declared to allow reliable comparison with literature (Chemical Physics Reviews 2022, 3 (4), 041301 - ACS Energy Lett. 5 (7), 2322–2326 (2020).) at which excitation intensity these values have been recorded?

Why so low value is observed? It could be due to a back energy transfer from upconverted singlets of the emitters towards the sensitizers? As showed in Fig. 1c, there is a not negligible overlap between the emitter photoluminescence and the sensitizers absorption. This back transfer should be investigated by time resolved photoluminescence spectroscopy. This could be the reason why at emitter concentration

higher than 80 – 100 μM the process yield starts to decrease (Fig. S14).

Response: The UCQY values reported in Figure 1d were normalized to 100%, and measured with an excitation power density of 10 W/cm^2 , above the TTA thresholds ($< 5.7 \text{ W/cm}^2$). We realize that it is inappropriate to term normalized values as UCQYs, and therefore used the term of “normalized upconversion emission efficiency” ($\bar{\Phi}_{UC}$) instead in the revised manuscript⁵⁻⁶.

Indeed, as pointed out by the reviewer, the back energy transfer from the BTTQD 2 emitters to IR806 sensitizers affect $\bar{\Phi}_{UC}$ due to the non-negligible spectral overlap between IR 806 absorption spectra and BTTQD 2 emission spectra. A lifetime attenuation of about 2.1 ns was observed when IR806 was mixed with high concentrations of BTTQD 2 ($c_{\text{IR806}} : c_{\text{BTTQD 2}} = 1:10$). The back energy transfer efficiency (Φ_{FRET}) was evaluated to be 12.5% (Figure R1.5). We can therefore assess the normalized internal upconversion emission efficiency, $\bar{\Phi}_{UC,g}$, using $\bar{\Phi}_{UC,g} = \bar{\Phi}_{UC} / (1 - \Phi_{\text{FRET}})$ by considering the back energy transfer from the emitter to the sensitizer (or donor). For the IR806-BTTQD 2 system, $\bar{\Phi}_{UC,g}$ was evaluated to be $\sim 4\%$ in aerated chloroform.⁵

Figure R1.5. Decay lifetimes of the spin-singlet state of BTTQD 2 in the freely-diffusing form (black) and in the IR806-BTTQD 2 (red) mixture (chloroform, $c_{\text{IR806}} =$

10 μM , $c_{\text{BTTQD } 2} = 100 \mu\text{M}$).

Yet, this value is several fold lower than that of well-investigated PtOEP-DPA TTA system in absence of oxygen. It is well known that $\bar{\Phi}_{UC,g}$ can be expressed as:

$$\Phi_{uc,g} = f \times \Phi_{ISC} \times \Phi_{TTET} \times \Phi_{TTA} \times \Phi_F \quad (\text{Eq. R1.7})$$

Where Φ_{ISC} , Φ_{TTET} , Φ_{TTA} , Φ_F are quantum efficiencies of the intersystem crossing, TTET, TTA, and the emitter fluorescence, while f is the spin-statistic factor for the TTA process. For Φ_{TTA} , it is described by the relation⁷

$$I_{UC}(t) \propto \left(\frac{1 - \Phi_{TTA}}{e^{k_T t} - \Phi_{TTA}} \right)^2, \quad (\text{Eq. R 1.8})$$

where k_T is the decay rate of emitter triplet and I_{UC} is the intensity of delayed fluorescence at a specific time. The fit of the experimental data yields $k_T = 2.34 * 10^4 \text{ s}^{-1}$ and $2.43 * 10^4 \text{ s}^{-1}$, and $\Phi_{TTA} = 95.3\%$ and 95.9% in the absence and presence of oxygen, respectively (power density: 10 W/cm^2). Clearly, Φ_{TTA} is not the primary cause of low $\bar{\Phi}_{UC}$ in the IR806-BTTQD 2 system. In our bicomponent TTA system in deaerated chloroform, $\Phi_{ISC} = 72.2\%$, $\Phi_{TTET} = 71.8\%$, $\Phi_F = 62.7\%$, while the corresponding ones in the PtOEP-DPA TTA system are $\Phi_{ISC} = 100\%$ (1.39-fold higher), $\Phi_{TTET} = 98\%$ (1.36-fold higher), $\Phi_F = 96\%$ (1.53-fold higher). Multiplication of these different folds will result in about 4-fold lower TTA-UC efficiency for our system than the PtOEP-DPA system. On the other hand, we need to note that $\Phi_{uc,g} \approx 4\%$ could also be obtained in aerated conditions (almost unchanged compared to the deaerated condition). While the upconversion quantum yield of the PdOEP-DPA TTA system decreases close to zero (near-complete quenching) in the presence of oxygen.⁸ This means that although our TTA system has lower upconversion emission efficiency than the well-established standard models with excitation in the visible range (532 nm), it is advantageous for real-world applications, usually enriched with oxygen molecules, due to the oxygen immunity. Moreover, we believe that further optimizing molecular structures to improve these photophysics parameters can further improve the TTA-UC efficiency.

Action: We have added the incident power density, included Figure R1.5 as Figure S17, and added the corresponding discussions on page 6, paragraph 1 as follows:

“We evaluated the normalized upconversion emission efficiency ($\bar{\Phi}_{UC}$) of IR806/BTTQD **1-3** to be 2.28%, 3.46%, and 1.33% in aerated chloroform, respectively (Incident power density ≈ 10 W/cm²).^{5 6} It is should be pointed out that the upconversion parameters in this work ($\bar{\Phi}_{UC}$) are lower than those of some previously reported systems. One factor that cannot be ignored is the back energy transfer from BTTQD2 to IR806 that quenches TTA-UC. We evaluated the back energy transfer efficiency (Φ_{FRET}) to be 12.5% (**Figure S17**), therefore determining the normalized internal upconversion emission efficiency, $\bar{\Phi}_{UC,g} = \bar{\Phi}_{UC}/(1 - \Phi_{FRET})$, to be $\sim 4\%$ in aerated chloroform.^{5”}

We have added the following corresponding discussion regarding the low efficiency in the revised manuscript on page 13, paragraph 2:

“However, current system has a lower normalized upconversion emission efficiency ($\bar{\Phi}_{UC,g}=4\%$, incident power density 10 W/cm²) in comparison with the well-established PtOEP/DPA TTA-UC system due to the lower quantum efficiencies of ISC, TTET, and the emitter fluorescence, $\Phi_{ISC} = 72.2\%$, $\Phi_{TTET} = 71.8\%$, $\Phi_F = 62.7\%$ (in deaerated case), as compared to the ones in the PtOEP-DPA TTA system, $\Phi_{ISC} = 100\%$ (1.39-fold higher), $\Phi_{TTET} = 98\%$ (1.36-fold higher), $\Phi_F = 96\%$ (1.53-fold higher), resulting in about 4-fold lower TTA-UC efficiency assuming the two systems have identical spin-statistic factor. Designing emitter molecules with more stable structures and higher fluorescence quantum yield, and introduction of heavy atom in the sensitizer, can improve the upconversion emission quantum yield significantly.”

(5)- Page 9 line 208

The authors calculate the sensitizer to emitter ET rate by looking at the change in time of the sensitizer transient singlet absorption band lifetime at 500, which accelerates in presence of the emitter. But I guess they should look at the triplet state transient

absorption, which lifetime should be significantly reduced in the case of efficient ET. IN any case the ET rate estimate here, in terahertz regime, does not correlate at all with the rise time of the upconverted emission reported in the Fig. S13 (megahertz scale) which should mirror the ET process. Some comparison with complementary ET rate measurements are need to clarify this point and the flowing discussion on the ET step in the system.

Response: We measured the decay time at 500 nm and the rise time at 585 nm of IR 806 in femtosecond TA (Figure 3a and b) to investigate its intersystem crossing rate and efficiency in absence and presence of BTTQD 2. We did use nanosecond TA to measure triplet decay at 580 nm (Figure 3c and d) to investigate the TTET rate between the sensitizer and the emitter. We found that after the introduction of the emitter, the triplet lifetime of IR806 decreases from 3.01 μs to 1.18 μs , and the TTET rate was evaluated to be $5.56 \times 10^5 \text{ s}^{-1}$ in aerated chloroform. Similarly, in the case of deoxygenation, TTET rate was evaluated as $5.15 \times 10^5 \text{ s}^{-1}$, consistent well with the estimated TTET rate of $5.74 \times 10^5 \text{ s}^{-1}$ and $5.21 \times 10^5 \text{ s}^{-1}$ in absence and presence of oxygen, obtained from the classical Stern-Volmer studies (consult Figure R 1.1). As such, the obtained TTET rate are on the same scale as the rise rate (2 μs , megahertz scale) of TTA-UC emission (Figure S15 or Figure R1.7).

Action: We performed a systemic investigation of TTET rate using the classical Stern-Volmer equations, and included the corresponding methods in Sections 4.2, and corresponding results in Figure S22 and Figure S23 in the revised supporting information.

(6)- Page 9 line 220

“The observed 3-fold higher intersystem crossing rate of IR806-BTTQD 2 (kISC= $6.1 \times 10^9 \text{ s}^{-1}$ 220) than that of IR806 (kISC= $1.7 \times 10^9 \text{ s}^{-1}$ 221) further confirmed the occurrence of TTET, which expedites the inter system crossing via depleting triplets of IR 806 via BTTQD 2”

These conclusions must be supported more strongly. Considering the ET is a diffusion

limited process in solution, so I wonder how it can be so fast to be competitive with the ISC process at the dye concentrations employed. Moreover, the ISC occur prior to ET, therefore all molecules during excitation does not have electron in the triplet state. So, the ET should not affect at all the ISC. The observed variation should be ascribed to other mechanism due to the presence of the UC emitter.

Response: We misinterpreted the data and fully agree to the reviewer's point that the ISC can not be affected by TTET. We believe that the enhanced k_{ISC} rate of IR806 is due to the heavy atom effect of sulfur (S) in BTTQD 2, which can enhance the spin orbit coupling (SOC) proportional to Z_{eff}^4 , especially if the emitter concentration is much larger than that of the sensitizer.⁹⁻¹¹ To prove this, we studied the effect of various BTTQD 2 concentrations on the rise time of IR806 triplet state. As shown in Figure R1.6, elevated BTTQD 2 concentrations indeed accelerate the rise process for populating the triplets, confirming that the heavy atom effect of sulfur in BTTQD 2 increase k_{ISC} of IR806.

Figure R1.6. Transient kinetics for populating IR806 triplet (T_1) in a solution containing IR 806 and BTTQD 2 with varied molar ratios of IR806: BTTQD 2 = 1:0, 1:5, and 1:10 (IR 806 is fixed at $0.5 \cdot 10^{-3}$ M). The excitation is performed with a nanosecond laser at 800 nm with a pulse energy of $2 \mu\text{J}$.

Action: We have corrected the misinterpretation (see below) on page 10, paragraph 1 in the revised manuscript, and included Figure R1.6 as Figure S20 in the revised supporting information.

“Kinetic analysis at 500 nm and 580 nm revealed a decay time with a rate of $1.05 \times 10^8 \text{ s}^{-1}$ and a rise time with an average rate constant of $4.59 \times 10^8 \text{ s}^{-1}$ (k_{ISC}), illustrating an intersystem crossing (ISC) efficiency of 44.0%. With the presence of BTTQD 2 annihilator, kinetic analysis at 500 nm showed almost constant decay of singlet excited state absorption ($1.10 \times 10^9 \text{ s}^{-1}$), but an accelerated $k_{ISC} = 7.94 \times 10^8 \text{ s}^{-1}$, corresponding to an ISC efficiency of 72.2% (Figure 3b). Notably, observation of higher ISC efficiency of IR806 in presence of BTTQD 2 can possibly be attributed to the heavy atomic effects of sulfur atoms in the BTTQD structure.⁹⁻¹¹ The dependence of IR806 T₁ rising time on the concentration of the BTTQD 2 further confirmed this (Figure S20).”

(7)- Figure S13.

The author present the data obtained by time resolved spectroscopy of the upconverted emission. The provided comment are not clear at all. First, the rise time mirrors no only the TTA rate that created upconverted singlets, but mainly the ET rate that produced the triplet population. This rise time should therefore be discussed referring to the ET rate. Second, the upconversion emission intensity does not reach plateau, so the steady state equilibrium condition is not reached. Which laser excitation is used here? What is the pulse width and the energy per pulse?

Third, the upconversion emission decays quickly, with no very long time signal in the ms time scale. The intensity decay appears to be a single exponential function. Is that realistic? The excitation intensity is so low to have the limit case in which TTA is negligible and therefore we are looking at the emitters triplet lifetime? Again excitation conditions are mandatory to point out what happen at get much details and information on the proposed system (The Journal of Chemical Physics 153 (11), 114302)

Response: We appreciate the reviewer's comments that helped us better understand the dynamic processes in our system. First, we agree with the reviewer on the comment on

the rise time should mainly account for TTET process. In the upconversion lifetime measurement (Figure R 1.7, down panel), we observed a rise time about 2 μs ($k_{S1-rise}^A$), which is commensurate with TTET rate of $5.15 * 10^5 \text{ s}^{-1}$ (see our response to question 5 of reviewer 1), confirming the conclusion.

Second, the upconversion luminescence decay curves were collected by Edinburgh FLS1000 spectrofluorometer equipped with an 808 nm diode laser operating in pulse mode (pulse width: 40 μs , pulse train: 50 KHz, energy per pulse: 0.2 μJ for low power and 10 μJ for high power).

Third, according to the comment of reviewer, we remeasured the TTA-UC decay process at both low and high excitation powers. Time-resolved measurements of 650 nm TTA luminescence under 808 nm laser excitation confirm the upconversion delayed fluorescence generation. At low excitation power (0.2 μJ per pulse), the upconversion emission intensity decreases exponentially with time, with a characteristic decay time $\tau_{UC} = 21.4$ and $20.6 \mu\text{s}$ for deaerated and aerated sample, respectively. In this case, the primary inactivation channel of the emitter triplet is spontaneous radiative/non-radiative decay, while the TTA is negligible, so the lifetime of the emitter triplet at this time can be estimated to be twice the emission lifetime of the upconversion ($2\tau_{UC} = \tau_T^A$). This analysis gives a τ_T^A of $\sim 40 \mu\text{s}$ for the IR806-BTTQD **2** system.^{7, 12-13} By increasing the excitation power density up to 10 μJ per pulse, the concentration of the receptor triplet states is thus increased, which makes TTA annihilation more efficient for a short time and ultimately results in a shortened average lifetime of the upconversion emission. This result is consistent with the observed evolution of emitter triplet density with time and excitation intensity.

Figure R1.7. Time-resolved luminescence decay of TTA-UC emission (650 nm) from BTTQD 2 under photoexcitation at ~808 nm in deaerated (a) and aerated (b) chloroform. The corresponding rising dynamics of IR806-BTTQD 2 is shown below (a) and (b) under low power excitation. Low power: 0.02 mJ per pulse, High power: 1 mJ per pulse, pulse width: 25 KHZ, $c_{IR806} = 10 \mu M$, $c_{BTTQD2} = 100 \mu M$.

Action: Figure S13 was replaced with Figure R1.7, which was included as Figure S15 in the revised supporting information. The excitation details are included the Figure caption, while corresponding discussions here have also been added below Figure S15.

In addition, we have clarified this in the revised main text on page 6, paragraph 1:

“The observed long rise time, with a timescale about 2 μs in both aerated and deaerated samples, signifies the taking place of triplet fusion upconversion (**Figure S15**)”

(8) FIGURES:

Figure captions and panel should be carefully revised for example to avoid that labels and legends overlap the data plots and to give the required detailed information of the reported data crucial for the reader.

Response: Following the suggestion of the reviewer, we carefully revised the figure

captions and panels to resolve the overlapping problem, and added the relevant experimental conditions and measurement details to the corresponding figure captions.

Action: Experimental details have been added to Figures 2 and 3, Figure S7, Figure S13, Figure S15, Figure S17, and Figures S20-24. In addition, we have carefully revised figures, avoiding overlapping labels with data plots.

Reviewer #2 (Remarks to the Author):

In this article, the authors presented a class of near-infrared (NIR) TTA upconversion with eigen oxygen immunity consisting of IR806 molecules as sensitizers and three chemically synthesized BTTQD dyes with defined dimer structure as annihilators. The authors claimed that such TTA system can be used for sensitive and specific detection of ONOO⁻ and related hepatotoxicity in vivo. Basically, the concept of TTA is not new, the authors utilized IR806 molecules as sensitizers and chemically synthesized BTTQD dyes as annihilators, which put in some new components in TTA. However, such technical input, although interesting, does not greatly change TTA performance as compared to other counterparts in conventional TTA. Importantly, the authors did not present the rationale and necessity why TTA here can be used for free radical sensing, especially many well-functioned probe systems like small molecule, polymers and nanoparticles etc have been established and performed very well for radical species imaging in vitro and in vivo. Along the whole system, many important information is missing in the main context. Together with the weak points below, the novelty and current scope studies do not support the manuscript published in Nature Communications.

Response: We thank the reviewer for the comments. Indeed, TTA processes have been observed for decades, and many TTA systems with high upconversion quantum yields have been reported, including full-organic,¹⁴⁻¹⁵ metal-organic,¹⁶⁻¹⁷ and hybrid systems.¹⁸⁻¹⁹ However, absolutely most or even all reported systems suffer from severe oxygen quenching of upconversion, hindering their real-world applications such as in vivo applications that are pervasive with oxygen molecules.

In this work, we introduce a new type of NIR-excitable biocomponent molecular TTA-UC system with high resistance to oxygen, reveal the underpinning photophysics, and apply it successfully to in vivo sensing. Our results give the solid demonstration that molecular TTA-UC systems can be oxygen-insensitive, thereby having promise for real word applications. We believe that our work would open numerous opportunities and excite immediate interests in applications of TTA-UC systems toward industrial uses. We hope the reviewer can agree to our point.

We are sorry that we did not present the rationale and necessity why TTA here can be used for free radical sensing in vivo due to our unclear writing. Indeed, many well-functioned probe systems like small molecule, polymers and nanoparticles have been established and performed well for radical species imaging in vitro or ex vivo, but few of them have been used in vivo (*c.f.* Table S7 in SI). This is because most excitations are in the visible range (400-700 nm), which have limited penetration to biological tissues and could induce substantial tissue autofluorescence background, lowering imaging contrast and detection sensitivity (*Chemical Reviews*, 2016, 116 (5), pp 2826–2885). Tissue has a transparent biological window in the spectral range of 650-1000 nm, allowing light to penetrate deep into tissue (~ up to centimeter). As both excitation (808 nm) and emission (~ 680 nm) are within this window, light scattering and absorption would be minimized, thus allowing high contrast deep tissue imaging and sensing (see Figure S26). On the other hand, TTA-UC is an anti-Stokes emission which enables imaging and detection virtually with zero-background, since autofluorescence, if any, will have longer wavelength than the excitation while TTA-UC signal has shorter wavelength than the excitation. Moreover, the nonlinear two-photon nature of TTA-UC (below threshold) offers higher spatial imaging due to confined excitation. Importantly, TTA-UC materials can achieve optical upconversion emission at an excitation intensity acceptable to organisms (all in vivo experiments use a uniform power density of 100 mW/cm² in this work). These advantages make optical upconversion through TTA very attractive for in vivo imaging and sensing applications (*Biomaterials*, 2019, 201, 77-86).

Action: We have added the following comments in the revised manuscript to clarify the rationale and necessity for the use of TTA here for free radical sensing:

page 14, paragraph 1: “As both excitation (808 nm) and emission (~ 680 nm) are within the first biological window (650-1000 nm), light scattering and absorption would be minimized, thus allowing high contrast deep tissue imaging and sensing. A lateral resolution of 1 mm was demonstrated for imaging through biological tissue of thickness of 5 mm, showing their promise for deep-tissue biophotonics (Figure S26).”

Page 16, paragraph 1: “TTA-UC materials can achieve optical upconversion at

excitation power densities acceptable to organisms (all *in vivo* experiments use a uniform power density of 100 mW/cm² in this work). Moreover, the anti-Stokes emission enables imaging and detection virtually with zero-background, since autofluorescence, if any, will have longer wavelength than the excitation while TTA-UC have shorter wavelength than the excitation.²⁰ These advantages are not available in organic small molecules, inorganic semiconductor nanomaterials (QDs or CDs) and lanthanide-doped upconversion nanoparticles (UCNPs).²¹⁻²³ Therefore, TTA-based materials would be more suitable for imaging and sensing applications *in vivo*. Especially, considering that conventional TTA-UC systems would be significantly quenched in after exposure to oxygen, that are enriched in blood, rendering them challenge for *in vivo* applications,²⁴ while the insensitivity to oxygen molecules single out our TTA-UC system for *in vivo* imaging and sensing applications.”

Regarding the specific comments, we will provide our point-to-point response below.

Main concerns:

1. Figure 2c, Figure 4e and others, to check the oxygen resistance and stability in aqueous solutions, the laser power density of 100 W/cm² was applied, will the molecules survive under such large power irradiation? And such high dosage may also limit the applications in living systems.

Response: We are sorry for this apparent typo, which should be 100 mW/cm² instead of 100 W/cm². We have corrected this problem in the revised manuscript.

Action: 100 W/cm² has been corrected into 100 mW/cm².

2. Similarly, figure 2 c, d, under the degassed conditions, the some certain photostability ~ 2hr was reported here, which is not long in terms for bioimaging and other studies. What about prolonged the stability? And also, the polymer encapsulated TTA- NMs in Figure 4e, the stability is not so good.

Response: Per the suggestion of reviewer, we extended the time scale of the

photostability test to 8 hours. As expected, TTA-UC of IR806-BTTQD **2** in both aerated and vacuum-degassed solution maintained $> 80\%$ upconversion under prolonged 808 nm laser irradiance (100 mW/cm^2), showing high resistance to oxygen molecules (Figure R2.1, Figure 2c and 2d in the revised manuscript).

Figure R2.1. (a) Emission stabilities of upconversion in IR806-BTTQD **2** and PtOEP-DPA solutions with/without degassing under continuous laser irradiation at 808 and 532 nm (100 mW/cm^2). (b) Photographic images of TTA upconversion emission in (c) taken at varied time in aerated solution.

Figure R2.2. (a) Upconversion emission stability of TTA-NMs in PBS buffer over 7 days (Ex: 808 nm, 100 mW/cm^2 , fluorescence spectra were measured after continuous light exposure for two hours each day). (b) Size distribution of TTA-NMs dispersed in PBS buffer over 7 days.

In the previous version of the manuscript, the photostability of TTA-NMs were

investigated under 808 nm continuous laser irradiation (fluorescence spectra were measured after exposure to continuous-wave 808 nm laser for two hours each day), retaining over 70% of initial TTA-UC emission intensities after 3 days. We further optimized the preparation process of nanomicells by increasing the feeding ratio of F127-NH₂ to TTA biocomponent molecules. The optimized TTA-NMs retained >80% of its initial intensities after 7 days (Figure R2.2 a), and light exposure for two hours each day was also applied. Moreover, the DLS size remains almost unchanged over 7 days, indicating the high colloidal stability of TTA-NMs in PBS buffer, favorable for bioapplications.

Action: Figure R2.1a and b has been used to substitute Figure 2c and 2d in the revised manuscript. Figure R2.2a has been included as Figure 4e, and Figure R2.2b has been included as Figure S32a in the revised supporting information.

Page 13 paragraph 1: “remarkable upconversion photostability was demonstrated in PBS buffer under intermittent irradiation of 808 nm laser for 7 days (100 mW/cm², fluorescence spectra were measured after continuous light exposure for 2 h each day).”

Page 16 paragraph 1: “The results showed that the particle size and absorption intensity of TTA-NMs and Cy7-coated TTA-NMs did not change in the simulated *in vivo* environment, indicating the less likelihood of agglomeration and component loss of these nanoparticles *in vivo* (Figure S32 and S33).”

3. Looks like the QY values of TTA-UC IR806/BTTQD 1-3 are not high and efficient?

Response: The normalized upconversion emission efficiency of IR806/BTTQD 2 was determined to be 4% after correcting the back energy transfer from BTTQD 2 to IR806. Indeed, this value is not as high as that of well-established TTA system with excitations in the visible range, such as PtOEP/DPA (~ 20% normalized upconversion emission efficiency, excited at 532 nm), but it is considerable when comparing TTA-UC with excitation in the NIR range (IR806-Rubrene, 1.3%, $\lambda_{\text{ex}} = 808 \text{ nm}^{25}$; PdPc-FDPP, 3.2%, $\lambda_{\text{ex}} = 730 \text{ nm}^{26}$). Importantly, our TTA-UC system are stable in oxygen environment over one week with near-unaltered emission intensity, while conventional highly

efficient TTA-UC system such as PtOEP/DPA will be quenched completely after exposure to oxygen within half day (Figure 2a). In addition, we have carefully investigated the photophysics of our TTA-UC system, and found that the lower efficiency compared with the well-established PtOEP/DPA TTA UC system is mainly due to the lower quantum efficiencies of ISC, TTET, and the emitter fluorescence, $\Phi_{ISC} = 72.2\%$, $\Phi_{TTET} = 71.8\%$, $\Phi_F = 62.7\%$, as compared to the ones in the PtOEP-DPA TTA system, $\Phi_{ISC} = 100\%$ (1.39-fold higher), $\Phi_{TTET} = 98\%$ (1.36-fold higher), $\Phi_F = 96\%$ (1.53-fold higher), resulting in about 4 -fold lower TTA-UC efficiency assuming the two systems have identical spin-statistic factor (*please see our response to the comment 4 of reviewer 1*). We believe that designing emitter molecules with more stable structures and higher fluorescence quantum yield, and introduction of heavy atom in the sensitizer, can improve the upconversion emission quantum yield significantly.

Action: We have added the corresponding discussion in the revised manuscript on page 13, paragraph 2 as follows:

“However, current system has a lower normalized upconversion emission efficiency ($\bar{\Phi}_{UC,g} = 4\%$, incident power density 10 W/cm^2) in comparison with the well-established PtOEP/DPA TTA-UC system due to the lower quantum efficiencies of ISC, TTET, and the emitter fluorescence, $\Phi_{ISC} = 72.2\%$, $\Phi_{TTET} = 71.8\%$, $\Phi_F = 62.7\%$ (in deaerated case), as compared to the ones in the PtOEP-DPA TTA system, $\Phi_{ISC} = 100\%$ (1.39-fold higher), $\Phi_{TTET} = 98\%$ (1.36-fold higher), $\Phi_F = 96\%$ (1.53-fold higher),³ resulting in about 4-fold lower TTA-UC efficiency assuming the two systems have identical spin-statistic factor. Designing emitter molecules with more stable structures and higher fluorescence quantum yield, and introduction of heavy atom in the sensitizer, can improve the upconversion emission quantum yield significantly.”

4. And in Figure 4, TTA was encapsulated within polymer nanoparticles, some very important data like entrapment efficiency of each TTA component inside the particles

was missing.

Response: We evaluated the encapsulation efficiency of TTA components in nanomicelles by comparing characteristic absorption difference of IR806 and BTTQD2 before and after encapsulation in TTA (Figure R2.3). The encapsulation efficiency of IR806 and BTTQD 2 were quantified to be 31% and 73%, respectively.

Figure R2.3. Absorption spectra of IR806, BTTQD 2, and TTA-NMs.

Action: We have included Figure 2.3 as Figure S25, and added the corresponding discussion in the revised manuscript on page 13, paragraph 3:

“By comparing the absorbance changes before and after the experiment, we evaluated the encapsulation efficiency of IR806 and BTTQD 2 to be 31% and 73%, respectively (Figure S25).”

5. In order to sustain “on-off” effect, sounds like extra Cy7 was amidated onto TTA-NMs. The rationale why use Cy7 (also IR806 already inside polymer structure) and detailed characterization of Cy7 coating (e.g. coating efficiency etc..) all are missing.

Response: We selected Cy7 as the energy receptor on the surface of TTA-NMs for three reasons:

- (i) The spectral overlap between the absorption spectra of Cy7 and TTA-UC

emission spectra of IR806-BTTQD 2 system will induce energy transfer from BTTQD 2 to Cy7, resulting in TTA upconversion quenching (Figure R2.4a).

(ii) Cy7 molecules has strong absorption at 808 nm and thereby compete with IR806 sensitizer molecules for absorbing TTA excitation photons, thereby further enhancing TTA-UC quenching (Figure R2.4b).

It is the two reasons that TTA-UC was quenched by about 12-fold after loading Cy7 onto the surface of TTA-NMs (Figure 4d).

(iii) Surface Cy7 could have specific reaction with ONOO^- , cleaving the polymethyl chains in its molecular structure through one-electron oxidation reactions to cleave its absorption (Figure R2.4c).²⁷ This will cleave the TTA-UC quenching effect of Cy 7, thereby turning on quenched TTA-UC emissions (Figure R2.4d).

Figure R2.4. (a) Absorption spectra of Cy7 and the upconversion emission spectra of TTA-NMs under photoexcitation at 808 nm (100 mW/cm^2). (b) Absorption spectra of Cy7 and IR806 in H_2O , the red dotted line represents the location of the 808 nm excitation wavelength. (c) Changes of Cy7 absorption spectra before and after 30 μM

ONOO⁻ treatment. (d) Changes of the TTA-UC emission spectra of Cy7-coated TTA-NMs before and after 30 μM ONOO⁻ treatment.

In addition, we also evaluated the coating efficiency of Cy7 on TTA-NMs by spectroscopic absorption analysis (Figure R2.5). Appearance of the apparent Cy7 absorption peak in Cy 7-coated TTA-NMs demonstrates the successful amidation of Cy 7 onto the TTA-NMs surface. Moreover, according to the standard calibration curve for Cy7 (Figure 2.5 a), the amount of Cy7 conjugated to the TTA-NMs was determined to be 42.6 wt%. The ratio of Cy7 to IR806 molecules in a single nanomicelle is estimated to be about 52:1.

Figure R2.5. (a) The absorption spectra of Cy7 with different concentrations (in chloroform). Inset: the standard curve for the absorbance of Cy7 at 790 nm. (b) The absorbance change of TTA-NMs before and after loading Cy7.

Action: We have included Figure R2.4 as Figure S29, and Figure R 2.5 as Figure S31. We have also added the corresponding discussions in the revised manuscript on Page 16, paragraph 2:

“Surface-attached Cy7 dyes have three important roles: (i) Elicitation of energy transfer from BTTQD 2 to Cy7 due to spectral overlapping between Cy7 absorption and BTTQD 2 emission, resulting in upconversion quenching (Figure S29a). (ii) Competition with IR806 for absorbing TTA excitation photons, further enhancing TTA-UC quenching (Figure S29b). It is the two reasons that TTA-UC was quenched by about 12-fold after Cy7 loading (Figure 4d). (iii) Surface Cy 7 could specifically react with ONOO⁻, cleaving the polymethyl chains in its molecular structure through one-

electron oxidation reactions (**Figure S29c**),^{23, 28} thereby turning on the quenched TTA-UC for biosensing. Note that IR806 are stable towards ONOO⁻ according to selective experiments (**Figure S30**). Thus, with the introduction of ONOO⁻, Cy7 on the surface of TTA-NMs will be cleaved while the internal sensitizer IR806 will remain intact, that is, upconversion emission will be restored. According to the standard calibration curve for Cy7 based on its absorbance at 780 nm, the amount of Cy7 conjugated to the TTA-NMs was determined as 42.6 wt% (**Figure S31**). Considering that 31% of IR806 was encapsulated into the hydrophobic core of TTA-NMs, the ratio of surface Cy7 molecules to internal IR806 molecules was estimated to be ~52: 1”

6. What about stability of Cy7, Cy7 coated TTA-NMs and TTA-NMs like in blood, body? The fluorescent detection or imaging of peroxyntirite ONOO- by small molecule probes, semi-conducting polymers, and nanoparticles etc have been well established. For what reason, the authors claimed “ Yet, imaging ONOO- levels in vivo remain a challenge.....” At Page 13, line 299? This also raised the concern regarding the necessity and rationale to use TTA-NMs for ONOO imaging? Any obvious merits? The authors did not clearly show all these important information.

Response: We evaluated the long-term stability of Cy7, TTA-NMs and Cy7-coated TTA-NMs in simulated body fluids (SBF, pH = 7.4) and blood (fetal bovine serum, 10%). First, dynamic light scattering (DLS) was measured to assess the changes in particle size of TTA-NMs and Cy7-coated TTA-NMs after materials were stored in different media for various time (1-7 days). As shown in Figure R2.6, the particle sizes in different media did not show significant changes over time, indicating the outstanding stability of TTA-NMs and Cy7-coated TTA-NMs in biological mimicking systems.

Figure R2.6. The particle size distributions of TTA-NMs (a) and Cy7 coated TTA-NMs (b) over a period of 7 days in PBS buffer (PH = 7.4), simulated body fluid (SBF, PH = 7.4) and fetal bovine serum (FBS, 10%).

Next, we evaluated the absorbance of Cy7 dye, TTA-NMs, and Cy7-coated TTA-NMs over time in different media. Like the stability of the particle size, the absorbance of these materials shows a decrease of less than 5% over a time scale of 7 days (Figure R2.7). The results show that there is almost no component loss in the simulated biological environments, further proving the excellent stability of TTA-NMs and Cy7-coated TTA-NMs.

Figure R2.7. Time dependence of absorption from Cy7 dye, TTA-NMs and Cy7-coated TTA-NMs over a period of 7 days in SBF (a, PH = 7.4) and FBS (b, 10%).

Regarding the necessity and rationale to use TTA-NMs for ONOO⁻ imaging, we are sorry that we did not clearly present the advantages of TTA upconversion over other materials due to our unclear writing. Indeed, for detection of ONOO⁻, several probes

have been developed including organic small molecules and inorganic semiconductor nanomaterials²¹⁻²², but few have been used in vivo (*c.f.* Table S7 in SI). This is because most excitation light is in the visible range (400-700 nm), which has limited penetration to biological tissues and induce substantial tissue autofluorescence background, lowering detection sensitivity (*Chemical Reviews*, 2016, 116 (5), pp 2826–2885). Tissue has a transparent biological window in the spectral range of 650-1000 nm, allowing light to penetrate deep into tissue. As both excitation (808 nm) and emission (700 nm) are within this window, light scattering and absorption would be minimized, thus allowing high contrast deep tissue imaging and high sensitivity sensing (see Figure R2.8, Figure S26). On the other hand, TTA-UC is an anti-Stokes emission which enables imaging and detection with virtually zero-background, since autofluorescence, if any, would have longer wavelength than the excitation while TTA-UC has shorter wavelength than the excitation. Moreover, the nonlinear two-photon nature of TTA-UC (below threshold) offers higher spatial imaging due to confined excitation. These advantages enable optical upconversion very attractive for in vivo imaging and sensing applications. Though lanthanide-doped upconversion nanoparticles (UCNPs) have been attempted for detection of ONOO⁻ in vivo,^{23,28} the limited absorption cross section of lanthanide ions (10^{-19} cm²), 3-4 orders of magnitude lower than that of an organic dye molecule, demands high laser irradiance (> 1-5 W/cm²). In contrast, TTA-UC only requires an ultralow light irradiance such as 100 mW/cm² here. Despite these advantages, successive detection of ONOO⁻ in vivo using TTA-UC has not been achieved, as current TTA-UC nanosystems are prone to luminescence quenching in aqueous medium due to the presence of oxygen and water molecules, and are not selectively responsive to ONOO⁻ radical. Therefore, we claimed detection of ONOO⁻ in vivo remain a grand change.

Action: We have included Figure R2.6 as Figure S32, and Figure R 2.7 as Figure S33 in the revised manuscript. Moreover, we added the corresponding discussions in the revised manuscript as follows:

Page 14, paragraph 1: “As both excitation (808 nm) and emission (~ 680 nm) are within the first biological window (650-1000 nm), light scattering and absorption would be

minimized, thus allowing high contrast deep tissue imaging and sensing.²⁹ A lateral resolution of 1 mm was demonstrated for imaging through biological tissue of thickness of 5 mm, showing their promise for deep-tissue biophotonics (**Figure S26**). Moreover, remarkable upconversion photostability was demonstrated in PBS buffer under intermittent irradiation of 808 nm laser for 7 days (100 mW/cm², fluorescence spectra were measured after continuous light exposure for 2 h each day).²⁷

Page 16, paragraph 1: “Note that TTA-UC materials can achieve optical upconversion at excitation power densities acceptable to organisms (all *in vivo* experiments use a uniform power density of 100 mW/cm² in this work). Moreover, the anti-Stokes emission enables imaging and detection virtually with zero-background, since autofluorescence, if any, will have longer wavelength than the excitation while TTA-UC have shorter wavelength than the excitation.²⁴ These advantages are not available in organic small molecules, inorganic semiconductor nanomaterials (QDs or CDs) and lanthanide-doped upconversion nanoparticles (UCNPs).^{61, 63-64} Therefore, TTA-based materials would be more suitable for imaging and sensing applications *in vivo*. Especially, considering that conventional TTA-UC systems would be significantly quenched in after exposure to oxygen, that are enriched in blood, rendering them challenge for *in vivo* applications,²³ while the insensitivity to oxygen molecules single out our TTA-UC system for *in vivo* imaging and sensing applications, such as *in vivo* biosensing ONOO⁻ here.”

7. And the detection selectivity in Figure S24 is not convincing. And in Figure S25, the referee is quite skeptical of the free radical production done by APAP treatment. Lack of standard probes to clearly prove the suitability.

Response: We re-performed the detection selectivity experiment of with optimized TTA-NMs towards ONOO⁻. Specifically, various reactive oxygen species (ROS) with the same concentration (20 μM) were added to the Cy7-coated TTA-NMs solution (5 mg/mL). We evaluated the specificity of the probe to various ROS by observing the

recovery of upconversion emission at 690 nm from Cy7-coated TTA-NMs. As shown in Figure R2.9, under NIR excitation at 808 nm, ONOO^- achieved a ~90% recovery rate TTA-UC of the probe, while all other ROS showed <18% recovery rate for upconversion emission, clearly showing the high specificity of Cy7-coated TTA-NMs toward ONOO^- .

Figure R2.9. The change of upconversion emission intensities of Cy7-coated TTA-NMs (5 mg/mL in PBS) toward various analytes (20 μM). The upconversion emission spectra were acquired 1 minute after treatment TTA-NMs with different ROS/RNS.

We took two steps to confirm that APAP treatment could induce ONOO^- that causes hepatotoxicity. First, we demonstrated APAP treatment could induce hepatotoxicity. Second, we demonstrated that hepatotoxicity is due to ONOO^- . The mice were injected through tail vein with APAP and then the blood samples were collected from the vena cava at various time points (from 0~180 min, with an interval of 30 minutes). We monitored changes of aspartate aminotransferase (AST) and alanine aminotransferase (ALT), two standard biomarkers for liver disease, in blood samples to determine the development of hepatotoxicity.³⁰ As shown in Figure R2.10, the contents of AST and ALT in control groups (PBS and TTA-NMs) hardly changed with time. However, levels of two biomarkers of liver disease increased over time in the blood of APAP-treated mice, increasing by about four times compared to the initial level at 3 hours, clearly indicating that APAP induced hepatotoxicity.

Figure R2.10. Changes of AST and ALT contents over time toward different drugs treatment ($n = 3$ for each group).

Further, to demonstrate that hepatic toxicity is indeed caused by excess ONOO^- , we utilized aminophenyl fluorescein (APF) as the standard probe, which is selectively responsive to ONOO^- and whose fluorescence intensity would be greatly enhanced upon exposure to ONOO^- .³¹ As shown in Figure R2.11a, the mice injected with APAP were sacrificed 30 minutes later and the livers were harvested and converted into tissue fluids. With 100 μL of tissue fluid added to the APF solution, the fluorescence intensity of APF was enhanced by ~ 20 times (Figure R2.11b). These results indicated that the liver of mice was enriched with excessive ONOO^- after APAP treatment, leading to severe hepatotoxicity.

Figure R2.11. (a) Schematic diagram illustrating the process to obtain liver tissue fluid of the APAP treated mouse and its addition to APF solution. (b) Fluorescence spectra of APF (1 mL; 10 μ M) in the absence and presence of liver tissue fluid (Ex = 495 nm).

Action: We have included Figure R2.9 as Figure S35, Figure R 2.10 as Figure S36, and Figure R2.11 as Figure S37 in the revised manuscript. Moreover, we added the corresponding discussions in the revised manuscript as follows:

Page 18, paragraph 1: “a set of radicals, such as superoxide anion radical ($\bullet\text{O}_2^-$), hydroxy radical ($\bullet\text{OH}$), and chlorine oxide radical (ClO^-) were tested demonstrating high specific detection of ONOO^- using TTA-NMs (Figure S35)”

Page 19, paragraph 1: “Detection of hepatitis related biomarkers (AST and ALT) confirmed that APAP treatment could induce hepatotoxicity (Figure S36), while the employment of aminophenyl fluorescein (APF), a standard probe specific to ONOO^- , further confirmed that APAP induced hepatotoxicity is due to the produced ONOO^- (Figure S37).”

8. Unless the real detection limit tested, the authors can not claimed “..... which demonstrated sensitive and specific detection of ONOO⁻ via turning on upconversion,.....” as in page 15, line 365

Response: We have evaluated the detection limit of TTA-NMs for ONOO⁻ according to the formula:

$$\text{LOD} = \frac{3\delta}{K}$$

Measurement of the plot slope (K, Figure S34) and the standard deviations (δ) of five replicate measurements of blank TTA-NMs solutions (Table S6) were performed, determining the limited of detection to be ~9 nM.

Action: We have added this information on page 17, paragraph 2, as follows:

“Furthermore, measurement of the plot slope (K) and the standard deviations (δ) of five replicate measurements of blank TTA-NMs solutions (**Table S6**) determined the limited of detection ($\text{LOD} = 3\delta/K$) to be 9 nM, about 2-50 times lower than that of previous results on ONOO⁻ sensing (**Table S7**)”.

Others:

1. The authors described the synthesis of the molecules. Yet, the authors only scanned some NMR spectra, but no detailed data assignment.

Response: We have detailed the data assignment in the revised supporting information, as follows:

Figure S1. $^1\text{H-NMR}$ of BTTQD 1. $^1\text{H NMR}$ (400 MHz, Chloroform- d) δ 8.91 (d, $J = 1.9$ Hz, 2H), 8.07 (dd, $J = 8.3, 1.9$ Hz, 2H), 7.96 (d, $J = 8.3$ Hz, 2H), 7.92 (s, 4H), 1.32 (dd, $J = 8.3, 4.3$ Hz, 84H). MALDI-TOF MS, calculated for $\text{C}_{78}\text{H}_{94}\text{N}_8\text{S}_4\text{Si}_4$ [m/z] 1382.6, found 1382.6, CCDC No.: 2099068.

Figure S2. $^1\text{H-NMR}$ of BTTQD 2. $^1\text{H NMR}$ (400 MHz, Chloroform- d) δ 8.88 (d, $J = 1.8$ Hz, 2H), 8.03 (dd, $J = 8.3, 1.8$ Hz, 2H), 7.93 (d, $J = 8.2$ Hz, 2H), 7.87 (q, $J = 8.2$ Hz, 8H), 1.33 (dd, $J = 16.4, 4.7$ Hz, 84H). MALDI-TOF MS, calculated for $\text{C}_{84}\text{H}_{98}\text{N}_8\text{S}_4\text{Si}_4$ [m/z] 1458.6, found 1458.6.

Figure S3. $^1\text{H-NMR}$ of BTTQD 3. $^1\text{H NMR}$ (400 MHz, Chloroform- d) δ 9.45 (d, J = 14.5 Hz, 4H), 8.93 (s, 2H), 8.10 (s, 4H), 8.02 (d, J = 8.6 Hz, 2H), 7.90 (q, J = 10.2, 7.4 Hz, 10H), 7.55 (d, J = 7.8 Hz, 4H), 1.40 (d, J = 19.4 Hz, 84H). MALDI-TOF MS, calculated for $\text{C}_{100}\text{H}_{110}\text{N}_4\text{S}_2\text{Si}_4$ $[\text{M}+\text{H}]^+$ 1542.7, found 1543.7.

2. Fig S20, data not convincing and the trend of cell viability quite inconsistent.

Response: We re-performed the cytotoxicity study of TTA-NMs, and the concentration range of TTA-NMs for the cytotoxicity tests was extended to 0 ~ 400 $\mu\text{g/mL}$ (0 ~ 120 $\mu\text{g/mL}$ in initial manuscript). As shown in Figure R2.12, the survival rates of HeLa cell were higher than 93% in all cases after 24 h incubation.

Figure R2.12. Cell viability (%) of HeLa cells treated with TTA-NMs at different concentrations (0 ~ 400 $\mu\text{g/mL}$).

Action: We have included Figure R 2.12 as Figure S27 in the revised SI.

3. Many figures in supporting information lack of experimental error bar.

Response: We thank the reviewer for the comment. We have supplemented the error bar for the corresponding figures in the revised supporting information (Figure S16, S19, S27, S32, S33, S35 and S36).

Action: The error bar for Figure S16, S19, S27, S32, S33, S35 and S36 has been added in the revised supporting information.

Reviewer #3 (Remarks to the Author):

This manuscript reports a molecular system for triplet-triplet annihilation aided upconversion (TTA UC). The TTA UC of the system is not quenched by oxygen, which is unique. The results are interesting, I recommend acceptance of the manuscript after revisions.

Response: We thank the reviewer for the encouraging comments on our work and the recommendation for publication after revision.

Line 220-222, with intermolecular TTET, the ISC quantum yield of IR860 could not be higher than the native IR806, except there is strong interaction between IR860 and the energy acceptor, such as aggregation. The description should be clarified;

In solution, the intermolecular TTET should be on time scale of microseconds, because the process is diffusion controlled, it can not be much faster. But the authors observed fast TTET, this is unusual, more discussion should be made on this issue.

It seems the authors did not give clear, convincing explanation about the 'eigen oxygen immunity' of their system. According to the general photochemistry in solution, the conclusion is unlikely, e.g. the poorly quenched triplet state in aerated solution, fast intermolecular TTET in solution, etc.

Response: First, we thank the reviewer for the insightful comments on the ISC quantum yield. We agree that the ISC quantum yield of IR860 is independent of TTET and could not be higher than the native IR806 unless there is a strong interaction between IR806 and BTTQD 2. Femtosecond TA spectra did show an accelerated $k_{ISC} = 7.94 \times 10^8 \text{ s}^{-1}$ rate of IR806 when adding BTTQD 2 emitter, as compared to $k_{ISC} = 4.59 \times 10^8 \text{ s}^{-1}$ for pure IR806 molecules (Figures 3a, b). This is possibly due to the heavy atom effect of sulfur (S) in BTTQD 2, which can enhance the spin orbit coupling (SOC) of IR 806 that is proportional to Z_{eff}^4 , especially considering the emitter concentration is 10-fold larger than that of the sensitizer.⁹⁻¹¹ To verify this, we also investigated the concentration effect of BTTQD 2 on the rise time of IR806 triplet state, which clearly shows, indeed, faster risetimes at higher BTTQD 2 concentrations (see **Figure R1.6**, as well as the **response to comment 6 of reviewer 1**).

Action 1. We have clarified this in the revised manuscript on page 10, paragraph 1, as follows:

“Kinetic analysis at 500 nm and 580 nm revealed a decay time with a rate of 1.05×10^8 s⁻¹ and a rise time with an average rate constant of 4.59×10^8 s⁻¹ (k_{ISC}), illustrating an intersystem crossing (ISC) efficiency of 44.0%. With the presence of BTTQD 2 annihilator, kinetic analysis at 500 nm showed almost constant decay of singlet excited state absorption (1.10×10^9 s⁻¹), but an accelerated $k_{ISC} = 7.94 \times 10^8$ s⁻¹, corresponding to an ISC efficiency of 72.2% (**Figure 3b**). Notably, observation of higher ISC efficiency of IR806 in presence of BTTQD 2 can possibly be attributed to the heavy atomic effects of sulfur atoms in the BTTQD structure.⁹⁻¹¹ The dependence of IR806 T₁ rising time on the concentration of the BTTQD 2 further confirmed this (**Figure S20**).”

Second, we thank the reviewer for pointing out the erroneous ET rate in the original manuscript. The intermolecular TTET is indeed on the time scale of microseconds. We utilized both nanosecond TA spectra as well as classical Stern-Volmer equation to quantify the TTET rate, which was all calculated to be about 5.56×10^5 s⁻¹ and 5.15×10^5 s⁻¹ for deaerated and aerated samples, respectively (please see **the response to comment 1 of reviewer 1** for details).

Action 2. We have clarified this in the revised manuscript on page 12, paragraph 1, as follows:

“The TTET rate constants (k_{TTET}) of IR806-BTTQD 2 pair in absence and presence of oxygen were determined to be 5.56×10^5 s and 5.15×10^5 s, respectively (**Figure 3e**). We also re-evaluated the TTET rate and the corresponding oxygen quenching rate through classical steady/transient state measurements ($k_{TTET} = 5.59 \times 10^5$ s⁻¹ and 5.38×10^5 s⁻¹ in the absence and presence of oxygen, see **Section 4.3** and **Figure S22 and S23** in supporting information for more details), and the results showed a good correlation with the transient absorption results, further confirming the reliability of the parameters.”

Third, we believe the eigen oxygen immunity of our TTA system arise from the unique photophysics combined with its structural stability against oxygen molecules. Triplet

quenching by oxygen is the main and the well-known reason to depopulate the triplets, resulting in quenching of TTA. We evaluated the oxygen quenching rate to be $k_{O_2} = 1.13 * 10^5 \text{ s}^{-1}$ for the donor triplet (IR806), and $k_{O_2} = 0.9 * 10^3 \text{ s}^{-1}$ for the acceptor triplet (BTTQD **2**). As a result, the oxygen quenching efficiency η_{O_2} for the donor triplet can be estimated:

$$\eta_{O_2} = \frac{k_{O_2}}{(k_{O_2} + k_{TTET} + k_T^D)} \quad (\text{Eq R3.1})$$

in which $k_{TTET} = 5.56 * 10^5 \text{ s}^{-1}$ and $k_T^D = 2.19 * 10^5 \text{ s}^{-1}$. As a result, the oxygen quenching efficiency of the donor triplet was calculated to be $\eta_{O_2} = 12.7\%$.

Likewise, the oxygen quenching efficiency of the acceptor triplet can be evaluated using:

$$\eta_{O_2} = \frac{k_{O_2}}{(k_{O_2} + k_{TTA} + k_T^A)} \quad (\text{Eq R3.2})$$

in which $k_T^A = 2.34 * 10^4 \text{ s}^{-1}$ and k_{TTA} is dependent on the excitation light irradiance. At an extreme case where $k_{TTA} = 0$ such as TTA process at extreme low light irradiance where the first order process is significantly higher than the second order process, the maximum triplet quenching efficiency was evaluated to be $\eta_{O_2} = 3.7\%$, clearly indicating that there is nearly none oxygen quenching for the acceptor triplets. Taken together, oxygen induced triplet loss is very limited, on the level of 13%, therefore resulting in very stable TTA-UC emission for our bicomponent TTA-UC system. This information has been added to the revised SI as Section 4.4.

The second reason for eigen oxygen immunity of our TTA system is due to structural stability against oxidation of singlet oxygen molecules, which is formed by triplets sensitization of the surrounding oxygen molecules in the ground state (albeit inefficient in our system). The small amount of produced singlet oxygen could possibly induce destructive damage to the molecule structure. The driving force for molecular triplets as electron donor or acceptor can be evaluate according to Rehm Weller's equation:³²

$$\Delta G_{ET} = \Delta G_{ox} - \Delta G_{red} - \Delta E_{00} + w \quad (\text{Eq. R3.3})$$

Where ΔG_{ET} is the electron transfer driving force, ΔG_{ox} and ΔG_{red} represent oxidation/reduction potentials of electron donor and acceptor, respectively, ΔE_{00} is the

excitation energy (calculated from the corresponding phosphorescence), and w is a constant, representing the difference in Coulomb energy between the products and the reactants. According to the formula, when the oxidation potential of the electron donor becomes higher, the negativity of ΔG_{ET} decreases, representing the weakening of the driving force, therefore a high stability against oxygen molecules.

We performed a cyclic voltammetry experiment on IR806. As shown in Figure R3.1, the ΔG_{red} value was measured as -0.41 V for IR806. We then indirectly evaluated oxidation potential by looking at the highest occupied molecular orbital (HOMO) and (lowest unoccupied molecular orbital) LUMO levels of IR806.³³⁻³⁴

$$E_{LUMO} = -e(\Delta G_{red} + 4.4) \quad (\text{Eq. R3.4})$$

$$E_{HOMO} = -e(\Delta G_{ox} + 4.4) \quad (\text{Eq. R3.5})$$

$$E_{HOMO} = E_{LUMO} - E_g \quad (\text{Eq. R3.6})$$

E_g was calculated to be 1.31eV from the position of the absorption boundary of the IR806 absorption spectrum. Therefore, the oxidation potential of IR806 was evaluated to be 0.9 V. On the other hand, the oxidation potential of BTTQD **2** was evaluated to be 1.4 V. Both values are significantly higher than that of rubrene (0.4 V)³⁵ and DPA (0.2 V)³⁶ emitter, thereby showing high stability against oxygen damage than rubrene emitter. This can be clearly seen through distinct solution color changes over time in aerated solutions (Figure 2b and 2d).

We believe the ultralow oxygen quenching efficiency of both the donor (less than 13%) and the acceptor (less than 4%) triplets, combined with their high oxidation potentials against singlet oxygen damage, endows our bicomponent TTA system with eigen oxygen immunity.

Figure R3.1. Cyclic voltammogram (CV) curves of IR806.

Action 3. We have included Figure R3.1 as Figure S24 in the revised manuscript. Moreover, we have added corresponding discussions in the main text and conclusions in the abstract and conclusion part.

Page 12, paragraph 2: “Triplets quenching by oxygen is the main cause that results in the TTA-UC quenching.²³ In our TTA-UC system, the oxygen quenching efficiencies (η_{O_2}) for the sensitizer and annihilator triplets were evaluated to be 12.7% and <3.7%, respectively (see **Section 4.4** in supporting information for details), clearly showing that the involved triplets in both sensitizer and annihilator are insensitive to surrounding oxygen molecules. Moreover, both IR806 molecules (0.9 V) and BTTQD 2 molecule (1.4 V) have high oxidation potentials, much higher than the well-established annihilator rubrene molecule (0.4 V) and DPA molecule (0.2 V).⁵⁵⁻⁵⁶ This means much higher structural stability of both IR806 and BTTQD 2 molecules against reactive oxygen species (ROS), such as singlet oxygen (1O_2), superoxide anion ($O_2^{\cdot-}$), and hydroxyl radical ($HO\cdot$), formed during triplet oxygen quenching, in good agreement with experimental observations in **Figure 2**. ROS is Albeit ROS might be very limited, these highly reactive chemicals could still possibly induce destructive damage to the molecule structures. According to Rehm Weller's equation, $\Delta G_{ET} = \Delta G_{ox} - \Delta G_{red} -$

$\Delta E_{00} + w$, where ΔG_{ET} is the electron transfer driving force (typically having negative values), ΔG_{ox} and ΔG_{red} represent oxidation/ reduction potentials of electron donor and acceptor, respectively, ΔE_{00} is the excitation energy (calculated from the corresponding phosphorescence), and w is a constant, representing the difference in Coulomb energy between the products and the reactants.⁵⁷ High oxidation potential implies smaller negative numbers, and therefore, less driving force for molecular as electron donor to oxygen molecules (see **Figure S24** and discussion in supporting information for more details).⁵⁸⁻⁵⁹ Taken together, both ultralow oxygen quenching efficiencies for IR 806 and BTTQD2 triplets as well as their high structural stabilities against oxygen damage endow the oxygen immune properties for the developed IR806-BTTQD TTA system.”

Abstract: “Indeed, both the sensitizer and the annihilator molecules were shown to have ultralow triplet oxygen quenching efficiencies ($\eta_{O_2} < 13\%$ for the sensitizer, $< 3.7\%$ for the annihilator), and have high oxidation potentials against oxygen damage, collectively endowing the bicomponent TTA system with eigen oxygen immunity.”

Conclusion: “The ultralow oxygen triplet quenching efficiency ($\eta_{O_2} = 12.7\%$ and 3.7% for IR806 and BTTQD2, respectively), and the high oxidation potentials of both sensitizer and annihilator molecules against oxygen damage confer oxygen immunity to the IR806-BTTQD2 TTA system.”

Reference

1. Wang, T.; Wang, S.; Liu, Z.; He, Z.; Yu, P.; Zhao, M.; Zhang, H.; Lu, L.; Wang, Z.; Wang, Z.; Zhang, W.; Fan, Y.; Sun, C.; Zhao, D.; Liu, W.; Bunzli, J. G.; Zhang, F., A hybrid erbium(III)-bacteriochlorin near-infrared probe for multiplexed biomedical imaging. *Nat Mater* **2021**, *20* (11), 1571–1578.
2. Cheng, Y. Y.; Fückel, B.; Khoury, T.; Clady, R. G. C. R.; Tayebjee, M. J. Y.; Ekins-Daukes, N. J.; Crossley, M. J.; Schmidt, T. W., Kinetic Analysis of Photochemical Upconversion by Triplet–Triplet Annihilation: Beyond Any Spin Statistical Limit. *The Journal of Physical Chemistry Letters* **2010**, *1* (12), 1795–1799.
3. Monguzzi, A.; Tubino, R.; Salamone, M. M.; Meinardi, F., Energy transfer enhancement by oxygen perturbation of spin-forbidden electronic transitions in aromatic systems. *Physical Review B* **2010**, *82* (12).
4. Monguzzi, A.; Mezyk, J.; Scotognella, F.; Tubino, R.; Meinardi, F., Upconversion-induced fluorescence in multicomponent systems: Steady-state excitation power threshold. *Physical Review B* **2008**, *78* (19).
5. Zhou, Y.; Castellano, F. N.; Schmidt, T. W.; Hanson, K., On the Quantum Yield of Photon Upconversion via Triplet–Triplet Annihilation. *ACS Energy Letters* **2020**, *5* (7), 2322–2326.
6. Ronchi, A.; Monguzzi, A., Sensitized triplet–triplet annihilation based photon upconversion in full organic and hybrid multicomponent systems. *Chemical Physics Reviews* **2022**, *3* (4).
7. Ronchi, A.; Capitani, C.; Pinchetti, V.; Gariano, G.; Zaffalon, M. L.; Meinardi, F.; Brovelli, S.; Monguzzi, A., High Photon Upconversion Efficiency with Hybrid Triplet Sensitizers by Ultrafast Hole-Routing in Electronic-Doped Nanocrystals. *Advanced materials* **2020**, *32* (37), e2002953.
8. Saenz, F.; Ronchi, A.; Mauri, M.; Vadrucci, R.; Meinardi, F.; Monguzzi, A.; Weder, C., Nanostructured Polymers Enable Stable and Efficient Low-Power Photon Upconversion. *Advanced Functional Materials* **2020**, *31* (1).
9. Chen, F.; Zhao, L.; Wang, X.; Yang, Q.; Li, W.; Tian, H.; Shao, S.; Wang, L.; Jing, X.; Wang, F., Novel boron- and sulfur-doped polycyclic aromatic hydrocarbon as multiple resonance emitter for ultrapure blue thermally activated delayed fluorescence polymers. *Science China Chemistry* **2021**, *64* (4), 547–551.
10. Li, C.; Liu, J.; Hong, Y.; Lin, R.; Liu, Z.; Chen, M.; Lam, J. W. Y.; Ning, G. H.; Zheng, X.; Qin, A.; Tang, B. Z., Click Synthesis Enabled Sulfur Atom Strategy for Polymerization-Enhanced and Two-Photon Photosensitization. *Angewandte Chemie* **2022**, *61* (21), e202202005.
11. Li, M.; Xie, W.; Cai, X.; Peng, X.; Liu, K.; Gu, Q.; Zhou, J.; Qiu, W.; Chen, Z.; Gan, Y.; Su, S. J., Molecular Engineering of Sulfur-Bridged Polycyclic Emitters Towards Tunable TADF and RTP Electroluminescence. *Angewandte Chemie* **2022**, *61* (35), e202209343.
12. Mattiello, S.; Mecca, S.; Ronchi, A.; Calascibetta, A.; Mattioli, G.; Pallini, F.; Meinardi, F.; Beverina, L.; Monguzzi, A., Diffusion-Free Intramolecular Triplet–

Triplet Annihilation in Engineered Conjugated Chromophores for Sensitized Photon Upconversion. *ACS Energy Letters* **2022**, *7* (8), 2435–2442.

13. Sun, W.; Ronchi, A.; Zhao, T.; Han, J.; Monguzzi, A.; Duan, P., Highly efficient photon upconversion based on triplet-triplet annihilation from bichromophoric annihilators. *Journal of Materials Chemistry C* **2021**, *9* (40), 14201–14208.

14. Parker, C.; Hatchard, C. J. P. o. t. C. S. o. L., Sensitised anti-stokes delayed fluorescence. ROYAL SOC CHEMISTRY THOMAS GRAHAM HOUSE, SCIENCE PARK, MILTON RD, CAMBRIDGE ...: 1962; pp 386-&.

15. Fückel, B.; Roberts, D. A.; Cheng, Y. Y.; Clady, R. G. C. R.; Piper, R. B.; Ekins-Daukes, N. J.; Crossley, M. J.; Schmidt, T. W., Singlet Oxygen Mediated Photochemical Upconversion of NIR Light. *The Journal of Physical Chemistry Letters* **2011**, *2* (9), 966–971.

16. Huang, L.; Wu, W.; Li, Y.; Huang, K.; Zeng, L.; Lin, W.; Han, G., Highly Effective Near-Infrared Activating Triplet-Triplet Annihilation Upconversion for Photoredox Catalysis. *Journal of the American Chemical Society* **2020**, *142* (43), 18460–18470.

17. Sasaki, Y.; Oshikawa, M.; Bharmoria, P.; Kouno, H.; Hayashi-Takagi, A.; Sato, M.; Ajioka, I.; Yanai, N.; Kimizuka, N., Near-Infrared Optogenetic Genome Engineering Based on Photon-Upconversion Hydrogels. *Angewandte Chemie International Edition* **2019**.

18. Wu, M.; Congreve, D. N.; Wilson, M. W. B.; Jean, J.; Geva, N.; Welborn, M.; Van Voorhis, T.; Bulović, V.; Bawendi, M. G.; Baldo, M. A., Solid-state infrared-to-visible upconversion sensitized by colloidal nanocrystals. *Nature Photonics* **2015**, *10* (1), 31–34.

19. Nienhaus, L.; Correa-Baena, J.-P.; Wiegold, S.; Einzinger, M.; Lin, T.-A.; Shulenberger, K. E.; Klein, N. D.; Wu, M.; Bulović, V.; Buonassisi, T.; Baldo, M. A.; Bawendi, M. G., Triplet-Sensitization by Lead Halide Perovskite Thin Films for Near-Infrared-to-Visible Upconversion. *ACS Energy Letters* **2019**, *4* (4), 888–895.

20. Huang, L.; Zhao, Y.; Zhang, H.; Huang, K.; Yang, J.; Han, G., Expanding Anti-Stokes Shifting in Triplet-Triplet Annihilation Upconversion for In Vivo Anticancer Prodrug Activation. *Angewandte Chemie* **2017**, *56* (46), 14400–14404.

21. Simoes, E. F.; da Silva, J. C.; Leitao, J. M., Carbon dots from tryptophan doped glucose for peroxy nitrite sensing. *Anal Chim Acta* **2014**, *852*, 174–80.

22. Zhang, J.; Zhen, X.; Zeng, J.; Pu, K., A Dual-Modal Molecular Probe for Near-Infrared Fluorescence and Photoacoustic Imaging of Peroxynitrite. *Anal Chem* **2018**, *90* (15), 9301–9307.

23. Ai, X.; Wang, Z.; Cheong, H.; Wang, Y.; Zhang, R.; Lin, J.; Zheng, Y.; Gao, M.; Xing, B., Multispectral optoacoustic imaging of dynamic redox correlation and pathophysiological progression utilizing upconversion nanoprobables. *Nat Commun* **2019**, *10* (1), 1087.

24. Askes, S. H. C.; Bonnet, S., Solving the oxygen sensitivity of sensitized photon upconversion in life science applications. *Nature Reviews Chemistry* **2018**, *2* (12), 437–452.

25. Wang, X. Y.; Wang, X.; Baryshnikov, G. V.; Valiev, R. R.; Fan, R. W.; Lu, S. T.; Agren, H.; Chen, G. Y., A hybrid molecular sensitizer for triplet fusion upconversion. *Chem Eng J* **2021**, *426*.

26. Ravetz, B. D.; Pun, A. B.; Churchill, E. M.; Congreve, D. N.; Rovis, T.; Campos, L. M., Photoredox catalysis using infrared light via triplet fusion upconversion. *Nature* **2019**, *565* (7739), 343–346.
27. Oushiki, D.; Kojima, H.; Terai, T.; Arita, M.; Hanaoka, K.; Urano, Y.; Nagano, T., Development and application of a near-infrared fluorescence probe for oxidative stress based on differential reactivity of linked cyanine dyes. *Journal of the American Chemical Society* **2010**, *132* (8), 2795–801.
28. Peng, J.; Samanta, A.; Zeng, X.; Han, S.; Wang, L.; Su, D.; Loong, D. T.; Kang, N. Y.; Park, S. J.; All, A. H.; Jiang, W.; Yuan, L.; Liu, X.; Chang, Y. T., Real-Time In Vivo Hepatotoxicity Monitoring through Chromophore-Conjugated Photon-Upconverting Nanoprobes. *Angewandte Chemie* **2017**, *56* (15), 4165–4169.
29. Chen, G.; Yang, C.; Prasad, P. N., Nanophotonics and nanochemistry: controlling the excitation dynamics for frequency up- and down-conversion in lanthanide-doped nanoparticles. *Acc Chem Res* **2013**, *46* (7), 1474–86.
30. Huang, X.-J.; Choi, Y.-K.; Im, H.-S.; Yarimaga, O.; Yoon, E.; Kim, H.-S., Aspartate Aminotransferase (AST/GOT) and Alanine Aminotransferase (ALT/GPT) Detection Techniques. *Sensors* **2006**, *6* (7), 756–782.
31. Li, X.; Tao, R. R.; Hong, L. J.; Cheng, J.; Jiang, Q.; Lu, Y. M.; Liao, M. H.; Ye, W. F.; Lu, N. N.; Han, F.; Hu, Y. Z.; Hu, Y. H., Visualizing peroxynitrite fluxes in endothelial cells reveals the dynamic progression of brain vascular injury. *Journal of the American Chemical Society* **2015**, *137* (38), 12296–303.
32. Finikova, O. S.; Chen, P.; Ou, Z.; Kadish, K. M.; Vinogradov, S. A., Dynamic Quenching of Porphyrin Triplet States by Two-Photon Absorbing Dyes: Towards Two-Photon-Enhanced Oxygen Nanosensors. *J Photochem Photobiol A Chem* **2008**, *198* (1), 75–84.
33. Liu, R.; Zhang, L.; Zhao, J.; Luo, Z.; Huang, Y.; Zhao, S., Aptamer and IR820 Dual-Functionalized Carbon Dots for Targeted Cancer Therapy against Hypoxic Tumors Based on an 808 nm Laser-Triggered Three-Pathway Strategy. *Advanced Therapeutics* **2018**, *1* (5).
34. Zhang, H.; Chen, Y.; Liang, M.; Xu, L.; Qi, S.; Chen, H.; Chen, X., Solid-phase synthesis of highly fluorescent nitrogen-doped carbon dots for sensitive and selective probing ferric ions in living cells. *Anal Chem* **2014**, *86* (19), 9846–52.
35. Uttiya, S.; Miozzo, L.; Fumagalli, E. M.; Bergantin, S.; Ruffo, R.; Parravicini, M.; Papagni, A.; Moret, M.; Sassella, A., Connecting molecule oxidation to single crystal structural and charge transport properties in rubrene derivatives. *J. Mater. Chem. C* **2014**, *2* (21), 4147–4155.
36. Tinker, L. A.; Bard, A. J., Electrochemistry in liquid sulfur dioxide. 1. Oxidation of thianthrene, phenothiazine, and 9,10-diphenylanthracene. *Journal of the American Chemical Society* **2002**, *101* (9), 2316–2319.
37. Kanofsky, J. R.; Sima, P. D., Structural and Environmental Requirements for Quenching of Singlet Oxygen by Cyanine Dyes †. *Photochemistry and Photobiology* **2007**, *71* (4), 361–368.
38. Renikuntla, B. R.; Rose, H. C.; Eldo, J.; Waggoner, A. S.; Armitage, B. A., Improved photostability and fluorescence properties through polyfluorination of a

cyanine dye. *Org Lett* **2004**, *6* (6), 909-12.

Editorial Note: Reviewers 4 and 5 were recruited to look over the responses to Reviewer 2 and have provided additional comments (see below).

REVIEWER COMMENTS

Reviewer #1 (Remarks to the Author):

The authors provide now the requested complete investigation of the effect of the oxygen presence on the kinetic of excited state recombination and energy transfer processes for the sensitizer and annihilator species employed. The data reported the performed analysis validate and fully support the conclusion provided in the first version of the manuscript.

The authors clarify also the origin of the upconversion quantum yield observed, pointing out its relation with intrinsic characteristics of the employed chromophores employed

In my opinion, only the discussion about the effect of the presence of the annihilator on the ISC process on the sensitizers is still unclear. Additional investigations are required, so I would remove this section from this paper, because there is no clear conclusion and it no the focus of this work.

In the main t text there are no figure showing nor the upconversion emission spectrum, nor its decay kinetics. I would like to see that in figure 1, not only in the Supplementary information file (Figs. S14, S15), to directly demonstrate to the reader the occurrence of TTA sensitized upconversion.

Minor revisions:

Caption of figure 1 - Pease indicate the excitation wavelength for the upconversion data in panel d

Figure 3 panel d - please check the curves labels (aerated/deareated)

As requested by the Editor, I also considered the authors' answers to the point raised by Reviewer #2. About technical comments, it looks like the authors provided all the details and additional information on material composition structure and performance requested by the Reviewer.

Nevertheless, Reviewer 2# expressed also a big concern about the use of the proposed TTA-UC system for in vivo applications, highlighting the fact that the authors did not explain clearly why the TTA-UC should be used instead of traditional materials use to sense radicals in biological systems.

I partially agree with him, but I feel that the in vivo application is not the focus of the manuscript, as clearly indicated also by the title. In my opinion the most relevant result is the resistance against oxygen, which is still a crucial problem for the real application of any TTA-UC pairs and especially for NIR activated upconverters. By addressing his technical comments, the authors provide evidences of even better oxygen resistance in time and the biological system, a particular harsh environment in this sense, so further supporting their results.

Therefore, upon the further revision that I requested above, I can suggest to consider the manuscript for

publication.

Reviewer #3 (Remarks to the Author):

The revision is satisfactory, I recommend acceptance of the manuscript in its current form.

Reviewer #4 (Remarks to the Author):

The authors design an upconversion system to survive in an aerated environment. I agree with reviewers 1 and 3 that this is ultimately a crucial area of improvement for the entire TTA field. The system described here by the authors is potentially very exciting, however I have substantial concerns about the data and mechanism that prevents me from recommending publication at this time.

First, from a mechanism perspective, it seems that the innovation is the development of durable molecules, as the degradation differences play out over hours and days rather than seconds. This is an important development, to be sure, but is not sufficiently discussed and modeled. The authors rely on the limited change of triplet lifetime to argue for their improved mechanism, but really that's to be expected – a millisecond triplet system will be more greatly affected by the environment than a microsecond one, but will also have much greater efficiencies, both of which the authors observe.

I strongly suggest the authors do not add the factor of two to their efficiencies, as this has become the standard in the field. If they insist, they must include in the main text the fact that the factor of two was added.

The study of the degradation, or lack of it, in these upconversion systems is underdone. More clear comparisons should be provided. How do the efficiencies of the two systems (say as an example, rubrene vs. the dimers) compare in air-free conditions? How do they compare in initial brightness before the decay? Normalizing Fig 2 removes potentially useful information. This is especially crucial, as there are two separate mechanisms for oxygen damaging TTA as discussed above – interference with the TTA process (as discussed extensively here by the authors) and degradation of annihilator molecules themselves. These differences need to be understood, and given the days timeline, I suspect it is the latter. Absorption measurements can make more clear what is occurring.

I find it odd that none of the systems improve upon degassing – I would expect at least the DPA system would show improved efficiencies. To that end, have the authors measured the efficiency of their system under inert conditions?

What effect does the dimerization have? Does the monomer have the same properties?

PbS thin film upconversion is an inappropriate comparison for thresholds, as it only absorbs a small percentage of the light. A better control would be rubrene with the described sensitizer – what is its threshold? Similarly, the rubrene system would be the best reference for comparison, rather than DPA.

I find it odd that ISC depends on the annihilator concentration. Further evidence for this should be provided.

The DPA system is typically much higher than 16% (out of 100%), so the authors should carefully benchmark their efficiency statements.

Did the upconversion system maintain its efficiency in the micelles? From the absorption spectrum, I suspect the efficiency is much lower due to aggregation.

I'm a bit confused by the efficiency calculations. The overall efficiency should be the multiple of ISC, TET, TTA, and fluorescence and gives 31% by my calculation, almost an order of magnitude higher than the 4% reported. What's going on?

Minor: There are several typos in the added text, including pg 3: unraveled instead of untraveled

Lanthanide upconverters have virtually zero background; I'm not sure why they're included on the list of failed use cases.

Reviewer #5 (Remarks to the Author):

In this article, the authors present a class of near-infrared (NIR) TTA upconversion, characterized by its remarkable oxygen immunity. The approach utilizes non-organometallic IR806 molecules as sensitizers ($\lambda_{\text{ex}}=808$ nm) and three meticulously synthesized BTTQD dyes with a well-defined dimer structure as annihilators ($\lambda_{\text{em}}=650$ nm). This TTA system exhibits the ability to selectively detect ONOO⁻ and address related hepatotoxicity *in vivo*. While I concur with the comments made by reviewer #2. The concept of near-infrared (NIR) TTA is not new, and some strategies of breaking oxygen quenching were also developed. I recognize that the authors' contributions may not drastically revolutionize the landscape of *in vivo* applications. Therefore, I think the novelty and current scope studies do not support the manuscript published in Nature Communications together with the weak points below.

1. For Figure 4c, what's the PDI value?
2. The concentration of toxicity experiments was only conducted up to 400 $\mu\text{g}/\text{mL}$, whereas in the cytophagocytosis experiment, the concentration exceeded 400 $\mu\text{g}/\text{mL}$, indicating that the concentration range of the toxicity experiments was insufficient.
3. The results shown in Figure S34 indicate that the limit of detection is 9 nM in the solution. However, in Table S7, the authors state that this result pertains to *in vivo* data.
4. How to calculate the encapsulation efficiency should be listed in the experimental section.

5. In Figure S38, there was wrong spelling.
6. In Experiment section 3.6, what's "NaNO"?

Response to the comments

Reviewer #1:

The authors provide now the requested complete investigation of the effect of the oxygen presence on the kinetic of excited state recombination and energy transfer processes for the sensitizer and annihilator species employed. The data reported the performed analysis validate and fully support the conclusion provided in the first version of the manuscript.

The authors clarify also the origin of the upconversion quantum yield observed, pointing out its relation with intrinsic characteristics of the employed chromophores employed

In my opinion, only the discussion about the effect of the presence of the annihilator on the ISC process on the sensitizers is still unclear. Additional investigations are required, so I would remove this section from this paper, because there is no clear conclusion and it is not the focus of this work.

In the main text there are no figure showing nor the upconversion emission spectrum, nor its decay kinetics. I would like to see that in figure 1, not only in the Supplementary information file (Figs. S14, S15), to directly demonstrate to the reader the occurrence of TTA sensitized upconversion.

Response: We are grateful for the reviewer's acknowledgment that the concerns, both his/or hers and those of reviewer 2, have been successfully addressed in the revised manuscript.

We concur with the suggested deletion of the discussion about the effect of the annihilator on the ISC of the IR806 sensitizer, recognizing the necessity for further investigations. This adjustment has been implemented as suggested.

In addition, per the suggestion, we have not only combined Figs. S14 and S15 as Figure 1e, but also included the measured corresponding TTA-UC spectra as Figure 1d. The revised Figure 1 (**Figure R1** below) now clearly illustrates the occurrence of TTA upconversion.

Figure R1. (a) Chemical structures of IR806 and BTTQD 1-3. BTTQD 1 contains a phenyl linker and thiadiazole end cap, while BTTQD 2 contains a biphenyl bridge and thiadiazole end cap, contrasted with BTTQD 3 contains a biphenyl bridge and naphthalene end cap. (b) Schematic illustration of involved TTA-UC processes in IR806/BTTQD 1-3 systems. (c) Normalized absorption and emission spectra of IR806 and BTTQD 1-3 in chloroform. (d) Normalized upconversion emission spectra of IR806-BTTQD 1-3 solution (in chloroform) with corresponding photographic images shown in the inset ($\lambda_{\text{ex}}=808 \text{ nm}$, $10 \text{ W}/\text{cm}^2$). (e) Time-resolved luminescence decay of TTA-UC emission of IR806/BTTQD 2 at 650 nm in deaerated (top) and aerated (below) solution under low and high power excitations, with corresponding rising profiles shown on the right. low power: 0.02 mJ per pulse; high power: 1 mJ per pulse, pulse width: 25 kHz. (f) Logarithmic plots of upconversion emission intensities against

excitation power densities for IR806-BTTQD 1-3 in chloroform. In (c-f), $C_{IR806} = 1 \times 10^{-5} \text{ M}$, $C_{BTTQD 1-3} = 1 \times 10^{-4} \text{ M}$.

Action: We have included Figure R1 as revised Figure 1, and added the corresponding discussion in the revised manuscript on page 6, paragraph 1 as follows:

“At low excitation power, the TTA-UC intensity decays exponentially with time, with a characteristic decay time $\tau_{UC} = 21.4$ and $20.6 \mu\text{s}$ for deaerated and aerated sample, respectively. In this case, the primary depopulation channel of the annihilator triplet is the spontaneous radiative/non-radiative decay (negligible TTA), and the triplet lifetime of the annihilator could be estimated to be twice τ_{UC} ($2\tau_{UC} = \tau_T^A$), giving a τ_T^A of $\sim 40 \mu\text{s}$.⁴²⁻⁴⁴ At high excitation power, elevated concentration of annihilator triplets induce efficient TTA process, resulting in a shortened TTA-UC lifetime, as shown in **Figure 1e**.⁴⁵”

Minor revisions:

Caption of figure 1 - Please indicate the excitation wavelength for the upconversion data in panel d

Response: The excitation wavelength is 808 nm, which has been indicated in both the figure and the caption.

Figure 3 panel d - please check the curves labels (aerated/deaerated)

Response: The marked curve labels are correct.

As requested by the Editor, I also considered the authors' answers to the point raised by Reviewer #2.

About technical comments, it looks like the authors provided all the details and additional information on material composition structure and performance requested by the Reviewer.

Nevertheless, Reviewer 2# expressed also a big concern about the use of the proposed TTA-UC system for in vivo applications, highlighting the fact that the authors did not explain clearly why the TTA-UC should be used instead of traditional materials use to

sense radicals in biological systems.

I partially agree with him, but I feel that the in vivo application is not the focus of the manuscript, as clearly indicated also by the title. In my opinion the most relevant result is the resistance against oxygen, which is still a crucial problem for the real application of any TTA-UC pairs and especially for NIR activated upconverters. By addressing his technical comments, the authors provide evidences of even better oxygen resistance in time and the biological system, a particular harsh environment in this sense, so further supporting their results.

Therefore, upon the further revision that I requested above, I can suggest to consider the manuscript for publication.

Response: We value the constructive comments from reviewer 1 and his/or her support for the publication of this work. As correctly grasped by reviewer 1, the primary objective of this study is to introduce an innovative oxygen-immune TTA upconversion system with potential broad applications spanning from biophotonics to photonics. In vivo detection of peroxynitrite radical in nitrosative hepatotoxicity was provided as an example to showcase one of these potential applications.

Reviewer #3 (Remarks to the Author):

The revision is satisfactory, I recommend acceptance of the manuscript in its current form.

Response: We are grateful for the positive response from reviewer 3 regarding our revision and the kind recommendation for the publication of this work.

Reviewer #4:

The authors design an upconversion system to survive in an aerated environment. I agree with reviewers 1 and 3 that this is ultimately a crucial area of improvement for the entire TTA field. The system described here by the authors is potentially very exciting, however I have substantial concerns about the data and mechanism that prevents me from recommending publication at this time.

Response: We express our gratitude to the reviewer for acknowledging the potential excitement of this work for the entire TTA field and for providing crucial comments that have significantly enhanced the quality of this manuscript. To address the substantial concerns raised by the reviewer, we have conducted additional experiments and provided clarifications, as detailed in our point-to-point response to the specific comments outlined below. We trust that the revised manuscript now meets the expectations of the reviewer.

1. First, from a mechanism perspective, it seems that the innovation is the development of durable molecules, as the degradation differences play out over hours and days rather than seconds. This is an important development, to be sure, but is not sufficiently discussed and modeled. The authors rely on the limited change of triplet lifetime to argue for their improved mechanism, but really that's to be expected – a millisecond triplet system will be more greatly affected by the environment than a microsecond one, but will also have much greater efficiencies, both of which the authors observe.

Response: We appreciate the reviewer's recognition of the significance of our work in the successful development of the oxygen-immune TTA upconversion system, as well as the insightful comments on the underlying mechanisms. We agree with the reviewer that triplets with a millisecond lifetime are generally more sensitive to the environment. Triplets with longer lifetimes have higher probabilities of interacting with diffused oxygen molecules, potentially leading to the production of highly-reactive singlet oxygen molecules (as seen in Type-II photodynamic therapy) or engaging in direct electron transfer to surrounding substrate molecules (such as oxygen or water),

resulting in the formation of highly-reactive radicals like hydroxyl radicals ($\bullet\text{OH}$) and superoxide (O^{2-}) ions (as in Type-I photodynamic therapy).

Nevertheless, the lifetime of triplets is not the sole factor determining the stability and efficiency of the TTA system. Triplets with short lifetimes can also experience efficient oxygen quenching. For instance, triplets of thio-pentamethine cyanine dye with a microsecond-range lifetime ($\sim 319 \mu\text{s}$) were reported to exhibit a singlet oxygen generation quantum yield of 99%, translating to a 99% oxygen quenching efficiency (Chem. Sci., 2021, 12, 13809-13816). Instead, we believe that the stability of the TTA system crucially relies on both the quantum efficiency of triplet quenching in an aerated environment and the structural stability of both TTA pairs against reactive oxygen species (ROS), such as singlet oxygen or free radicals, generated during triplet quenching. This consideration motivated us to conduct a series of relevant experiments to explore the underlying mechanism of the oxygen immunity of our TTA system.

2. I strongly suggest the authors do not add the factor of two to their efficiencies, as this has become the standard in the field. If they insist, they must include in the main text the fact that the factor of two was added.

Response: In response to the reviewer's suggestion, we have ceased using two-fold multiplication in the definition of upconversion quantum yield (UCQY) in the revised manuscript. The theoretical maximum of UCQY is now set at 50%, aligning with the standard in the field.

3. The study of the degradation, or lack of it, in these upconversion systems is underdone. More clear comparisons should be provided. How do the efficiencies of the two systems (say as an example, rubrene vs. the dimers) compare in air-free conditions? How do they compare in initial brightness before the decay? Normalizing Fig 2 removes potentially useful information. This is especially crucial, as there are two separate mechanisms for oxygen damaging TTA as discussed above – interference with the TTA process (as discussed extensively here by the authors) and degradation of annihilator molecules themselves. These differences need to be

understood, and given the days timeline, I suspect it is the latter. Absorption measurements can make more clear what is occurring.

Response: We appreciate the reviewer's valuable comments. In accordance with the suggestions, we conducted a comprehensive comparison of the degradation of the developed IR806-BTTQD upconversion systems against the controls. In the revised Figure 2 (**Figure R2** below), we have reversed the intensity normalization to illustrate the time-dependent changes in upconversion intensities for IR806-BTTQD **1**, IR806-BTTQD **2**, and IR806-BTTQD **3**, in comparison to the controls of IR806-rubrene and PtOEP-DPA. This comparison was conducted during extended periods of storage in air (up to 7 days) and under prolonged laser irradiance (up to 480 mins) in both aerated and deaerated solutions.

Under air-free conditions (or before decay), the IR806-BTTQD **2** system exhibits greater brightness than the IR806-rubrene control (NIR excitation) but is less bright than the PtOEP-DPA control (visible excitation). Remarkably, the brightness of the IR806-BTTQD **2** system remains nearly constant after a 7-day storage in air and under continuous-wave laser irradiation at 808 nm for over 480 mins in air. This is in stark contrast to both the IR806-rubrene and PtOEP-DPA controls, which were completely quenched after a 2-day storage or 200 mins of laser irradiation in air. These results distinctly showcase the oxygen immunity of TTA-UC in the developed IR806-BTTQD**2** system, as opposed to the vulnerability observed in the IR806-rubrene and PtOEP-DPA controls.

Figure R2. (a) The changes of upconversion emission intensities of IR806-BTTQD 1-3, IR806-rubrene, and PtOEP-DPA solution during 7 days storage in air. (b) Photographic images of IR806-BTTQD 2 and IR806-rubrene solution over the 7-day storage in air. (c) Emission stabilities of upconversion in IR806-BTTQD 2, IR806-rubrene, and PtOEP-DPA solution with/without degassing under continuous-wave laser irradiation at 808 or 532 nm. (d) Photographic images of TTA upconversion emission in (c) taken at varied time in aerated solution. IR806-BTTQD 1-3 and IR806-rubrene were dissolved in chloroform, while PtOEP-DPA was dissolved in tetrahydrofuran with concentrations of $c_{\text{IR806}} = 1 \times 10^{-5} \text{ M}$, $c_{\text{BTTQD}} = 1 \times 10^{-4} \text{ M}$, $c_{\text{Rubrene}} = 1 \times 10^{-4} \text{ M}$, $c_{\text{PtOEP}} = 1 \times 10^{-4} \text{ M}$, $c_{\text{DPA}} = \times 10^{-2} \text{ M}$. For IR806-BTTQD 1-3 and IR806-rubrene, $\lambda_{\text{ex}}=808 \text{ nm}$, power density= 100 mW/cm^2 , while for PtOEP-DPA, $\lambda_{\text{ex}}=532 \text{ nm}$, power density= 5 mW/cm^2 .

To further elucidate the oxygen immunity of IR806-BTTQD **2**, we assessed the change in its upconversion quantum yield (UCQY) compared to that of the IR806-rubrene control during prolonged exposure to 808 nm laser (long-term TTA-UC) in both aerated and deaerated conditions (see **Figure R3**). Additionally, absorbance measurements were taken for both IR806-BTTQD2 and the IR806-rubrene control to unveil the structural stabilities of the involved molecules (refer to **Figure R4**).

The UCQY of IR806-BTTQD2 remained consistent at approximately 1.7% before laser overexposure in both aerated and deaerated solutions. Subsequently, it displayed only a marginal 10% decrease after 48 hours of laser irradiation in both aerated and deaerated solutions. In stark contrast, the UCQY of the IR806-rubrene control exhibited variations in aerated (0.2%) and deaerated (0.3%) solutions before laser overexposure, indicating significant oxygen-induced triplet quenching. Furthermore, the UCQY was observed to plummet close to zero (nearly a 100% decrease) in the aerated solution but remained relatively stable (with only a 23% decrease) in the deaerated solution after 48 hours of laser irradiation. This distinct trend suggests the instability of the molecular structure of the TTA pair in the aerated IR806-rubrene solution during the TTA-UC process.

Figure R3. Time-dependent UCQYs of IR806-BTTQD **2** (in chloroform) and IR806-rubrene (in toluene) systems in the presence (a) and absence (b) of oxygen under continuous laser irradiation at 808 nm (100 mW/cm^2). UCQYs were measured at power density of 10 W/cm^2 with $c_{\text{IR806}} = 1 \times 10^{-5} \text{ M}$, $c_{\text{Rubrene/BTTQD 2}} = 1 \times 10^{-4} \text{ M}$.

Figure R4. Absorption spectra of (a, c) IR806-rubrene (in toluene) and (b, d) IR806-BTTQD 2 (in chloroform) in aerated (a, b) and deaerated (c, d) solution during long-time laser irradiation ($\lambda_{\text{ex}} = 808 \text{ nm}$, 100 mW/cm^2). $c_{\text{IR806}} = 1 \times 10^{-5} \text{ M}$, $c_{\text{Rubrene/BTTQD 2}} = 1 \times 10^{-4} \text{ M}$.

Indeed, a conspicuous and rapid decrease in the absorption peak of rubrene ($\sim 500 \text{ nm}$), contrasting with the unchanged absorption peak of IR806 ($\sim 800 \text{ nm}$), was observed in the aerated IR806-rubrene solution during long-term TTA-UC (refer to **Figure R4a**). In contrast, the absorption peaks of both rubrene and IR806 remained nearly constant in the deaerated solution (see **Figure R4c**). As for the developed IR806-BTTQD2 system, the absorption peaks of both IR806 and BTTQD2 ($\sim 580 \text{ nm}$) remained consistent in both aerated and deaerated solutions during long-term TTA-UC (**Figure R4b, d**). These findings indicate that surrounding oxygen molecules can compromise the structure of rubrene but do not impact the structures of the synthesized BTTQD2 emitter and IR806 sensitizer. This observation is reasonable as rubrene has a low oxidation potential ($\sim 0.4 \text{ V}$), while both BTTQD 2 ($\sim 1.4 \text{ V}$) and IR806 (0.9 V) possess higher oxidation potentials, demonstrating greater structural stability against

the reactive oxygen species (ROS) produced during TTA-UC.

In summary, oxygen molecules induce significant triplet quenching and molecular structure instability (rubrene) of TTA pairs during TTA-UC in the IR806-rubrene control. In contrast, the effects on both triplet quenching and molecular structure stability in the developed IR806-BTTQD2 system are limited or negligible. These results provide additional substantiation for our conclusions regarding the mechanism underlying the oxygen immunity.

Action: We have reverted the normalization of intensities in both Figure 2(a) and (c) and introduced IR806-rubrene as a new control in both Figure 2(c) and (d). Additionally, we have incorporated Figure R3 as Figure S19 and Figure R4 as Figure S23 in the revised SI. Corresponding discussions have been included in the revised manuscript.

Page 9, paragraph 3: “To further illustrate the phenomenon of oxygen immunity, we examined the change of Φ_{UC} for both IR806-BTTQD 2 and the IR806-rubrene control in **Figure 2c** (**Figure S19**). The Φ_{UC} of IR806-BTTQD 2 remained consistent at approximately 1.7% before laser overexposure in both aerated and deaerated solutions. Subsequently, it displayed only a marginal 10% decrease after 48 hours of laser irradiation in both aerated and deaerated solutions, demonstrating remarkable stability. This consistency aligns with the observed TTA-UC intensity change in **Figure 2c**. Conversely, the Φ_{UC} of the IR806-rubrene control showed variation in aerated (0.2%) and deaerated (0.3%) solutions before laser overexposure, indicating pronounced oxygen-induced triplet quenching. Furthermore, the Φ_{UC} was observed to plummet to near-zero (close to a 100% decrease) in the aerated solution but remained relatively stable (with only a 23% decrease) in the deaerated solution after 48 hours of laser irradiation. This distinctive trend suggests the instability of the molecular structure of the TTA pair in aerated IR806-rubrene solution during the TTA-UC process. In conclusion, oxygen molecules exert limited (or negligible) effects on Φ_{UC} and molecular structure stability in the developed IR806-BTTQD 2 system.”

Page 14, paragraph 2: “An additional factor potentially contributing to TTA-UC quenching is the structural instability of TTA molecule pairs. In this context, the transfer of triplet energy to surrounding oxygen molecules might generate reactive oxygen

species (ROS), such as singlet oxygen, capable of damaging the molecular structure of TTA pairs. This has been implied from distinct Φ_{UC} change trend in aerated and deaerated IR806-rubrene solutions (**Figure 2c**). To investigate this possibility, we also examined the corresponding absorption spectra of TTA pairs shown in **Figure 2c**. For the developed IR806-BTTQD **2** system, the absorption peaks of both IR806 and BTTQD **2** remained nearly constant in both aerated and deaerated solutions during long-term TTA-UC (**Figure S23**). In contrast, an evident and rapid decrease in the rubrene absorption peak, contrasted with the unchanged absorption peak of IR806, was observed in aerated IR806-rubrene solution (**Figure S23a**). Furthermore, absorption peaks of both rubrene and IR806 remained nearly identical in deaerated solution (**Figure S23c**). These results reveal that ROS produced during triplet depletion can impair the structure of rubrene,²³ but are unable to compromise the structures of synthesized BTTQD **2** emitter and IR806 sensitizer. This is reasonable, as rubrene has a low oxidation potential (~ 0.4 V), while both BTTQD **2** (~ 1.4 V) and IR806 (0.9 V) (**Figure S24**) have higher oxidation potential,⁵⁶⁻⁵⁸ thereby showing higher structural stability against produced ROS during TTA-UC.”

4. I find it odd that none of the systems improve upon degassing – I would expect at least the DPA system would show improved efficiencies. To that end, have the authors measured the efficiency of their system under inert conditions?

Response: Upon degassing, TTA-UC brightness of IR806-BTTQD **2** remained almost unchanged, but the brightness of PtOEP-DPA and IR806-rubrene were indeed increased (see revised **Figure 2**). This can also be clearly seen in measured UCQYs of IR806-BTTQD2, versus the controls of PtOEP-DPA and IR806-rubrene, in aerated and deaerated solutions (**Table R1**).

Table R1. UCQYs of IR806-BTTQD **2** (in chloroform), IR806-rubrene (in toluene), and PtOEP-DPA (in THF) systems in aerated and deaerated solutions. For IR806-BTTQD **1-3** and IR806-rubrene, $\lambda_{ex}=808$ nm, power density = 10 W/cm², while for

PtOEP-DPA, $\lambda_{\text{ex}}=532$ nm, power density = 5 mW/cm².

	IR806-BTTQD 2	IR806-rubrene	PtOEP-DPA
Deaerated	1.75%	0.31%	21.2%
Aerated	1.73%	0.23%	3.9%

5. What effect does the dimerization have? Does the monomer have the same properties?

Response: The dimer exhibits similar but red-shifted absorption and emission properties, characteristic of an extended π system, and possess similar excited energy level structures of the monomer. The monomer is also available as an emitter for TTA upconversion (**Figure R5** and **Table R2**). However, it was determined that the obtained upconversion quantum yield (UCQY) and threshold were approximately two-fold lower and higher, respectively, than the ones when using the dimer as the emitter.

Table R2. Calculated S₁/T₁ energies, and PLQY/UCQY of monomer and BTTQD 2 annihilators. $\lambda_{\text{ex}}=808$ nm, power density= 10 W/cm² for UCQY measurement; $\lambda_{\text{ex}}=450$ nm, power density= 5 mW/cm² for PLQY measurement.

	S ₁ (eV)	T ₁ (eV)	PLQY (%)	UCQY (%)
Monomer	2.14	1.21	87	1.06
BTTQD 2	2.09	1.17	63	1.73

Figure R5. (a) Normalized absorption and photoluminescence ($\lambda_{\text{ex}} = 450 \text{ nm}$) spectra of BTTQD 2 and its corresponding monomer. (b) Normalized upconverted emission spectra of solutions containing IR806 as sensitizer and monomer/BTTQD 2 as annihilators (IR806: $10 \mu\text{M}$, monomer or BTTQD 2: $200 \mu\text{M}$). (c) TTA-UC emission intensities as a function of the annihilator concentration with fixed IR806 concentration of $10 \mu\text{M}$. (d) Logarithmic plots of upconversion emission intensities against excitation power densities for IR806-monomer and IR806-BTTQD 2 in chloroform, respectively (IR806: $10 \mu\text{M}$, monomer or BTTQD 2: $200 \mu\text{M}$).

Action: We have added Figure R5 as Figure S17 and the energy level structure of the monomer in Table S4 in the revised SI. We also added the following discussions in the revised manuscript on page 7 and paragraph 1:

“We would like to emphasize that TTA-UC is also feasible using the monomer of BTTQD 2, which exhibits comparable singlet and triplet energy levels (Table S4). However, it results in a two-fold lower Φ_{UC} of approximately 1.06% and a two-fold higher TTA threshold of around 4 W/cm^2 (Figure S17). This rationalizes the preference

for utilizing the BTTQD 2 dimer as the annihilator, possibly attributed to the dimer having two simultaneously available triplet groups under photoexcitation.²⁹”

6. PbS thin film upconversion is an inappropriate comparison for thresholds, as it only absorbs a small percentage of the light. A better control would be rubrene with the described sensitizer – what is its threshold? Similarly, the rubrene system would be the best reference for comparison, rather than DPA.

Response: We agree with the reviewer that PbS thin film upconversion is inappropriate as a reference for the threshold comparison. Per the suggestion, IR806-rubrene has been adopted as a control reference in the revised manuscript, with a two-fold higher power density threshold of 3.6 W/cm² than that of IR806-BTTQD 2 (see revised Figure 2).

7. I find it odd that ISC depends on the annihilator concentration. Further evidence for this should be provided.

Response: We thank reviewer 4 for noting this, which was also echoed by reviewer 1. We considered this could be possibly ascribed to the heavy atom effect of sulfur in the annihilator, but we acknowledge that this warrants further detailed investigations. In accordance with the suggestion of reviewer 1, we have removed relevant discussions on this phenomenon from the revised manuscript. It's important to note that the removal of this discussion does not impact the conclusions drawn in our work.

8. The DPA system is typically much higher than 16% (out of 100%), so the authors should carefully benchmark their efficiency statements.

Response: We experimentally determined the UCQY of the PtOEP-DPA system to be 21% (out of 50%) in the degassed solution. More details can be found in our response to the comment 4.

9. Did the upconversion system maintain its efficiency in the micelles? From the absorption spectrum, I suspect the efficiency is much lower due to aggregation.

Response: The TTA upconversion efficiency in the micelles was determined to be

0.44%, about 4-fold lower than that of the organic phase (Ex=808 nm, 10 W/cm²) (Figure R6). But we would like to note that TTA upconversion in TTA-NMs is still sufficient for high contrast in vivo imaging experiments.

Figure R6. UCQY of TTA-NMs in PBS as a function of the incident power density under 808 nm excitation.

Action: We have included Figure R6 as Figure S26 in the revised SI, and added the corresponding comments in the revised manuscript on page 15, paragraph 1 as follows:

“The maximum Φ_{UC} of TTA-NMs was determined to be about 0.44%, about three-fold lower than that of the organic phase (Figure S26).”

10. I’m a bit confused by the efficiency calculations. The overall efficiency should be the multiple of ISC, TET, TTA, and fluorescence and gives 31% by my calculation, almost an order of magnitude higher than the 4% reported. What’s going on?

Response: The TTA-UC efficiency with a theoretical maximum of 50% can be calculated using equation 13 in the revised SI:

$$\Phi_{UC} = 0.5f\Phi_{ISC}\Phi_{TTET}\Phi_{TTA}\Phi_F$$

Previously, we measured $\Phi_{ISC} = 72.2\%$, $\Phi_{TTET} = 60.8\%$, $\Phi_F = 62.7\%$, $\Phi_{TTA} = 95.3\%$. If assuming the spin-statistic $f = 2/5$, this will result in $\sim 5\%$ UCQY.

We note that Φ_F of BTTQD 2 strongly depends on its concentration in solution (Figure R7), possibly due to aggregation-induced fluorescence quenching. At the optimized annihilator concentration $c_{\text{BTTQD 2}} = 1 \times 10^{-4}$ M for TTA-UC in IR806-BTTQD 2, Φ_F was actually determined to be $\sim 27.0\%$. Correcting $\Phi_F = 62.7\%$ into $\Phi_F = 27.0\%$, the UCQY could be theoretically estimated to be $\sim 2.3\%$, very close to the experimentally determined one of 2%.

Figure R7. PLQY of BTTQD 2 annihilator as a function of its concentration in chloroform.

Action: We have included Figure R7 as Figure S16 in the revised SI. Moreover, we have added the corresponding discussions in the revised manuscript and SI.

Page 7, paragraph 1: “Note that PLQY of BTTQD 2 was decreased about twice ($\sim 27.0\%$) at $c_{\text{BTTQD 2}} = 1 \times 10^{-4}$ M, compared to that of the diluted solution (Figure S16). Further approaches to maintain annihilator PLQYs at high concentrations can further increase Φ_{UC} .”

Section 5.1 of SI: “For TTA-UC, the upconversion quantum yield Φ_{UC} can be determined using equation (13) with a theoretical maximum of 50%.

$$\Phi_{\text{UC}} = 0.5f\Phi_{\text{ISC}}\Phi_{\text{TTET}}\Phi_{\text{TTA}}\Phi_{\text{F}} \quad (13)$$

where Φ_{ISC} , Φ_{TTET} , Φ_{TTA} , Φ_F are the quantum efficiencies of the intersystem crossing of the sensitizer, triplet-to-triplet energy transfer between the sensitizer and annihilator, triplet-triplet annihilation (TTA) process of the annihilator, and singlet fluorescence of the annihilator, respectively. The f represents the spin-statistical factor, which illustrates the probability of two triplets annihilating to form a usable singlet. For most cases with $E_{T_2} > 2E_{T_1}$, the spin-statistical factor could be assumed to be $f = 2/5$. We determined $\Phi_{ISC} = 72.2\%$, $\Phi_{TTET} = 60.8\%$, $\Phi_F = 27.0\%$, $\Phi_{TTA} = 95.3\%$. Therefore, Φ_{UC} could be theoretically estimated to be $\sim 2.3\%$, very close to the experimentally determined one of 2%.”

Minor:

There are several typos in the added text, including pg 3: unraveled instead of untraveled

Response: The word “untraveled” has been corrected into “unraveled” in the revised manuscript (page 3, paragraph 2). We also carefully proofread the entire manuscript and corrected all identified typos.

Lanthanide upconverters have virtually zero background; I’m not sure why they’re included on the list of failed use cases.

Response: Indeed, lanthanide upconverters can also achieve zero background bioimaging, but they usually demand higher excitation power density (typically, $\sim 1,000 - 10,000 \text{ mW/cm}^2$), about one or two orders of magnitude higher than that of TTA-UC for bioimaging applications. That’s why lanthanide upconverter was listed as a reference for comparison.

To avoid possible confusion, we have removed lanthanide upconverters as a comparison in the revised manuscript.

Reviewer #5:

In this article, the authors present a class of near-infrared (NIR) TTA upconversion, characterized by its remarkable oxygen immunity. The approach utilizes non-organometallic IR806 molecules as sensitizers ($\lambda_{ex}=808$ nm) and three meticulously synthesized BTTQD dyes with a well-defined dimer structure as annihilators ($\lambda_{em}=650$ nm). This TTA system exhibits the ability to selectively detect ONOO⁻ and address related hepatotoxicity in vivo. While I concur with the comments made by reviewer #2. The concept of near-infrared (NIR) TTA is not new, and some strategies of breaking oxygen quenching were also developed. I recognize that the authors' contributions may not drastically revolutionize the landscape of in vivo applications. Therefore, I think the novelty and current scope studies do not support the manuscript published in Nature Communications together with the weak points below.

Response: We express gratitude for the reviewer's critical comments regarding the novelty and the current scope of this study. We believe there might be a misconception that we would like to address. We acknowledge the existence of current near-infrared (NIR)-excitable triplet–triplet annihilation (TTA) systems (see references 12, 14-21) and current strategies to overcome oxygen quenching (see references 24-28), as clearly highlighted in the introduction. However, it is crucial to emphasize that our work significantly diverges from the existing literature, and the primary focus of our study does not revolve around in vivo applications.

As correctly pointed out by reviewers 1, 3, and 4, the primary focus of this study is to present an eigen oxygen-immune triplet–triplet annihilation (TTA) system, accompanied by comprehensive mechanistic investigations that elucidate the observed oxygen immunity phenomena. The described TTA upconversion system here capitalizes on the inherent properties of TTA pairs, rendering them unsusceptible to surrounding oxygen molecules. This approach stands in contrast to prevailing strategies aimed at mitigating the deleterious effects of oxygen, such as encapsulating TTA pairs with physical barriers to resist oxygen or incorporating antioxidants in solution or within nanoparticles to efficiently deplete local ambient oxygen, as illustrated in **Figure R8** of a recent review (Nature Reviews Chemistry, 2018, 2, 437).

The outlined eigen oxygen-immune TTA system here holds direct implications for a myriad of real-world applications, spanning from photonics to biophotonics. In vivo detection of ONOO⁻ in hepatotoxicity was exemplified to showcase one of these implicated applications. We trust that this clarification will address the misunderstanding raised by reviewer 5.

[REDACTED]

Figure R8. Three general strategies to prevent TTA systems from being quenched by oxygen. (*Nature Reviews Chemistry*, 2018, 2, 437).

1. For Figure 4c, what's the PDI value?

Response: The PDI was determined to be 0.147, which has been included in revised Figure 4c (**Figure R9** below).

Figure R9. Dynamic light scattering (DLS) analysis of TTA-NMs.

2. The concentration of toxicity experiments was only conducted up to 400 $\mu\text{g/mL}$, whereas in the cytophagocytosis experiment, the concentration exceeded 400 $\mu\text{g/mL}$, indicating that the concentration range of the toxicity experiments was insufficient.

Response: We conducted a toxicity experiment on TTA-NMs over a broader concentration range (0 - 2 mg/mL) (see **Figure R10**). Even at concentrations as high as 2 mg/mL, the survival rate of HeLa cells remained consistently close to 90% after 24 hours of incubation.

Figure R10. Cell viability of HeLa cells treated with TTA-NMs at different concentrations (0 - 2 mg/mL).

Action: We have included Figure R10 as Figure S28 in the revised SI.

3. The results shown in Figure S34 indicate that the limit of detection is 9 nM in the solution. However, in Table S7, the authors state that this result pertains to in vivo data.

Response: We thank the reviewer for pinpointing this error in Table S7 (corresponding to **Table R3** below). LOD of all referenced literature and this work have been determined ex-vivo. We have marked LOD column "Ex-vivo" in Table S7 in the revised SI.

Table R3. Updated Table S7 in the revised supporting information.

Probe	Mode of sensing		Ex/Em	Response time	LOD (Ex-vivo)	Ref
tryptophan doped carbon dots	Turn - off	In vitro	370/486 nm	<100 s	1.5 μ M	Anal. Chim. Acta. 2014, 852, 174
CySO ₃ CF ₃ organic molecule	Turn-on	In vivo	680/710 nm	<3 min	53 nM	Anal. Chem. 2018, 90, 9301
Hf-Uio-66-B(OH) ₂ MOF	Turn-on	Ex vivo	330/426 nm	1 min	9 nM	Inorg. Chem. 2018, 57, 16, 10128
MBTBE organic molecule	Turn-on	Ex vivo	520/569 nm	1 min	16 nM	Sens. Actuators B Chem. 2020, 303, 127284
Ru-organic complex	Turn - off	In vitro	468/600 nm	10 s	-	Spectrochim. Acta, Part A. 2012, 94, 340
TTA-NMs (Upconversion)	Turn-on	In vivo	808/690 nm	< 1 min	9 nM	This work

Action: We have included Table R3 as Table S7 in the revised SI.

4. How to calculate the encapsulation efficiency should be listed in the experimental section.

Response: The encapsulation efficiency of TTA pairs (IR 806, BTTQD2) and Cy 7 in TTA-NMs has been determined by the ratio of the corresponding absorbance before and after encapsulation into TTA-NMs.

Action: Details of determination of encapsulation efficiency have been added as Section 2.8 in the revised SI.

“2.8 Evaluation of the encapsulation efficiency.

The encapsulation efficiency was determined by comparing the characteristic absorbance differences of IR806 and BTTQD2 before and after encapsulation in TTA-NMs. Initially, the absorbance of IR806 solution (10 μM) and BTTQD2 solution (100 μM) were determined. Subsequently, the absorbance of the TTA-NMs aqueous solution, obtained after repeated centrifugation and washing, was measured. As the absorbance spectra of TTA-NMs encompass the characteristic absorption bands of both IR806 and BTTQD2, the encapsulation efficiency of IR806 and BTTQD2 was estimated by the ratio of characteristic absorbance peak before and after encapsulation in TTA-NMs. The load efficiency of Cy7 onto the TTA-NMs surface can also be estimated using the same method.”

5. In Figure S38, there was wrong spelling.

Response: We have corrected the wrong spelling of “excitatiuon” into “excitation”.

6. In Experiment section 3.6, what’s “NaNO”?

Response: “NaNO” should be “NaNO₂”, which has been corrected in the revised SI.

REVIEWER COMMENTS

Reviewer #1 (Remarks to the Author):

I consider this revision satisfactory, the authors strongly improved the experimental part as required by several Reviewer to support their findings, including in the manuscript or in the Supporting Information file all the missing details and additional information needed to give a full picture of the proposed molecular system for NIR photon upconversion.

Before publication, I think the manuscript need a “graphical” revision. For example, in Figure 1 please take care of the details: avoid that labels overlap with the graphs’ axis, such as in panel d, etc. etc.

This is not only an aesthetic point, but more clean figures matching the journal standards, will help a lot the readers.

Reviewer #4 (Remarks to the Author):

I thank the authors for their updates; the paper is much improved and I'm supportive of publication following addressing of the below concerns:

The upconversion emission looks substantially different from the fluorescence. Why?

Minor:

The process is “triplet-triplet annihilation” not “triplet-to-triplet annihilation) pg 2. The process is further typically accepted as two annihilator triplets meeting, not a sensitizer and annihilator triplet.

Reviewer #5 (Remarks to the Author):

I have completed my review of the manuscript titled “[Manuscript Title]” and have several comments that I believe need to be addressed to improve the clarity and scientific rigor of the study.

1. Writing Quality: The manuscript could benefit from substantial revisions to improve clarity. There are numerous misleading statements, such as the authors mentioned that “This observation is in sharp contrast to the typically investigated standard PtOEP-DPA system, where oxygen molecules was observed to enhance TTET rate by orders of magnitude.”, typically, oxygen would reduce the TTET rate from

sensitizer to annihilator and quenching the upconversion emission.

In addition, in the animal study, the authors mentioned that "30 mins later, PBS solution of Cy7-coated TTA-NMs (5 mg/mL, 100 μ L) were injected into the tail veins of two Kunming mice, and dynamic upconversion emission imaging was performed (Figure 5b).", however, in Figure 5d, the temporal changes of upconversion emission signals, the author mentioned n=3 for each group. According to my understand, the analysis of temporal changes of upconversion emission should base on the upconversion imaging. How many mice was used for the animal study? All of these could lead to confusion.

2. Stability Comparisons: The authors claim exceptional stability for their system; however, the comparisons are made under varying conditions like concentration and power. These differing conditions can significantly affect the results, rendering the data unconvincing as presented.

3. Photostability: While the authors attempt to elucidate factors affecting photostability, there seems to be an oversight regarding the poor inherent photostability of IR806 as a sensitizer. This aspect should be examined more thoroughly.

4. Application: The manuscript describes an energy transfer system based on upconversion luminescence and Cy7. However, the specific properties of this energy transfer have not been investigated in detail, which is crucial for understanding the system's functionality.

5. Nanoparticle Characterization: The structural characterization of the synthesized nanoparticles is inadequate. The TEM images provided do not convincingly demonstrate the quality of the micelles formed.

6. In Vivo Detection Application: The authors discuss the uptake of nanoparticles by Kupffer cells through phagocytosis, leading to accumulation in the liver. They also mention a significant signal enhancement in the liver within 1 minute. If the nanoparticles are phagocytized, how do they interact with reactive oxygen species in the liver so rapidly? Does the concentration of reactive oxygen species within the Kupffer cells increase so quickly upon reaching the liver, and if so, how is this substantial signal enhancement observed within just one minute?

Response to the comments

Reviewer #1:

I consider this revision satisfactory, the authors strongly improved the experimental part as required by several Reviewer to support their findings, including in the manuscript or in the Supporting Information file all the missing details and additional information needed to give a full picture of the proposed molecular system for NIR photon upconversion.

Before publication, I think the manuscript need a “graphical” revision. For example, in Figure 1 please take care of the details: avoid that labels overlap with the graphs’ axis, such as in panel d, etc. etc.

This is not only an aesthetic point, but more clean figures matching the journal standards, will help a lot the readers.

Response: We extend our appreciation to Reviewer #1 for recommending the publication of the revised manuscript, and for consistently providing instructive guidance to enhance the overall work quality.

Following the suggestion, we have adjusted the label position to avoid any overlap with the Figure axis. Additionally, we have made corresponding adjustments to the labels in other figures (Figures 2-5) to ensure clearer visuals.

Reviewer #4:

I thank the authors for their updates; the paper is much improved and I'm supportive of publication following addressing of the below concerns:

The upconversion emission looks substantially different from the fluorescence. Why?

Response: We express gratitude to the reviewer for acknowledging the success of our revisions and recommending the publication of our work.

We assume the reviewer's question was referring to the apparent distinction between the fluorescence spectra in Figure 1c and the upconversion emission spectra in Figure 1d. This discrepancy arises from the utilization of a low concentration ($C_{\text{BTTQD 1-3}} = 1 \times 10^{-5} \text{ M}$) in Figure 1c and a high concentration ($C_{\text{BTTQD 1-3}} = 1 \times 10^{-4} \text{ M}$) in Figure 1d. The differing concentrations lead to distinct fluorescence spectra (Figure R1a-c). However, when employing identical concentrations of $C_{\text{BTTQD 1-3}} = 1 \times 10^{-4} \text{ M}$ for both fluorescence and upconversion emission spectral measurements, two near-identical spectra were observed (Figure R1d-f). Note that the presence of IR 806 sensitizer also influences the fluorescence spectra slightly due to the existence of energy transfer from the BTTQDs annihilator to the IR 806 sensitizer (Supplementary Figure S15).

Figure R1. (a-c) The fluorescence emission spectra of BTTQD 1-3 at low ($1 \times 10^{-5} \text{ M}$) and high ($1 \times 10^{-4} \text{ M}$) concentration, respectively. (d-f) The fluorescence emission spectra of high-concentration samples of BTTQD 1-3 ($1 \times 10^{-4} \text{ M}$) with and without

the presence of IR806 (1×10^{-5} M) as well as the measured corresponding TTA-UC spectra.

Action: To avoid confusion, we substituted the fluorescence spectra of BTTQDs measured at low concentration in Figure 1c with the ones acquired at a high concentration of $c_{\text{BTTQD 1-3}} = 1 \times 10^{-4}$ M. The fluorescence spectra and upconversion emission spectra are near-identical in both Figure 1c and Figure 1d now.

Minor:

The process is “triplet-triplet annihilation” not “triplet-to-triplet annihilation) pg 2. The process is further typically accepted as two annihilator triplets meeting, not a sensitizer and annihilator triplet.

Response: We have corrected the term “triplet-to-triplet annihilation” (page 2) into “triplet-triplet annihilation” in the revised manuscript.

Reviewer #5:

I have completed my review of the manuscript titled "[Manuscript Title]" and have several comments that I believe need to be addressed to improve the clarity and scientific rigor of the study.

1. Writing Quality: The manuscript could benefit from substantial revisions to improve clarity. There are numerous misleading statements, such as the authors mentioned that “This observation is in sharp contrast to the typically investigated standard PtOEP-DPA system, where oxygen molecules was observed to enhance TTET rate by orders of magnitude.”, typically, oxygen would reduce the TTET rate from sensitizer to annihilator and quenching the upconversion emission.

Response: Thank the reviewer for the valuable suggestion to improve the clarity of our writing, which we acknowledge as important. We will address the specific comments highlighted by the reviewer as confusing in the following.

Recent studies demonstrate that in the classical PtOEP-DPA TTA-UC system, oxygen molecules significantly enhance the triplet-triplet energy transfer (TTET) rate (from PtOEP triplet to DPA triplet) by orders of magnitude (refs. 44, 54, and 55). Simultaneously, the collision between oxygen molecules and DPA molecules results in substantial nonradiative depopulation of DPA triplet states, leading to low upconversion emissions. This phenomenon is attributed to the generation of a finite transient moment for the lowest spin forbidden $T1 \rightarrow S0$ transition of the DPA triplet. Despite its counterintuitive nature, this finding represents a notable advancement, deepening the photophysics understanding of the classical PtOEP-DPA system.

Here, this comparison comment highlights the distinct role of oxygen in our TTA-UC system and the classical PtOEP-DPA system, and serves to aid readers in accurately understanding the differences between the two systems. Consequently, we intend to retain this sentence.

In addition, in the animal study, the authors mentioned that “30 mins later, PBS solution of Cy7-coated TTA-NMs (5 mg/mL, 100 μ L) were injected into the tail veins

of two Kunming mice, and dynamic upconversion emission imaging was performed (Figure 5b).”, however, in Figure 5d, the temporal changes of upconversion emission signals, the author mentioned n=3 for each group. According to my understand, the analysis of temporal changes of upconversion emission should base on the upconversion imaging. How many mice was used for the animal study? All of these could lead to confusion.

Response: We are sorry for this confusing description due to our unclear writing. This description intended to express that two representative Kunming mice, with one from the experimental group (APAP treatment) and the other one from the control group (PBS treatment), were utilized for upconversion imaging. In each group, there are three mice (n=3). For temporal changes of upconversion emission in Figure 5d, the average intensities of three mouse (mean \pm standard deviation) in each group were utilized to plot Figure 5d.

Action: To prevent misunderstanding, we have made a clearer description as follows:

page 11, paragraph 1: “After a 30-minute interval, a PBS solution containing Cy7-loaded TTA-NMs (5 mg/mL, 100 μ L) was intravenously administered to mice that had been treated with APAP (n=3), forming the experimental group. In parallel, the same injection was administered to mice without APAP treatment, establishing the control group (n=3). Subsequently, dynamic upconversion emission imaging was conducted, and Figure 5c displays representative imaging results from one mouse in each group.”

2. Stability Comparisons: The authors claim exceptional stability for their system; however, the comparisons are made under varying conditions like concentration and power. These differing conditions can significantly affect the results, rendering the data unconvincing as presented.

Response: We thank the reviewer for making the comment. We adopted two TTA-UC controls for a convincing comparison, including the established IR 806-rubrene and the classic PtOEP-DPA systems. When compared with the IR 806-rubrene control, the excitation and concentrations utilized for the two systems are identical ($C_{IR806} = 1 \times$

10^{-5} M, $c_{\text{BTTQD}} = 1 \times 10^{-4}$ M, $c_{\text{Rubrene}} = 1 \times 10^{-4}$ M, $\lambda_{\text{ex}} = 808$ nm, power density = 100 mW/cm²), giving a convincing comparison.

For the comparison with the classic PtOEP-DPA system that is well-known to the field, the adopted excitation and concentrations are indeed different ($c_{\text{PtOEP}} = 1 \times 10^{-4}$ M, $c_{\text{DPA}} = \times 10^{-2}$ M, $\lambda_{\text{ex}} = 532$ nm, power density = 5 mW/cm²), as the two systems have different absorbance and optimized parameters. However, this control was used, together with the IR806-rubne control, to illustrate that the oxygen-induced quenching is prevalent in the TTA field. While our described IR806-BTTQD 2 system is in sharp contrast with commonly seen or well-established TTA-UC systems.

3. Photostability: While the authors attempt to elucidate factors affecting photostability, there seems to be an oversight regarding the poor inherent photostability of IR806 as a sensitizer. This aspect should be examined more thoroughly.

Response: We conducted additional experiments to assess the photostability of IR806, demonstrating its consistent luminescence in both aerated and deaerated solutions under 808 nm laser irradiance at a power density of approximately 100 mW/cm² for an extended period of 180 minutes (see Figure R2). This power density aligns with common usage in investigations related to TTA-UC systems for solar and bioapplications. The confirmed photostability aligns with the sustained absorption of IR 806, as depicted in Supplementary Figure S23, and the observation of stable TTA-UC in Figure 2. This stability is primarily attributed to IR 806's ultralow triplet oxygen quenching efficiency ($\eta_{\text{O}_2} < 13\%$).

However, it is crucial to emphasize that the photostability of IR806, like other organic dyes, is contingent upon the intensity of the incident light or the excitation power density in a well-aerated solution. As power densities escalate, a gradual reduction in photostability is anticipated. This decline is attributed to the increasing generation of singlet oxygen during triplet quenching, even at low efficiencies, with the quantity of singlet oxygen rising proportionally to the exposure to light. Our recent investigation indicates that even minimal amounts of generated singlet oxygen

can induce oxidation in the double bonds of the conjugated carbon chain, leading to structural damage of IR 806 dye (Nano Lett. 2023, 23, 7001–7007). This phenomenon elucidates the observed marginal decline in photostability in the IR 806-BTTQD 2 system over an extended period (Figure 2a). To achieve sustained photostability across all excitation power densities, it is imperative to develop sensitizers with null triplet oxygen quenching efficiencies.

To maintain focus, we refrain from delving into detailed mechanisms explaining the observed slight photoinstability of IR 806 at the commonly employed power densities (approximately 100 mW/cm²). We trust that this work can serve as a catalyst, inspiring scientists to contemplate the photophysics required to attain stable TTA-UC in air. We hope the reviewer concurs with this approach.

Figure R2. The normalized emission intensities at emission peak (830 nm) of IR806 in aerated chloroform ($c = 1 \times 10^{-5}$ M) under continuous-wave 808 nm laser irradiance at 100 mW/cm² in the presence or absence of oxygen, over a period of 180 mins, respectively.

4. Application: The manuscript describes an energy transfer system based on upconversion luminescence and Cy7. However, the specific properties of this energy

transfer have not been investigated in detail, which is crucial for understanding the system's functionality.

Response: We appreciate the valuable comments from the reviewer. In the previous manuscript, we attributed the TTA-UC quenching mechanisms of TTA-NMs by Cy7 to two factors: (i) the competitive absorption of the 808 nm excitation light between Cy7 and IR 806, and (ii) the Förster resonance energy transfer (FRET) process from the annihilator BTTQD 2 to Cy7. Corresponding evidence was provided in Supplementary Figure 31 (Figure 30 in the previous SI). The reviewer's comments prompted us to recognize the necessity for a more thorough investigation into these mechanisms to elucidate the primary and secondary factors leading to this quenching.

To delve into the specific nature of energy transfer from upconversion emission to Cy7, we conducted upconversion lifetime measurements on TTA-NMs before and after loading with Cy7 (refer to Figure R3). Upon introducing Cy7 to the surface of TTA-NMs, the lifetime decreased from 13.87 to 8.94 μs . This observation clearly indicates a nonradiative FRET process with an efficiency of approximately 35.5%. However, this efficiency alone is insufficient to explain the entirety of the TTA-UC quenching effect (> 90%) in TTA-NMs (Figure 4d). Consequently, the competitive absorption effect of the excitation light between Cy7 and IR 806 should contribute to about 54.5% of the observed quenching process.

Figure R3. UC emission lifetimes of TTA-NMs in PBS (5 mg/mL) before and after

the introduction of Cy7 to the surface.

Action: We have included Figure R3 to the revised Supplementary Information (SI) as Figure S31b, and added the following sentences in the manuscript and SI.

Page 19, paragraph 1: “It is the two reasons account for the approximately 12-fold quenching of TTA-UC after Cy7 loading (Figure 4d). The FRET process is determined to be responsible for approximately 35.5% of the quenching, while the competitive absorption contributes to about 64.5% of the observed quenching effect (Supplementary Figure 31a-c).”

SI, the legend of Figure S31: “Upon introducing Cy7 to the surface of TTA-NMs, the lifetime decreased from 13.87 to 8.94 μ s, indicating the occurrence of a Förster resonance energy transfer (FRET) process with an efficiency of approximately 35.5%. However, this efficiency alone is insufficient to explain the entirety of the TTA-UC quenching effect (> 90%) in TTA-NMs (Figure 4d). The competitive absorption effect of the excitation light between Cy7 and IR 806 should contribute to about 64.5% of the observed quenching process.”

5. Nanoparticle Characterization: The structural characterization of the synthesized nanoparticles is inadequate. The TEM images provided do not convincingly demonstrate the quality of the micelles formed.

Response: The synthesized nanomicelles is a type of organic nanoparticles in nature. When utilized for bioapplications, its ability for functional fluorescence imaging is much more important than the structure itself.

However, to address the reviewer’s concern, we performed a careful TEM examination of the microstructure of TTA nanomicelles (TTA-NMs) (Figure R5). The synthesized TTA-NMs exhibit a spherical structure with high uniformity and monodispersity. The magnified TEM image of a representative individual particle consists of two regions: (1) the hydrophobic core region tightly encapsulating organic molecules (darker region), and (2) the hydrophilic shell region composed of PEO and PPO segments that shows high transparency to electron beams (brighter region in the

peripheral region). The observed nanomicelle morphology is consistent with the typical micellar morphology reported in previous studies (*ACS Appl. Mater. Interfaces* 2014, 6, 5113–5121; *Langmuir* 2020, 36, 8, 2082–2092; *Angew. Chem. Int. Ed.* 2021, 60, 26725-26733). Furthermore, the particle size distribution extracted from the TEM images indicates that the core diameter of the nanomicelles is around ~18 nm, which correlates well with the DLS (Dynamic Light Scattering) results (~17 nm).

Figure R4. (a) Transmission electron microscopy (TEM) of TTA-NMs in PBS buffer, scale bar is 50 nm. The inset represents an enlarged image of the circled particles. The white dashed circle represents the hydrophobic core region of the nanomicelle. (b) The corresponding histogram of nanomicelle sizes obtained from TEM images (~100 particles).

Action: We have incorporated Figures R4a and b as Figures 4b and c, respectively, in the revised manuscript. Additionally, the original Figure 4c has been moved to Supplementary Information (SI) as Figure S26. We also added the following discussions in the revised manuscript.

page 15 and paragraph 1: “The morphology of TTA-NMs was determined by transmission electron microscopy (TEM) to be quasi-spherical (Figure 4b). The darker region on individual particles corresponds to the dense hydrophobic core, while the brighter surrounding areas correspond to the external hydrophilic chains. This is consistent with the reported typical microscopic morphology of

nanomicelles.⁵⁹⁻⁶¹ Furthermore, the particle size distribution extracted from TEM images indicates that the core diameter of the nanomicelles is around ~18 nm (**Figure 4c**), which correlates well with the DLS (Dynamic Light Scattering) results (~17 nm, Supplementary Figure 26).”

6. In Vivo Detection Application: The authors discuss the uptake of nanoparticles by Kupffer cells through phagocytosis, leading to accumulation in the liver. They also mention a significant signal enhancement in the liver within 1 minute. If the nanoparticles are phagocytized, how do they interact with reactive oxygen species in the liver so rapidly? Does the concentration of reactive oxygen species within the Kupffer cells increase so quickly upon reaching the liver, and if so, how is this substantial signal enhancement observed within just one minute?

Response: We would like to clarify that accumulation of nanoparticles in liver organ is mainly attributed to the liver's role as a biological filtration system, which sequesters a substantial percentage (30–99%) of administered nanoparticles from the bloodstream. The underlying mechanism for this phenomenon involves several processes including not only Kupffer cell phagocytosis, but also the presence of fenestrations in liver sinusoidal endothelial cells, and distinctive blood flow patterns and microcirculation within the liver (*Journal of Controlled Release* 240 (2016) 332–348).

In our experimental protocol, 30 minutes prior to the formal injection of the probe, an overdose of APAP (0.05 mM, 100 μ L) is initially administered via intraperitoneal injection to establish a mouse model of liver toxicity. In other words, a significant amount of ONOO⁻ had already been generated in the liver of the mice about 30 mins before intravenous injection of TTA-NMs. The observation of turn-on upconversion luminescence in liver within 1 minute should indicate a rapid accumulation of TTA-NMs in liver through the rapid blood circulation process, and reports the presence of ONOO⁻ in the liver organ. Moreover, the ONOO⁻ detection is supposed to take place at the organ level where inflammation is widespread throughout the entire liver organ, rather than at the cellular level.

Action: To avoid confusion, we have deleted sentences related to Kupffer cell phagocytosis, and the description on the ability to perform minute-scale temporal resolution of imaging. Moreover, we added the following clarifications in the revised manuscript

page 21 and paragraph 1: “After intravenous injection into mice, nanomedicines commonly tend to accumulate in the liver. This inclination is attributed to the liver's role as a biological filtration system, which sequesters a substantial percentage (30–99%) of administered nanoparticles from the bloodstream.^{64, 69} This feature also enables passive accumulation of the developed Cy7-coated TTA-NMs nanoprobe (~20 nm) into the liver organ”.

page 21 and paragraph 1: “After a 30-minute interval, a PBS solution containing Cy7-loaded TTA-NMs (5 mg/mL, 100 μ L) was intravenously administered to mice that had been treated with APAP (n=3), forming the experimental group. In parallel, the same injection was administered to mice without APAP treatment, establishing the control group (n=3). Subsequently, dynamic upconversion emission imaging was conducted (**Figure 5b**), and **Figure 5c** displays representative imaging results from one mouse in each group.”

REVIEWERS' COMMENTS

Reviewer #4 (Remarks to the Author):

My query has been satisfactorily resolved.

Reviewer #5 (Remarks to the Author):

The author has essentially addressed all the questions and publication is acceptable.

Response to the comments

Manuscript ID: NCOMMS-22-47037C

Title: Molecular Near-Infrared Triplet-Triplet Annihilation Upconversion with Eigen Oxygen Immunity

Reviewer #4 (Remarks to the Author):

My query has been satisfactorily resolved.

Response: We are delighted that the reviewer found our responses satisfactory and appreciate all the valuable comments provided during the revision process.

Reviewer #5 (Remarks to the Author):

The author has essentially addressed all the questions and publication is acceptable.

Response: We express our appreciation to the reviewer for finding the revised files very satisfactory and recommending them for publication in the current form.